# An integrated and homogenized global surface solar radiation dataset and its reconstruction based on a convolutional neural network approach

Boyang Jiao[1,#], Yucheng Su[2], Qingxiang Li*[1,#], Veronica Manara[3],Martin Wild[4]

[1]School of Atmospheric Sciences, Sun Yat-sen University, and Key Laboratory of Tropical Atmosphere−Ocean System, Ministry of Education, Zhuhai 519082, China

[2]Meteorological Bureau of Zhuhai, Zhuhai 519082, China

[3]Department of Environmental Science and Policy, Università degli Studi di Milano, via Celoria 10, 20133, Milano, Italy

[4]Institute for Atmospheric and Climate Science, ETH Zurich, Zurich, Switzerland

[#]Southern Laboratory of Ocean Science and Engineering (Guangdong Zhuhai), Zhuhai 519082, China

*Correspondence to:* Qingxiang Li (liqingx5@mail.sysu.edu.cn)

**Abstract**

Surface solar radiation (SSR) is an essential factor in the flow of surface energy, enabling accurate
capturing of long-term climate change and understanding the energy balance of Earth's atmosphere
system. However, the long-term trend estimation of SSR is subjected to significant uncertainties due to
the temporal inhomogeneity and the uneven spatial distribution of the *in situ* observations. This paper
develops an observational integrated and homogenized global-terrestrial (except for Antarctica))
stational SSR dataset ($SSRIH_{station}$) by integrating all available SSR observations, including the existing
homogenized SSR results. The series is then interpolated in order to obtain a $5°×5°$ resolution gridded
dataset ($SSRIH_{grid}$). On this basis, we further reconstruct a long-term (1955-2018) global land (except for
Antarctica) SSR anomalies dataset with a $5°×2.5°$ resolution ($SSRIH_{20CR}$) by training improved partial
convolutional neural network deep learning methods based on the reanalysis 20CRv3. Based on this, we
analysed the global land (except for Antarctica) /regional scale SSR trends and spatiotemporal variations:
The reconstruction results reflect the distribution of SSR anomalies and have high reliability in filling
and reconstructing the missing values. At the global land (except for Antarctica) scale, the decreasing
trend of the $SSRIH_{20CR}$ (-1.276 ± 0.205 W/m$^2$ per decade) is smaller than the trend of the $SSRIH_{grid}$ (-
1.776 ± 0.230 W/m$^2$ per decade) from 1955 to 1991. The trend of $SSRIH_{20CR}$ (0.697 ± 0.359 W/m$^2$ per
decade) from 1991 to 2018 is also marginally lower than that of the $SSRIH_{grid}$ (0.851 ± 0.410 W/m$^2$ per
decade). At the regional scale, the difference between the $SSRIH_{20CR}$ and $SSRIH_{grid}$ is more significant
in years and areas with insufficient coverage. Asia, Africa, Europe and North America cause the global
dimming of the $SSRIH_{20CR}$, while Europe and North America drive the global brightening of the
$SSRIH_{20CR}$. Spatial sampling inadequacies have largely contributed to a bias in the long-term variation
of global /regional SSR. This paper's homogenized gridded dataset and the Artificial Intelligence
reconstruction gridded dataset (Jiao and Li, 2023) are all available at
https://doi.org/10.6084/m9.figshare.21625079.v1.

**1 Introduction**

Energy flows at the Earth's surface play an essential role in climate change and human activity and link to physical processes such as global warming, glacier retreating, hydrological cycle, and carbon budget (Hoskins and Valdes, 1990; Peixoto et al., 1992; Trenberth and Fasullo, 2013; Wild, 2012). As a critical factor characterizing surface energy flows, Surface Solar Radiation (SSR) largely determines the climatic conditions and ecological environment in which we live. Therefore, a more accurate and comprehensive analysis of the SSR fluxes will help better understand the Earth's atmospheric system. *In situ* observations provide the most accurate baseline data for measuring SSR. They allowed for the first time the detection of decadal changes in SSR known as "dimming and brightening" (Wild et al., 2005), especially considering that they cover a longer period concerning another type of data like for example satellite data (Pfeifroth et al., 2018). Even observational data often have uneven distribution and missing data with respect to the satellite data, especially in areas with complex orography (Manara et al., 2020).

The sources of *in situ* SSR observations are mainly collected from the Global Energy Balance Archive (GEBA) (Wild et al., 2017) and the World Radiation Data Centre (WRDC) (Tsvetkov et al., 1995). Furthermore, other SSR station series are obtained from the high quality Baseline Surface Radiation Network (BSRN) (Driemel et al., 2018) and the data centres of individual national hydrometeorological services. However, two issues still need to be addressed: 1) the inhomogeneity of station data resulting from station relocations and instrumentation changes severely impacts the climate change assessment. For the regions with a relatively high density of stations, like Europe (Manara et al., 2019; Manara et al., 2016; Sanchez-Lorenzo et al., 2013a; Sanchez-Lorenzo et al., 2015; Sanchez-Lorenzo et al., 2013b), Japan (Ma et al., 2022) and China (Ju et al., 2006; Wang, 2014; Wang et al., 2015; Wang and Wild, 2016; Yang et al., 2018b; You et al., 2013), much previous work has redefined the degree and timing of "dimming and brightening" by addressing the inhomogeneity of the SSR data series. For example, in Spain, the average annual homogenized SSR series has a significant increasing trend (+ 3.9 W/m$^2$ per decade) during the 1985–2010 period (Sanchez-Lorenzo et al., 2013a). The period of dimming observed in Italy's homogenized SSR series is not apparent in the 1960s and early 1970s when the raw series (inhomogenized) are taken into account (Manara et al., 2016). The direct measurements of SSR show a level trend from 1961 to 2014 over Japan, while their homogenization series display a decreasing trend (0.8-1.6 W/m$^2$ per decade) (Ma et al., 2022). In China, homogenization largely eliminated the dramatic

non-climatic rise of the early 1990s and also reduced the increasing trend from 1990 to 2016 (Yang et
al., 2018b). However, most of the research was still limited to regional scales. 2) The issue of limited
spatial sampling of long observational stations and their uneven distribution especially over areas with
complex orography. Considerable efforts have been devoted to filling in /interpolating the missing values
in climate datasets ("spatial analysis") (Collins, 1996; Erxleben et al., 2002; Scudiero et al., 2016). The
traditional spatial interpolation methods commonly used include Inverse Distance Weighted (Fisher et
al., 1993; Shepard, 1968), Kriging (Krige, 1951), Thin-Plate Splines (Bookstein, 1989) et cetera. Since
the 1980s, physical parametric interpolation (Feng and Wang, 2021; Tang et al., 2019) and Bayesian
fusion schemes (Aguiar et al., 2015) based on multi-source observational data were widely used, when
the emergence of highly accurate and relatively precise satellite data. However, the resulting fusion
datasets cover a too short period to investigate their decadal and multi-decadal variations and to study
the underlying causes. The spatial, temporal, and spectral coverage of a single satellite is limited, and
multiple satellite data are therefore often used in tandem with each other; however, such a
discontinuity in time and space can introduce inhomogeneity into a dataset (Evan et al., 2007; Feng
and Wang, 2021; Shao et al., 2022). Reanalysis products are an important complement containing
long-term SSR data, therefore have been widely used in climate studies (Huang et al., 2018; Jiao et
al., 2022; Urraca et al., 2018; Zhou et al., 2018a; Zhou et al., 2017) due to the dynamically consistent
and spatiotemporally complete atmospheric fields with high resolution and open access to data.
However, existing studies have shown that reanalysis products generally overestimate multi-year
mean SSR values compared to observations over land (He et al., 2021). With the continuous
development of climate system simulations, model data from the Coupled Model International
Program (CMIP) have become an important resource for conducting climate change research (Gates
et al., 1999; Zhou et al., 2019). Previous studies have shown that the models used in CMIP6
overestimate the global mean SSR (He et al., 2023; Jiao et al., 2022; Wild, 2020). The rise of deep
learning and big data techniques has brought about an explosion of artificial intelligence (AI). Machine
learning is increasingly being used in spatial interpolation, such as the spatial reconstruction of surface
temperature datasets (Huang et al., 2022; Kadow et al., 2020; Cao et al., 2022), the spatial and temporal
reconstruction of turbulence resolution (Fukami et al., 2021), etc. Furthermore, it shows high accuracy
and low uncertainty in reproducing and predicting SSR (Leirvik and Yuan, 2021; Tang et al., 2016; Yang
et al., 2018a; Yuan et al., 2021). However, long-term homogenized SSR datasets with global terrestrial
coverage have yet to be developed, resulting in significant uncertainties in assessing global SSR variation
(Jiao et al., 2022).
Therefore, developing a more homogeneous and comprehensive global long-term SSR climatic dataset
that provides a better benchmark for observational constraints on the global surface energy balance
/budget remains a valuable and challenging task. This paper first homogenizes and grids the most
extensive collection of available global SSR station observations. Then, the missing grid boxes /years
are spatially interpolated using a convolutional neural network (CNN) approach to obtain a globally
covered land surface SSR anomalies dataset. Finally, the reconstructed datasets are initially analysed and
evaluated. Thus, the paper is divided into seven main sections. The data resources are introduced in
Section 2.  Section 3 presents the data homogenization, and the CNN model reconstruction methods. The
data homogenization and verification are shown in Section 4. Section 5 gives the AI reconstruction results.
Section 6 is the availability of the datasets. Conclusions are provided at the end of the paper.
**2 Data**
Nine SSR datasets are collected to derive the global SSR variable. In particular, six datasets contain data
from observational stations (Section 2.1): two global ground-based measurement datasets (GEBA,
WRDC) and four homogenized products at regional and country levels (Europe, China, Japan and Italy).
Three of the adopted datasets are reanalysis data (Section 2.2.1): Fifth generation European Centre for
Medium-Range Weather Forecasts (ECMWF) reanalysis (ERA5), 20th Century Reanalysis version 3
(20CRv3) reanalysis data and the Coupled Model Intercomparison Project Phase 6 (CMIP6) historical
simulation output (125). Specifically, the ERA5 data are used to fill the data over oceans and Antarctica
(Section 3.2.1), 20CRv3 data and CMIP6 simulations are used for the AI model training (Section 5.1)
and reconstruction. All have been listed in Table 1.
**2.1 In situ observational Data**
**2.1.1 Global datasets**
There are two main sources of raw SSR data (see Table 1): the ETH Zurich GEBA with monthly data
from 2,445 globally distributed stations, starting from 1922 until 2020, and the WRDC dataset with
monthly globally distributed data from 1136 stations since 1964. The first one is available for download

at https://geba.ethz.ch (Last access: 2022.7. 2). The second one published the first SSR radiation balance data in 1965 and then its publication has been issued four times a year since 1993 and is available for download at http://wrdc.mgo.rssi.ru/ (Last access: July 2021).

**2.1.2 National (regional) homogenized station datasets**

1) Chinese homogenized SSR dataset

The China Meteorological Radiation Fundamental Elements Monthly Value Data Set has been downloaded at http://www.nmic.cn. The homogenized SSR dataset in China is released by the National Meteorological Information Centre (NMIC), China Meteorological Administration (CMA) (Yang, 2016). The data are available for the period between Jan 1950 to Dec 2014, and the follow-up data are extended with raw observations from NMIC. They used the sunshine duration (SSD) data from nearby stations to construct an arguably better reference to identify inhomogeneities in the SSR data. Then, a combined metadata and the maximum penalty t-test (PMT) method was used to detect the change points. Finally, they were adjusted by a quantile matching (QM) algorithm (Wang and Feng, 2013). The final homogenized SSR station dataset was converted to gridded data using the first difference method (FDM (Peterson et al., 1998)) and is available for download at http://www.nmic.cn. Last Access: September 2022.

2) Japanese homogenized SSR dataset

Ma et al. (Ma et al., 2022) released a Japanese SSR homogenized dataset in 2022 spanning the period between 1870 and 2015. First, they homogenized SSD based on PMF (penalized maximal F test) and QM algorithms. They then used the homogenized SSD from the previous step as a reference series, combined with metadata and PMT, to detect change points. Finally, they adjusted the change points by the QM algorithm. For more details on data descriptions, the adopted methodology and downloading data refer to https://data.tpdc.ac.cn/en/data/45d73756-3f5a-4d27-82a4-952e268c20e8/, Last Access: March 2022.

3) European homogenized SSR data

A homogenized dataset of European SSR stations was developed by Sanchez-Lorenzo et al. (Sanchez-Lorenzo et al., 2015) and is currently available as a full public download at https://agupubs.onlinelibrary.wiley.com/doi/full/10.1002/2015JD023321. They selected the 56 longest Central European SSR series available in GEBA dataset with data for the period comprised between

1922 and 2012. They adjusted them to ensure temporal homogeneity homogenizing the data with the
Standard Normal Homogeneity Test (Alexandersson, 1986) and the Craddock test (Craddock, 1979).
4) Italian homogenized SSR dataset
The Italian homogenized SSR datasets are those published by (Manara et al., 2019; Manara et al.,
2016). As candidate stations to use as reference series, they selected the ten series located in the same
area of the series to be tested and that series correlate well with the test one. In particular, they tested the
change points with the Craddock test (Manara. et al., 2017) and when a break is identified by more than
one reference series the preceding portion of the series is corrected, leaving the most recent portion
unchanged. In this way, the SSR stations were homogenized, and then the missing values were
interpolated.
**2.2 Other datasets**
**2.2.1 Reanalysis**
ERA5 can be used to fill in SSR data from the oceans and Antarctica and carry out the global
reconstruction, taking into account its high spatial resolution and reliable performance of SSR (Jiao et
al., 2022; Liang et al., 2022). After the reconstruction, we removed the data for the ocean reanalysis and
maintain the data only in the land area (except for Antarctica). In addition, two SSR data products
(20CRv3, CMIP6) are used to train AI models. These are:
1) ERA5 (space-filling data): ERA5 is the fifth generation of the European Centre for Medium-Range
Forecasting reanalysis product, which currently publishes data from 1950 to the present (Hersbach et al.,
2020). In addition, ERA5 has an hourly output and an uncertainty estimate from the ensemble. The data
is based on the Integrated Forecasting Model Cy41r2 run in 2016, which contains a 4D-Var assimilation
scheme. In ERA5, SSR is obtained from a Rapid Radiation Transfer Model (RRTM) (Mlawer et al.,
1997). The present study utilizes monthly SSR data for the period 1955-2018 from ERA5 with a
resolution of 0.25 ° ×0.25 ° (last accessed in July 2022). It can be downloaded at
https://cds.climate.copernicus.eu
2) 20CRv3 (data for AI model training): The 20CR Project is an effort led by NOAA's Physical
Sciences Laboratory and CIRES at the University of Colorado, supported by the Department of Energy,
to produce reanalysis datasets spanning the entire 20th century and much of the 19th century (Slivinski
et al., 2019). 20CR provides a comprehensive global atmospheric circulation data set from 1850 to 2015.
Its chief motivation is to provide an observational validation dataset, with quantified uncertainties, for
assessing climate model simulations of the 20th century. 20CR uses an ensemble filter data assimilation
method which directly estimates the most likely state of the global atmosphere every three hours and
estimates the uncertainty in that analysis. The most recent version of this reanalysis, 20CRv3, provides
8-times daily estimates of global tropospheric variability across 75 km grids, spanning 1836 to 2015
(with an experimental extension from 1806 to 1835). The present study uses monthly SSR data of
20CRv3 (NOAA /CIRES /DOE 20CR, 80 members) from 1955-2015. We selected all 80 members of
the 20CR as input (1 for evaluation and to test reconstruction, the other 79 for training the CNN model).
The SSR of 20CRv3 has a spatial resolution of 0.7°×0.7° (Last accessed: May 2022). The download is
available at https://portal.nersc.gov/archive/home/projects/incite11/.

**2.2.2 CMIP6 models output**

3) CMIP6 models output (data for AI model training): the Coupled Model Intercomparison Project,
driven by the World Climate Research Program, is now in its 6th phase. Specifically, CMIP6 is
considered as the current state of the art way of producing future climate simulations, including predicting
future SSR based on different climate scenarios (Zhou et al., 2018b). It provides an important resource
for studying current and future climate change (Eyring et al., 2016). The historical simulations of CMIP6
are designed to reproduce observed climate and climate change, constrained by radiative forcing. Its
historical simulation spans between 1850 and 2014. In this study, we selected 125 members out of a total
of 507 members from several CMIP6 large ensemble models (with more than 10 realizations/runs) with
high correlation coefficients with observations as input to train and validate the CNN model (1 for
evaluation and to test reconstruction, the other 124 for training the CNN model). We selected the monthly
downward shortwave radiation from 1955 to 2014 (see Table S1 in the Supplemental Material (SM)).
Last access July 2022. Download at: https://esgf-node.llnl.gov/search/cmip6.

**3 Methods**

**3.1 Data Quality Control (QC) and homogenization**

The SSR data homogenization method is only applied to the two inhomogenized *in situ* observations datasets (GEBA and WRDC). The Quality Control (QC) and homogenization flowchart (Figure 1) is divided into three steps: 1. QC; 2. Homogenization; 3. Integration and consolidation.

**3.1.1 QC**

The QC of SSR data includes the following steps:

1) Simple integration: integration of the GEBA (2445) and WRDC (1136) datasets removing stations with no data and leaving 2681 stations.

2) Removing duplicate stations: a. Stations with similar latitude and longitude. We consider two stations with totally identical latitude and longitude to be the same station; b. Stations less than 10km apart. We averaged the duplicate stations in this a and b case; c. Special duplicate stations: Stitching together data of the duplicate stations based on metadata from CMA.

3) Remove stations or years /months for which a climatic analysis cannot be established: we remove stations with records of less than ten years and values more than three times ($3\sigma$ criterion (Olanow and Koller, 1998) the standard deviation of the SSR anomalies.

4) Candidate stations (487) with a record length greater than 15 years in the period 1971-2000 are selected. We added stations (715) with more than 10 years of SSR records to increase the number of available stations for a better homogenization of the candidate stations (Figure 2).

**3.1.2 Station series homogenization**

This paper uses the RHtestV4 software package to test and adjust the SSR station data for homogeneity (http://etccdi.pacificclimate.org/software.shtml) (Wang and Feng, 2013). The package is based on the empirical penalty functions PMF (Wang, 2008a) and PMT (Wang, 2008b; Wang et al., 2007) for the homogenization test. It takes into account the lag-1 autocorrelation of the time series. It embeds a multiple linear regression algorithm to significantly reduce the problem of an unbalanced distribution of pseudo-identification rates and test efficacy. Also, RHtestV4 uses the QM algorithm (Vincent et al., 2012; Wang et al., 2010) and Mean-Adjustments to adjust the identified change points.

The specific steps are as follows:

1) Building the reference series
a. We processed the data from all stations series (715) into the annual first differences (FD) series
$e_i$(Eq. (1)) (Peterson et al., 1998).
b. We calculated the correlation of the annual FD series between the series from the potential reference
pool and the candidate stations.
c. We calculated the distance between the potential reference pool stations and candidate stations.
d. We selected potential stations according to the correlation coefficient (CC >= 0.6) between the series
from potential reference pool and candidate stations. And the potential stations also satisfy the limits in
distances (<= 500km) between the potential pool stations and candidate stations.
e. We obtain the reference FD series ($Re$)based on the $m$ potential reference series (Pe$_i$) and the CCs
($c_i$) between the potential reference series (Pe$_i$) and candidate stations series (Eq. (2)).
f. The synthesized reference FD series ($Re$) (Eq. (2)), plus the average of all potential reference series
($\bar{R}$), yields the final annual reference series ($R$) (Eq. (3)).

$$e_i = x_i - x_{i+1}$$

$$\text{i=1, 2, ..., n-1} \tag{1}$$

$$R_e = \frac{\sum_{i=1}^{m} Pe_i * c_i^2}{\sum_{i=1}^{m} c_i^2} \tag{2}$$

$$R = R_e + \overline{R} \tag{3}$$

$e_i$   Annual FD series,
$x_i$   Raw observational station SSR in the year i,
$Re$   Final reference series,
$Pe_i$   Potential reference series,
$c_i$   CC between the potential reference series and the candidate stations series.
2) Testing and adjusting the candidate series
The homogenization test algorithm used in this paper is the PMT. This method is a reference series-
dependent test for a normalized candidate series. It assumes that the linear trend of the time series is zero
and uses the degree of mean deviation at different points in the series to find change points. Furthermore,
it eliminates the effect of different sample lengths on the test results. At the same time, the method
introduces an empirical penalty factor, which effectively improves detection. We used the PMT to test
the homogeneity of the candidate series based on the reference series established in 1). We then adjusted
the statistically significant(p>0.05) changepoints obtained using the mean adjustment method (p>0.05).
We homogenize the monthly series for 66 stations (see Figure S1 in the SM).

**3.1.3 Integration and consolidation**

As can be seen from Figure 1, the candidate stations (487) are relatively sparse. To better adapt deep
learning methods for the dataset reconstruction later, we adjusted, added and integrated station series
based on the results of homogenized data from other scholars: 1) We added stations with more than 10a
overall (1955-2018) records but no more than 15a during the 1971-2000 period, and removed those
stations that were clearly inhomogeneous (25) and some years of the station (3); 2) We subsequently
integrate monthly SSR series for 116 stations based on the results of homogenization by other scholars
(China (56), Japan (8), Europe (2) and Italy (50)). After the above steps, we end up with a homogenized
dataset containing 944 stations (Figure 3). The details of the processing and classification are shown in
Table S2 (see in the SM).

**3.2 CNN model reconstruction methods**

The CNN deep learning model network architecture uses a U-shaped structure similar to the U-net
(Ronneberger et al., 2015). The advantage of using this model is: 1) both high and low-frequency
information of the picture can be retained, and when reconstructing the SSR data, not only the grid point
information close to the missing measurement point will be considered, but also information from more
distant locations (which may be remotely correlated with that missing measurement point); 2) This makes
the model convergence faster and more economical in terms of computational resources. The upper part
of the U-shaped structure, which has no down samples or a low number of down samples, represents the
high-frequency information of the graph. These sections contain much of the detail in the graph and the
relationships between similar grid points are conveyed by this section. The lower half of the U-shaped
structure is down-sampled more often and represents the lower frequency information of the graph. The
global radiation of a wide range of undulations is transmitted by it, and then the information at the various
levels of the U-shaped structure is connected and transmitted through the skip connection, allowing the
whole network to remember all the information of the picture very well. The model uses nearest
neighbour upsampling in the decoding phase, the skip links will concatenate two feature maps and two
masks as the feature and mask inputs for the next part of the convolution layer. The input to the last part
of the convolution layer will contain the original input image concatenated with the holes and the original
mask, allowing the model to replicate the gap-free pixels. The complex and variable nature of the sea-
land boundary then has a significant impact on the reconstruction, when we reconstruct the global land
SSR data. Therefore, we use partial convolution at the image boundaries with a suitable image padding,
ensuring that the padding content at the image boundaries is not affected by values outside the image.
The deep learning models' convolutional layers and loss functions have been described in the SM.
We further reconstruct a long-term (1955-2018) global SSR anomalies dataset ($SSRIH_{20CR}$) by using
improved partial CNN deep learning methods based on a "perfect" dataset. CNN consists of three parts.
A convolutional layer to reduce the number of weights by extracting local features, a pooling layer to
reduce peacekeeping and prevent overfitting, and a fully connected layer to output the desired result. In
this paper, a modified CNN network is used to model the reconstruction of the SSR data, with the
convolutional layer replaced by a partial convolution method and mask update. This method is the latest
in image restoration effects and can restore irregular holes, an advantage over other image restoration
methods that can only restore rectangular holes. Therefore, this paper uses the modified CNN model
(Kadow et al., 2020) to recover the missing part of the global terrestrial SSR (except Antarctica). The
specific reconstruction steps and processes are as in Figure 4.
**3.2.1 Data pre-processing**
The homogenized station data is converted to grid box anomalies using the Climate Anomalies Method
(CAM) (Jones et al., 2001). CAM is a commonly used method for converting station anomaly data to
gridded data. We divide all global areas into a $5° \times 5°$ grid, after which we calculate the SSR anomalies
(relative to 1923-2020) within the grid box by averaging the anomalies of all stations (at least 1 station
in it). If there are more than one site exists in the same grid box, the record length of this grid box is the
total length of all sites in that grid box. Finally, we removed the values that were more than three times
the standard deviation of the SSR anomaly time series after gridding. SSRs are all processed as daily
average anomalies, i.e., monthly anomalies divided by 30 (each month is approximated as 30 days). We
multiplied all the values by 30 again when the reconstruction is complete. The global land (except for
Antarctica) distribution and coverage of SSRs after gridding are shown in Figure 5 a, b.
As seen in Figure 5a, the SSR is spatially sparsely distributed across South America and Africa. As
shown in Figure 5b, SSR coverage increased yearly from 1950 until the mid-1970s, when it slowly
decreased. In 2013, the coverage rate decreased sharply due to untimely data submission. Considering
the SSR coverage above, we only kept the years (1955-2018) with data coverage of more than 8% of
global land (except for Antarctica) areas.
Comparisons show that the ERA5 has high spatial resolution and relatively reliable performance in
the temporal variations and long-term trends (Liang et al., 2022; Jiao et al., 2022). To obtain a higher
data coverage and ensure that the AI model runs well, we used the ERA5 to fill the SSR of homogenized
global gridded SSR in the Antarctic and ocean areas. However, if we use the SSR of ERA5 to directly
fill the SSR of homogenized global gridded SSR (SSRIH$_{grid}$) in the Antarctic and on the ocean areas,
then the relatively weaker ocean SSR variations (variabilities, decadal changes, trends, etc.) from ERA5
will inevitably introduce certain systematic biases in land SSR reconstruction due to the SSRs have the
lower coverage on the land. Therefore, we designed an algorithm to avoid excessive diffusion of SSR
system bias in terrestrial areas: we first calculated the ratios $\gamma_i$ ($i=1, 2, 3, ...., n$) between the SSR from
ERA5 and from SSRIH$_{grid}$ on the land in all $n$ years. For a single grid box, the $\gamma_i$ have small changes
and are regarded as a constant $\gamma_{median}$ (Eq (4)), and the $\gamma_{median}$ vary by latitude and longitude both on
the marine and the land areas. We then extrapolated the $\gamma_{median}$ for all the grid boxes along the land
and sea boundaries. If there is no observation there, then the adjacent ocean ERA5 SSR is used to take
its place after it is adjusted according to the differences between the SSR variations (represented by the
linear trends) for the different underlying surfaces (Eq (5).

$$\gamma_{median} = Median(\frac{OBS_{i\_land}}{ERA5_{i\_land}}), \tag{4}$$

$$OBS_{i\_O\&L}(land) = ERA5_{i\_O\&L}(Ocean) * \gamma_{median} * \frac{T_O}{T_L}, \tag{5}$$

$$i = 1,2,3......,n$$

$\gamma_{median}$: The median value of the ratios of OBS and ERA5 land SSR series,
$OBS_{i\_land}$: Land SSR for the year i from SSRIH$_{grid}$ in a single grid,
$ERA5_{i\_land}$: Land SSR for the year i from ERA5in a single grid,
$OBS_{i\_O\&L}(land)$: LandSSRalong the sea-land boundary(land) for the year $i$ from SSRIH$_{grid}$,
$ERA5_{i\_O\&L}(Ocean)$: Ocean SSR along the sea-land boundaryfor the year $i$ from ERA5,
$T_O$: Trend of ERA5 SSR on ocean areasin all $n$ years,
$T_l$: Trend of ERA5 SSR on areas in all $n$ years.

**3.2.2 AI Model reconstruction**

We use a server (configured with processor Intel (R) Core (TM) i7-8700 CPU @ 3.20GHz 3.19 GHz, RAM 32G, 64-bit OS, GPU model 516.94, NVIDIA GeForce 1080T version, Python 3.9.12 64-bit, CUDA 10.1) for AI models training. The specific training steps are as follows:

1) A total of 768 missing value masks (monthly masks between 1955 and 2018) were prepared for training and validation using '1' for existing and '0' for missing values;

2) The 20CRv3 /CMIP6 training set (monthly values between 1955 and 2015 /2014) and missing value masks are fed into the 20CR-AI /CMIP6-AI model for training;

3) We perform 1,500,000 training sessions with an interval of 10,000 sessions for the training output model.

Afterwards, the two AI models are validated against the root mean squared error (RMSE) /CCs of the reconstructed SSRs ($SSR_{20CR}$/$SSR_{CMIP6}$). The validation set SSRs, and the optimal number of training cycles is 1,100,000 (see Figure S2, Figure S3 and Figure S4 in the SM). The initial hyper-parameters of the model are set as follows; learning rate of 2e-4 and learning finetune of 5e-5. First, we set the batch size to 16 in the first 500000 iterations and fine-tuned it to 18 in the last 10000000 iterations, for a total of 1500000 iterations, to suppress the overfitting phenomenon generated during the training process, and validate the model every 10000 times and early stopping if the validation shows a decreasing trend, the final number of training times used is 1100000. Second, L2 regularization is also added to regulate the loss function (see Eq. (9) in the SM).

The training result models generated by the different AI models are obtained separately for the different training sets. The model is first used to reconstruct a reanalysis validation set with the same missing value mask as the original observation dataset. This is followed by a validation of the reconstruction against the original reanalysis dataset (calculation of CC and RMSE) to understand the discrepancies in the model reconstruction.

**4 Data homogenization and verification**

We homogenized the original monthly stations /gridded SSR time series ($SSRIH_{station}$ /$SSRIH_{grid}$) using the method in section 3.1.2. We selected six continental regions, excluding Antarctica and the Arctic, from the eight continents of the world defined by Xu et al. (Xu et al., 2018) (Asia, Africa, South America,

Europe, North America, Australia, Antarctica and the Arctic). The decreasing trend of the $SSRIH_{grid}$ is
consistent with the original gridded SSR series ($SSRI_{grid}$) during 1955-1991 while the increasing trend
during 1991-2018 is weaker. At the regional scale, the $SSRIH_{grid}$ has a generally similar variation to the
$SSRI_{grid}$, and the $SSRIH_{grid}$ usually more representative of climate change than $SSRI_{grid}$ at individual
stations.
Figure S5 (see in the SM) illustrates the long-term variations of global (Figure S5 (a) in the SM) and
continental land SSR (Figure S5 (b) in the SM) from the $SSRI_{grid}$ and $SSRIH_{grid}$ (except for Antarctica)
during 1955-2018. The most prominent change revolves around the adjustment around 1992: the SSR
anomalies were systematically adjusted upward from 1987 to 1992, while the SSR anomalies were
systematically adjusted downward from 1993 onwards. Thus, there is a significant decreasing trend for
both global land $SSRI_{grid}$ (-1.995 $\pm$ 0.251 $W/m^2$ per decade) and global land $SSRIH_{grid}$ (-1.776 $\pm$ 0.230
$W/m^2$ per decade) (except for Antarctica) from 1955 to 1991. While the increasing trend of the global
land $SSRIH_{grid}$ from 1991 to 2018 is 0.851 $\pm$ 0.410 $W/m^2$ per decade, slightly smaller than the increasing
trend of the $SSRI_{grid}$ (0.999 $\pm$ 0.504 $W/m^2$ per decade). It is worth noting that 1992 happened to be the
second year of the eruption of Mount Pinatubo, and the homogenized SSR data integrated in this paper
may be affected by this event. But overall, the homogenization also has limited effects on the global SSR
variations from Figure S5 (see in the SM), which is consistent with the influence of data homogenization
on a wide range of surface air temperatures (Brohan et al., 2006; Xu et al., 2013).
At the regional scale, the differences between the $SSRIH_{grid}$ and $SSRI_{grid}$ are more pronounced in Asia
and Europe (see Figure S5(b)in the SM). Asia's homogenized SSR show that the regional average SSR
has been declining significantly over the period 1958-90; this dimming trend mostly diminished over the
period 1991-2005 and was replaced by a brightening trend in the recent decade. The $SSRIH_{grid}$ in Asia is
higher than the $SSRI_{grid}$ from 1985 to 1990 and lower than the $SSRI_{grid}$ from 2012 to 2015. The $SSRIH_{grid}$
shows a more moderate short-term increase in Europe from 1960 to 1980. Note also that the Australian
raw data prior to 1988 were artificially detrended because at the time the Australia Weather Service was
afraid that the instruments would drift. Therefore, they detrended them and unfortunately did not store
the raw data, and the SSR evolution in Australia is artificial with no trend (Wild et al., 2005). In addition,
the $SSRI_{station}$ and $SSRIH_{station}$ comparisons for all 66 stations are shown in Figure S1 (see in the SM).

**5 AI reconstruction and comparison**

**5.1 Training of the AI model**

We produce two (20CRv3 /CMIP6) separate training and validation sets: we select the 1th member data of the reanalysis data and the model data, respectively, as the validation set, and the remaining 79 (124) ensemble members as the training sets, where each ensemble member included 732 (720) months of SSR data. Each validation set included 732 (720) samples, while the training sets contained 57828 (89280) ensemble members. All the above data, including the *in situ* observations, are then resampled to monthly anomalies of $5° \times 2.5°$.

We reconstruct the SSR of 20CRv3 /CMIP6 with missing values based on 20CRv3 /CMIP6 datasets using the method in section 3.2 and obtain two reconstructions, $SSR_{20CR}$ and $SSR_{CMIP6}$, respectively. The SSR of 20CRv3/CMIP6 with missing values uses the $SSRIH_{grid}$ mask between 1955 and 2015 /2014. We compare the global land (except for Antarctica) /regional annual anomalies variation of $SSR_{20CR}$ /$SSR_{CMIP6}$. The results show that $SSR_{20CR}$ is significantly more consistent with the validation set than $SSR_{CMIP6}$.

Figure 6(a) shows that the RMSE/CC of the $SSR_{20CR}$ (0.247 W/m$^2$ /0.970 W/m$^2$) are smaller /larger than those of $SSR_{CMIP6}$ (0.518 W/m$^2$ /0.937 W/m$^2$) with the original 20CR /CMIP6 dataset. The 20CR-AI model has a better reconstruction ability for SSR at the global land (except for Antarctica) scale. The RMSEs of the $SSR_{20CR}$ ($SSR_{CMIP6}$) are 1.460 (2.413) W/m$^2$, 1.109 (1.829) W/m$^2$, 2.219 (2.596) W/m$^2$ and 1.286 (2.235) W/m$^2$ in North America, Europe, Asia, and Northern Hemisphere, whereas these values are 1.116 (1.766) W/m$^2$, 0.622 (1.602) W/m$^2$, 1.877 (1.839) W/m$^2$ and 0.772 (1.679) W/m$^2$ in South America, Africa, Australia, and Southern Hemisphere concerning the original 20CR /CMIP6 dataset, respectively. In other words, the RMSEs of the $SSR_{20CR}$ are smaller than those of $SSR_{CMIP6}$for the original 20CR /CMIP6 dataset except for Australia. In addition, the CCs of the $SSR_{20CR}$ ($SSR_{CMIP6}$) are 0.958 (0.830) W/m$^2$, 0.958 (0.987) W/m$^2$, 0.886 (0.669) W/m$^2$, 0.930 (0.965) W/m$^2$, 0.938 (0.930) W/m$^2$, 0.943 (0.916) W/m$^2$, 0.936 (0.875) W/m$^2$ and 0.903 (0.822) W/m$^2$ in North America, Europe, Asia, Northern Hemisphere, South America, Africa, Australia, and Southern Hemisphere with respect to the original 20CR /CMIP6 dataset, respectively. That is, the CCs of the $SSR_{20CR}$ are larger than those of $SSR_{CMIP6}$to the original 20CR /CMIP6 dataset except for Europe.

Based on the above comparison, the higher uncertainty for CMIP6 model output possibly biases the
CMIP6-AI method. Thus, the accuracy of the $SSR_{20CR}$ is higher than that of the $SSR_{CMIP6}$ at both global
land (except for Antarctica) and regional scales. Therefore, we choose the reconstruction results of the
20CR-AI model as the final AI reconstruction dataset, and subsequent analysis in the following sections
is only based on this dataset.

**5.2 Comparison of the spatial and temporal variation characteristics**

We investigate the long-term trends and spatial and temporal variation of the $SSRIH_{20CR}$, compare the
differences between the $SSRIH_{20CR}$ and $SSRIH_{grid}$, and suggest: the area and magnitude of the high and
low centres of the $SSRIH_{20CR}$ are the same as those of the $SSRIH_{grid}$; the results of the global land (except
for Antarctica) reconstruction are consistent with "dimming and brightening"; the global dimming is
primarily dominated by decreasing trends in Asia, Europe Africa and North America, whereas Europe
and North America are contributors to the increasing trends.
Figure 7 shows the spatial distribution of the $SSRIH_{grid}$ and $SSRIH_{20CR}$ for the three months (July 1960,
July 1980, and July 2000). Figure S6 (see in the SM) displays the spatial distribution of annual $SSRIH_{grid}$
and $SSRIH_{20CR}$ from 1955 to 2018. Figure 7 also shows the area and the magnitude of the high and low
centres in the $SSRIH_{20CR}$ are the same as in the $SSRIH_{grid}$. The $SSRIH_{20CR}$ is mainly positive anomalies
in Africa and the Eurasian continent in July 1960, especially in India and the Middle East. Afterwards,
India showed a continuous and steady decline in SSR. This confirms the well-known phenomenon of
global dimming over India (Wild et al., 2009; Soni et al., 2016; Soni et al., 2012; Padma Kumari et al.,
2007; Kambezidis et al., 2012). In Australia, the $SSRIH_{20CR}$ is dominated by negative anomalies in July
1980 and positive anomalies in July 1960 and July 2000. In Greenland, the $SSRIH_{20CR}$ shows a large
positive anomaly during three months. In northern Russia, there is a high value in July 2000. The
reconstruction can better reflect the anomaly distribution of observation information, and the grid boxes
with the missing values are infilled and reconstructed, which has high reliability.
Figure 8 illustrates global land (except for Antarctica) annual anomalies variation and long-term trend
of the $SSRIH_{20CR}$ for the period of 1955-2018, 1955-1991 and 1991-2018. Table S3 in the SM
demonstrates the trends of global SSR change evaluation for various data sources on different scales.
Also, we compare the differences between the $SSRIH_{20CR}$ and $SSRIH_{grid}$. The minimum value of the
$SSRIH_{20CR}$ occurred in 1991 (-2.411 W/m$^2$). The decreasing trend of the $SSRIH_{20CR}$ from 1955 to 1991
(-1.276 $\pm$ 0.205 W/m$^2$ per decade) is slightly lower than that of the SSRIH$_{grid}$ (-1.776 $\pm$ 0.230 W/m$^2$ per
decade). After that, the SSRIH$_{20CR}$ turns to an increasing trend of 0.697 $\pm$ 0.359 W/m$^2$ per decade from
1991 to 2018. This suggests that the difference between SSRIH$_{20CR}$ and SSRIH$_{grid}$ may be caused by the
results observed in limited data coverage (such as in Africa and North America) (Figure 9). After
homogenization and reconstruction, the trend (-1.276 W/m$^2$ per decade) from 1955 to 1991 corresponds
to an overall reduction of -4.6 W/m$^2$ over the dimming period, while that (0.697 W/m$^2$per decade) from
1991 to 2018 correspond to an overall increase of 2 W/m$^2$ over the brightening period. This is in amazing
agreement with the -4 W/m$^2$ for the dimming period and the 2 W/m$^2$ for the brightening period based on
an overall surface energy budget assessment ((Wild, 2012) see their Figure 1). Also, similar conclusions
(incomplete coverage of observational data lead to an underestimation of global warming trends) have
been confirmed in global warming research (Gulev et al., 2021; Li et al., 2021).

Figure 9 demonstrates the long-term annual anomaly variations of the SSRIH$_{20CR}$ in different regions

and its results compared to the SSRIH$_{grid}$. Table S4 in the SM shows the evaluation in continental and
hemispheric SSRIH$_{20CR}$ change trends on different scales. The SSRIH$_{20CR}$ shows a similar annual
anomaly variation to the global land (except for Antarctica) average trend in North America and Asia,
reaches a minimum in the late 1970s or early 1990s, and follows a moderate reversal. In Europe, the
SSRIH$_{20CR}$ shows a decrease (-2.180 $\pm$ 1.866 W/m$^2$ per decade) between 1963 and 1978 before turning
to brightening (1.081 $\pm$ 0.312 W/m$^2$ per decade). In South America and Australia (Southern Hemisphere),
the SSRIH$_{20CR}$ shows no significant variation. In Africa, the SSRIH$_{20CR}$ has a dimming trend (-1.506 $\pm$
0.496 W/m$^2$ per decade) from the 1950s to the 1990s, after which it remains levelled off (0.340 $\pm$ 0.998
W/m$^2$ per decade). The SSRIH$_{20CR}$ shows a decreasing trend (-1.457 $\pm$ 0.246 W/m$^2$ per decade) until the
1990s in the Northern Hemisphere and a brightening (0.887 $\pm$ 0.415 W/m$^2$ per decade) afterwards. The
annual average anomaly variations in regions and globally show that Asia, Africa, Europe and North
America are the four contributors to the global dimming, while Europe and North America are two major
contributors to the "brightening". This is in general agreement with the results obtained by previous
machine learning (Yuan et al., 2021). In addition, the discrepancy between the SSRIH$_{20CR}$ and SSRIH$_{grid}$
is more significant in low-coverage areas (right) than in high-coverage regions (left). It is particularly
pronounced before 1980 and in South America. This suggests that the limited surface observations are
not representative of the continental variation in SSR.

The sources of error in the observational dataset can be divided into three types: (1) station error, the

uncertainties of individual station anomalies; Including measurement errors (which are not the focus of
the considerations in this manuscript) and errors due to homogenization. The errors due to
homogenization adjustment are always approximately normally distributed ((Jones et al., 2008), see
their Figure 5; also see Figure S9 in the SM) and therefore have limited impacts on the global average
SSR change (Figure S5 a, b). (2) sampling error, the uncertainties in a grid box mean caused by
estimating the mean from a small number of point values (Jones et al., 1997) ; and (3) bias error. It
generally refers to systematic errors such as urbanization together, which has not been discussed here.
However, even the sum of the above errors is much smaller than the errors due to limited data coverage
((Li et al., 2010), see their Figure 5). So, the focus of this study is to eliminate this kind of error through
the CNN reconstruction.
**6 Data availability**
Both the $SSRIH_{grid}$ (the homogenized monthly gridded SSR data over 1923-2020) and the $SSRIH_{20CR}$
(the monthly 20CR-AI model reconstructed SSR data for 1955-2018) are currently publicly available on
the figshare website under DOI at https://doi.org/10.6084/m9.figshare.21625079.v1 (Jiao and Li, 2023).
These datasets are also available at http://www.gwpu.net for free.
**7 Conclusion**
In this study, we integrate global station observations based on the raw observational SSRs from GEBA
and WRDC, combined with existing homogenized SSR datasets from other scholars. Also, we
homogenize the globally distributed station data using the RHtestV4 software package. An improved
CNN deep learning algorithm is subsequently used to reconstruct the SSR anomalies. Thus, a
reconstructed SSR anomaly dataset, $SSRIH_{20CR}$, is obtained based on training sets (20CRv3), for the
years 1955-2018, with a resolution of $5°×2.5°$. The main results are as follows:
1) The first integrated and homogenized global SSR monthly dataset is developed, which contains 944
stations in total and covers the longest periods (from the 1920s to recent years). A $5°×5°$ grid boxes
version of the monthly SSR anomalies dataset is derived.
2) This paper develops $5°×2.5°$ full-coverage monthly land (except for Antarctica) SSR anomalies
reconstructed datasets based on the above observations, using the 20CRv3 to train the AI model.
Comparative validations /evaluations show that the $SSRIH_{20CR}$ provides a reliable benchmark for global
SSR variations.
3) On average, the global annual SSR variations based on the $SSRIH_{grid}$ are not significantly different,
except that the increasing (brightening) trend after 1991 is a little smaller for the latter. The short-term
brightening SSR in Europe from the 1970s- to the 1980s disappear at the regional scale. At the same time,
the brightening SSR after the 1990s in Asia slowed or postponed.

**Author contributions**

Boyang Jiao: Software, Data curation, Writing- Original draft preparation, Visualization, Investigation.
Yucheng Su: Software, Data curation.
Qingxiang Li: Methodology, Supervision, Conceptualization, Validation, Writing - Review and Editing.
Veronica Manara: Providing the homogenized Italian dataset, Writing - Review and Editing.
Martin Wild: Writing - Review and Editing.

**Competing interests**

At least one of the (co-) authors is a member of the editorial board of Earth System Science Data.

**Disclaimer**

Publisher's note: Copernicus Publications remains neutral about jurisdictional claims in published maps
and institutional affiliations.

**Financial support**

This study is supported by the Natural Science Foundation of China (Grant: 41975105) and the National
Key R&D Program of China (Grant: 2018YFC1507705; 2017YFC1502301). The Global Energy
Balance Archive (GEBA) is co-funded by the Federal Office of Meteorology and Climatology
MeteoSwiss within the framework of GCOS Switzerland. Global dimming and brightening research at
ETH Zurich are supported by the Swiss National Science Foundation (Grant No. 200020 188601).
Veronica Manara was supported by the "Ministero dell'Università e della Ricerca" of Italy [grant FSE –
REACT EU, DM 10/08/2021 n. 1062].

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

**Captions of tables and Figures**
Table 1: List of information on the various types of data used in this paper.
Figure 1: Flowchart of quality control (QC) (first step), homogenization (second step) and integration
(third step).
Figure 2: Spatial distribution of candidate stations ("*") and added stations ("+"). The different colour
bars represent the length of the station record in months (Units: Month).
Figure 3: Spatial distribution of stations after homogenization (Units: Month), different colours
represent the length of station records in months
Figure 4: Flowchart of AI reconstruction.
Figure 5: (a) Spatial distribution of 5°x5°grid boxes (SSRIH$_{grid}$) obtained interpolating the
homogenized global land (except for Antarctica) SSR series. The different colours represent the length
(the sum of all records) of the station record, Units: Year. (b) Grid box coverage for the homogenized
global land (except for Antarctica) SSR (SSRIH$_{grid}$) except for Antarctica.
Figure 6: Reconstruction capabilities of the AI model.
Figure 7: Spatial distribution of the SSRIH$_{grid}$ (a1-3) and SSRIH$_{20CR}$ (b1-3) in typical months. 1-3 is
July 1960, July 1980, and July 2000, respectively.
Figure 8: Global land (except for Antarctica) time series of the annual anomaly variations SSR (relative
to 1971-2000) before/after reconstruction.
Figure 9: Same as Figure 8, but for regional annual anomaly variations.

**Table 1: List of information on the various types of data used in this paper**

|  | Abbreviation | Resolution | Time | Reference |
|---|---|---|---|---|
| *In situ*-Raw | GEBA (Station) | Monthly | 1922-2020 | (Wild et al., 2017) |
|  | WRDC (Station) | Monthly | 1964-2017 | (Tsvetkov et al., 1995) |
| *In situ*-Homo | China (Station) | Monthly | 1950-2016 | (Yang et al., 2018b) |
|  | Japan (Station) | Monthly | 1870-2015 | (Ma et al., 2022) |
|  | Europe (Station) | Monthly | 1922-2012 | (Sanchez-Lorenzo et al., 2015) |
|  | Italy (Station) | Monthly | 1959-2016 | (Manara et al., 2016; Manara et al., 2019) |
| Reanalysis / Model | ERA5 (Grid) | Monthly/ 0.25°×0.25° | 1950-2020 | (Hersbach et al., 2020) |
|  | 20CRv3 (Grid) | Monthly/ 0.7°×0.7° | 1940-2015 | (Slivinski et al., 2019) |
|  | CMIP6 (Grid) | Monthly/- | 1940-2014 | (Eyring et al., 2016) |


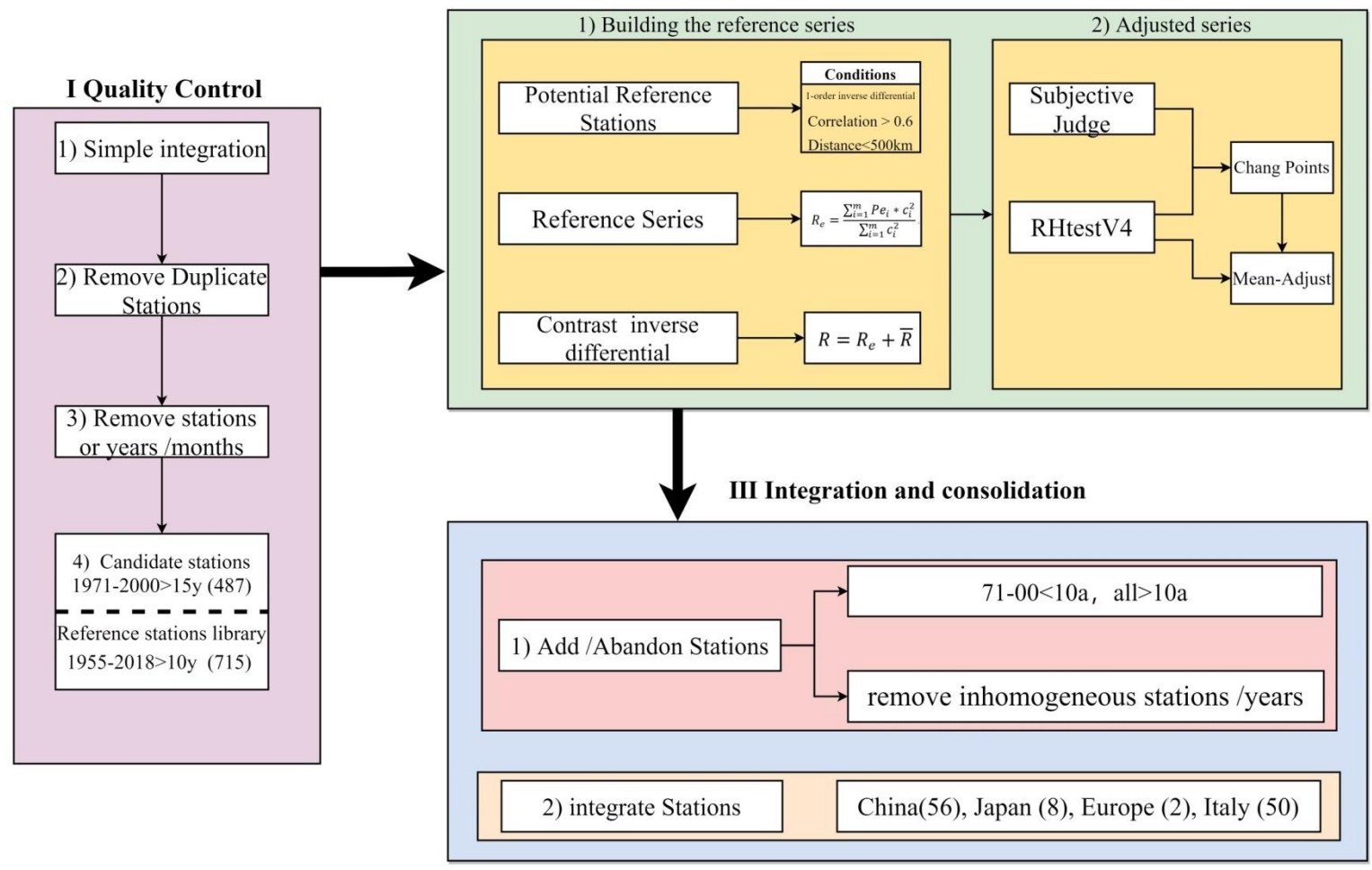


**Figure 1: Flowchart of quality control (QC) (first step), homogenization (second step) and integration (third step).**


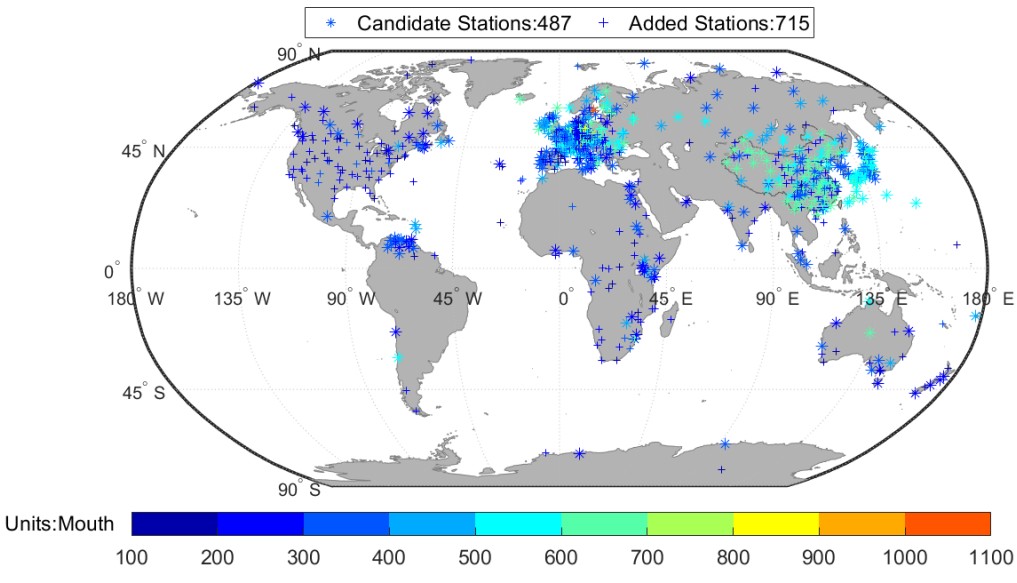

**Figure 2: Spatial distribution of candidate stations ("*") and added stations ("+"). The different colour bars**
**represent the length of the station record in months (Units: Month).**

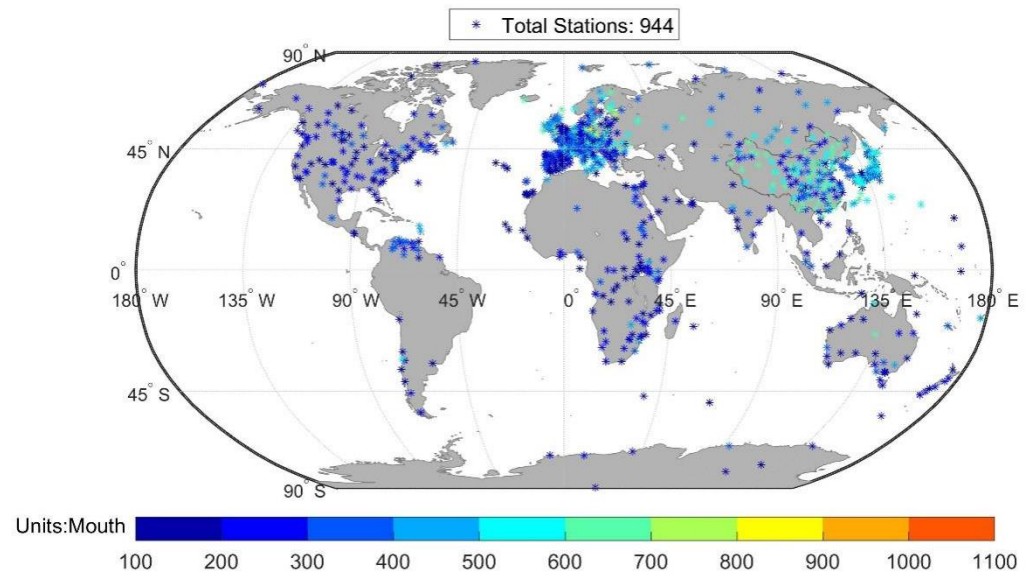

**Figure 3: Spatial distribution of stations after homogenization (Units: Month), different colours represent the**
**length of station records in months.**

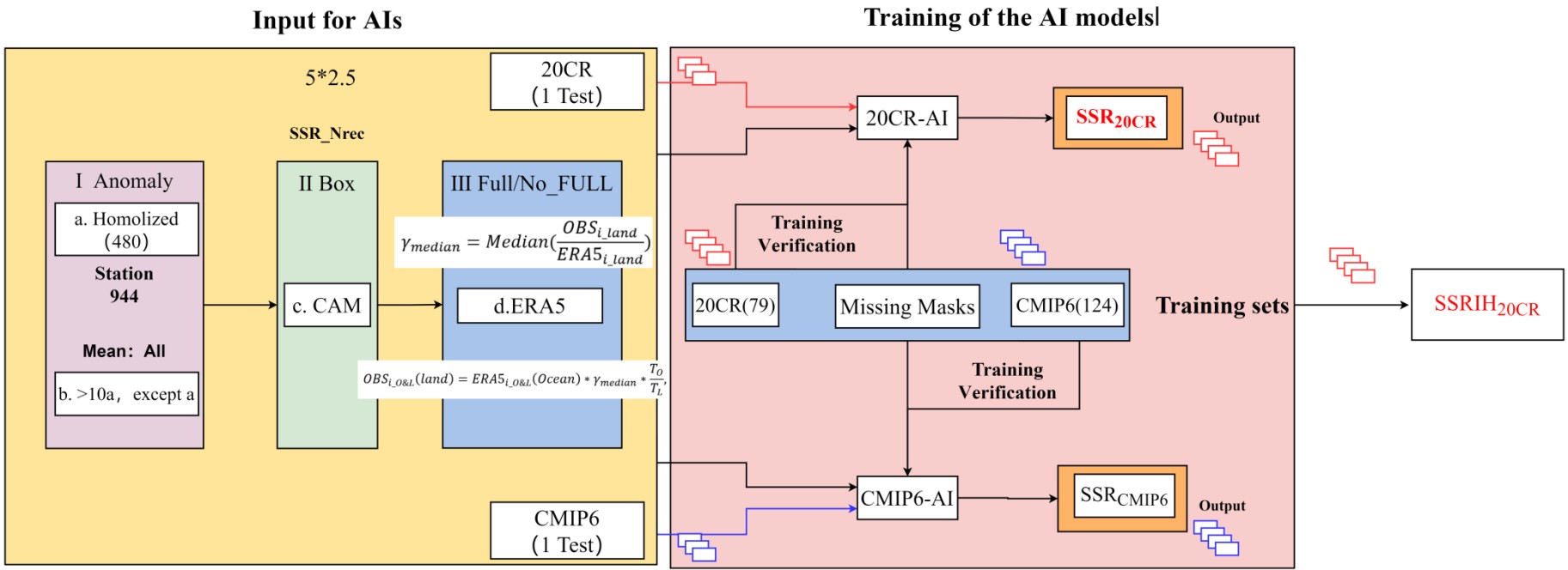


**Figure 4: Flowchart of AI reconstruction.**

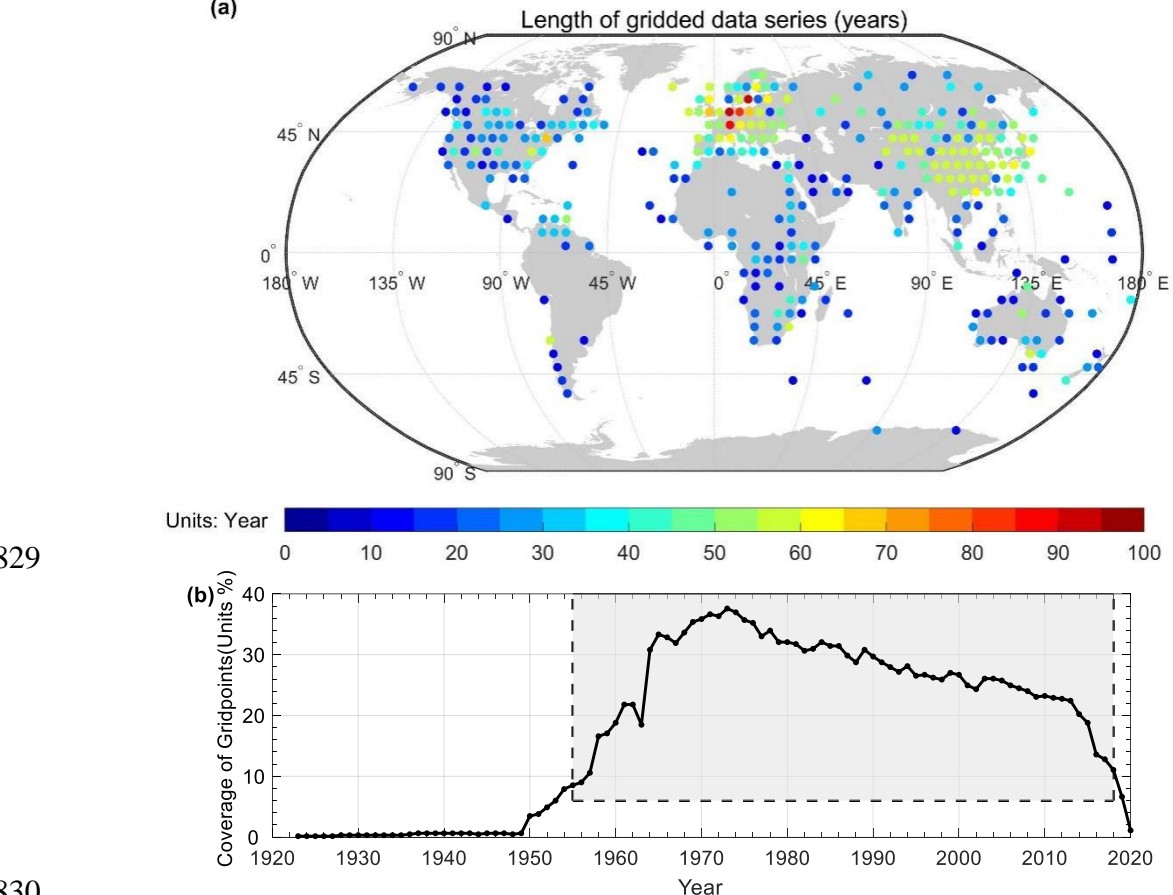



**Figure 5: (a) Spatial distribution of 5°×5°grid boxes (SSRIH$_{grid}$) obtained interpolating the homogenized**
**global land (except for Antarctica) SSR series. The different colours represent the length (the sum of all**
**records) of the station record, Units: Year. (b) Grid box coverage for the homogenized global land (except**
**for Antarctica) SSR (SSRIH$_{grid}$) except for Antarctica.**

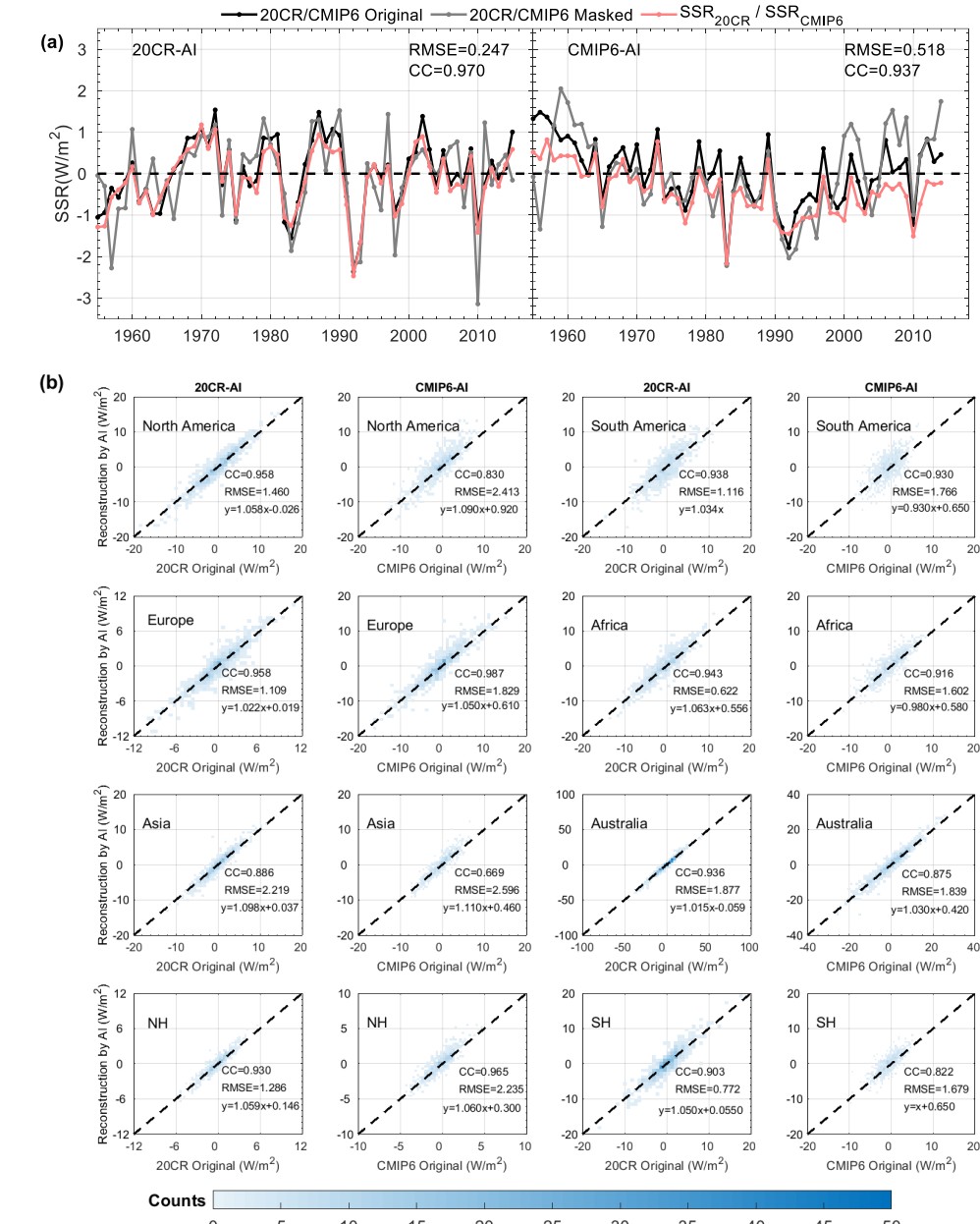



**Figure 6: Reconstruction capabilities of the AI model. (a) Global land (except for Antarctica) means time-series analysis and AI model reconstruction evaluation. The red line is the SSR of the reconstruction based on the 20CR-AI /CMIP6-AI model (SSR$_{20CR}$ /SSR$_{CMIP6}$); The grey line is the masked datasets with missing values of the SSRIH$_{grid}$. The solid black line is the 20CR and CMIP6 validation set (the SSR from the 1th member of 20CRv3 /CMIP6). (b) Comparisons of the SSR$_{20CR}$ (columns 1, 3) /SSR$_{CMIP6}$ (columns 2, 4) with the SSR from the 20CR and CMIP6 validation set. Colour bars represent counts with the same values for both. Figures also show the SSR$_{20CR}$ (SSR$_{CMIP6}$) correlation coefficient (CC), root mean squared error (RMSE) and fitting equation compared to the original dataset in different regions.**

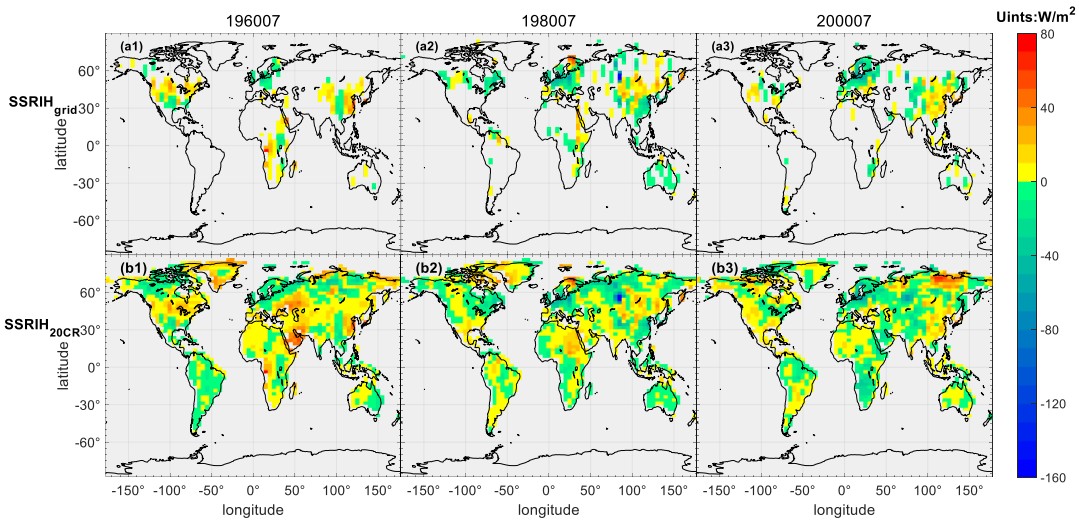


**Figure 7: Spatial distribution of the SSRIH$_{grid}$ (a1-3) and SSRIH$_{20CR}$ (b1-3) in typical months. 1-3 is July**
**1960, July 1980, and July 2000, respectively.**

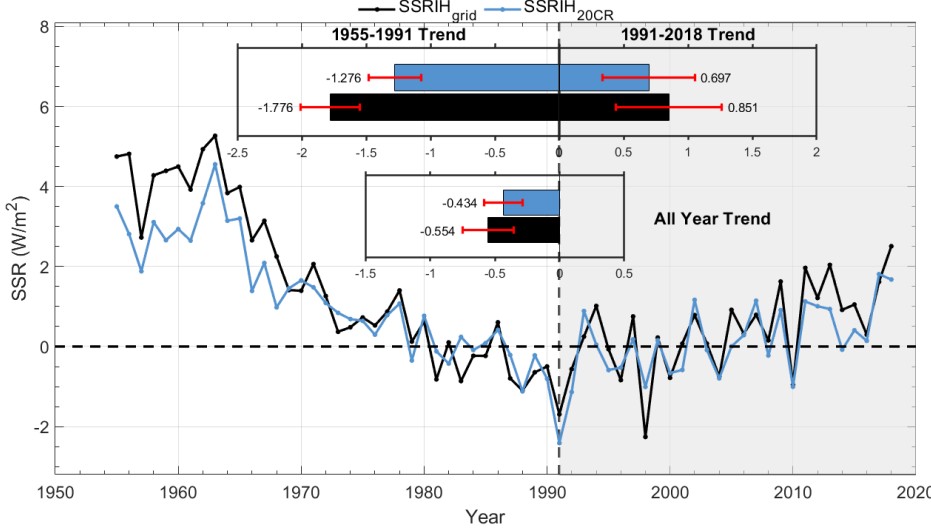


**Figure 8: Global land (except for Antarctica) annual SSR anomaly variations (relative to 1971-2000)**
**before/after reconstruction. The Black solid line represents the SSRIH$_{grid}$ annual anomalies. The solid blue**
**line represents the SSRIH$_{20CR}$ annual anomalies. The histograms represent the decadal trends of the**
**SSRIH$_{grid}$ /SSRIH$_{20CR}$ (unit: W/m$^2$ per decade) and their 95% uncertainty range from 1955 to 1991, 1991-**
**2018 and 1955-2018.**

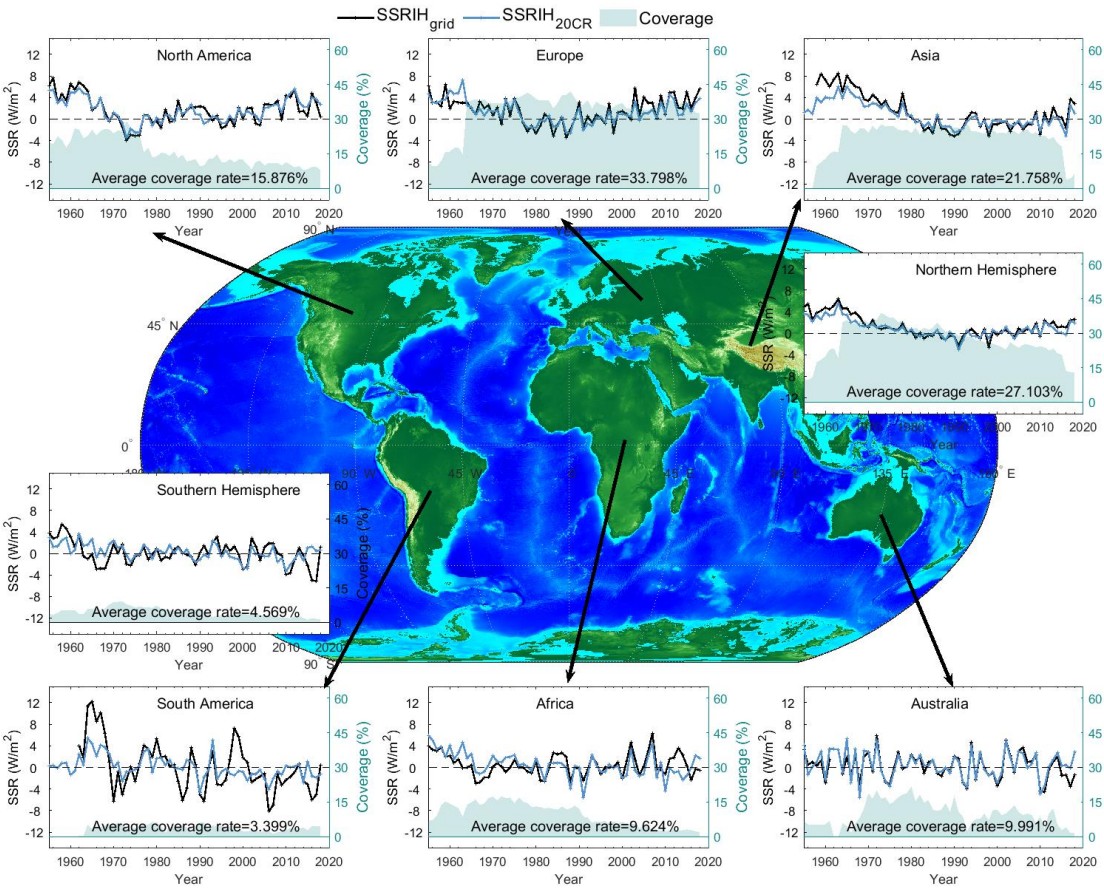


Figure 9: Same as Figure 8, but for regional annual anomaly variations. The green colour filling diagram

represents the variation in grid box coverage (before reconstruction).

# Supplemental Material to

# 'An integrated and homogenized global SSR dataset and its reconstruction based on a convolutional neural network approach'

Boyang Jiao[1,#], Yucheng Su[2], Qingxiang Li*[1, #], Veronica Manara[3], Martin Wild[4]

[1]School of Atmospheric Sciences, Sun Yat-sen University, and Key Laboratory of Tropical Atmosphere−Ocean System, Ministry of Education, Zhuhai 519082, China
[2]Meteorological Bureau of Zhuhai, Zhuhai 519082, China
[3]Department of Environmental Science and Policy, Università degli Studi di Milano, via Celoria 10, 20133, Milano, Italy
[4]Institute for Atmospheric and Climate Science, ETH Zurich, Zurich, Switzerland
[#]Southern Laboratory of Ocean Science and Engineering (Guangdong Zhuhai), Zhuhai 519082, China

*Correspondence to:* Qingxiang Li (liqingx5@mail.sysu.edu.cn)

**The SI file contains:**

1 Text (S1)

3 Table (S1, S3, S4)

8 Figures (S1(1-11), S2, S3, S4, S5, S6 (1-16), S7 and S8)

**Text S1 Convolutional Neural Network (CNN) deep learning model (convolutional layer, loss**
**function)**
Convolutional layer using partial convolution and mask update: The partial convolution operation and
the mask update function are called the partial convolution layer (Liu et al., 2018). The partial
convolution operation and the mask update function are called the partial convolution layer. The partial
convolution at each position can be expressed as

$$x' = \begin{cases} W^T \left( X \odot M \right) \dfrac{sum(1)}{sum(M)} + b, & if \; sum(M) > 0 \\ 0, & otherwise \end{cases} \qquad (1)$$

$\odot$ denotes element-by-element multiplication, where 1 and $M$ in the above equation have the same shape,
and all elements in 1 are 1. Eq. (1) illustrates that our output value depends only on the valid input and
that $\frac{sum(1)}{sum(M)}$ is used to adjust the amount of change in the valid value of the input.

$$m' = \begin{cases} 1, & if \; sum(M) > 0 \\ 0, & otherwise \end{cases} \qquad (2)$$

After each partial convolution operation, use equation (2) to update the mask Eq. (2) indicates that we
mark that position as valid whenever the convolution can adjust its output according to at least one valid
value. In other words, marking 1 where there is a value and 0 for the default part is the so-called binary
mask. This approach can be implemented in any deep learning structure as part of a forward delivery.
With enough partial convolutions, the input values will all eventually become valid, i.e., any masks will
all become 1. Partial convolution layers can be implemented by extending the existing standard Pytorch
library. The most straightforward implementation is to define a binary mask of the shape $C \times H \times W$
that is the same size as its associated image and feature values. And then, update the mask using a fixed
convolutional layer of the same size and operation as the partial convolutional layer, with the same weight
(weight of 1) and no bias.
The model loss function is set for each pixel reconstruction accuracy and the transition smoothness of
the repaired missing measurements to their surroundings. Let the input image be $I_i$, the initial binary
mask be $M$, the predicted value be $I_{out}$, and the actual value be $I_{gt}$. Eq. (3) and Eq. (4) calculate the loss
value for each pixel, where Eq. (3) calculates the default value portion of the loss value and Eq. (4)
calculates the actual value portion of the loss value.

$$\mathcal{L}_{hole} = ||(1 - M) \odot \left( I_{out} - I_{gt} \right)||_1 \qquad (3)$$

$$\mathcal{L}_{valid} = ||M \odot (I_{out} - I_{gt})||_1 \tag{4}$$

Define the Perceptual Loss function (Eq. (5)) and the Style Loss function (Eq. (6) and (7). Where $I_{comp}$ denotes the original data, where the valid value is the true value and $K_n$ denotes the normalization factor.

$$\mathcal{L}_{perceptual} = \sum_{n=0}^{N-1} ||\Psi_n(I_{out}) - \Psi_n(I_{gt})||_1 + \sum_{n=0}^{N-1} ||\Psi_n(I_{comp}) - \Psi_n(I_{gt})||_1 \tag{5}$$

$$\mathcal{L}_{style_{out}} = \sum_{n=0}^{N-1} ||K_n((\Psi_n(I_{out}))^T(\Psi_n(I_{out})) - (\Psi_n(I_{gt}))^T(\Psi_n(I_{gt})))||_1 \tag{6}$$

$$\mathcal{L}_{style_{comp}} = \sum_{n=0}^{N-1} ||K_n((\Psi_n(I_{comp}))^T(\Psi_n(I_{comp}) - (\Psi_n(I_{gt}))^T(\Psi_n(I_{gt})))||_1 \tag{7}$$

Finally, the Total Variation Loss function is defined in equation (8). This loss function effectively smoothes the image, reducing the total variation of the signal and removing unwanted details while retaining essential details such as edges.

$$\mathcal{L}_{tv} = \sum_{(i,j)\in P,(i,j+1)\in P} ||I_{comp}^{i,j+1} - I_{comp}^{i,j}||_1 + \sum_{(i,j)\in P,(i+1,j)\in P} ||I_{comp}^{i+1,j} - I_{comp}^{i,j}||_1 \tag{8}$$

First, we set the batch size to 16 in the first 500000 iterations and fine-tuned it to 18 in the last 10000000 iterations, for a total of 1500000 iterations, to suppress the overfitting phenomenon generated during the training process, and validate the model every 10000 times and early stopping if the validation shows a decreasing trend, the final number of training times used is 1100000. Second, L2 regularization is also added to regulate the loss function. The initial hyper-parameters of the model are set as follows; learning rate of 2e-4 and learning finetune of 5e-5.

The final loss function equation (9) is constructed by combining all the loss functions necessary for image restoration, and a validation set of 100 images confirms this equation's hyperparameters.

$$\mathcal{L}_{total} = \mathcal{L}_{valid} + 6\mathcal{L}_{hole} + 0.05\mathcal{L}_{perceptual} + 120\left(\mathcal{L}_{style_{out}} + \mathcal{L}_{style_{comp}}\right)$$
$$+ 0.1\mathcal{L}_{tv} + \alpha||\omega||_2^2 \tag{9}$$

**Table S1: CMIP6 numerical models for training the neural network. CMIP6 Historical monthly experiments between 1955 and 2014 are applied to train the CMIP6-AI.**

| | Source ID | N° | Ensemble |
|---|---|---|---|
| 1 | ACCESS-ESM1-5 | 40 | r1i1p1f1-r40i1p1f1 |
| 2 | CNRM-CM6-1 | 30 | r1i1p1f2-r30i1p1f2 |
| 3 | CNRM-ESM2-1 | 11 | r1i1p1f2-r11i1p1f2 |
| 4 | EC-Earth3 | 22 | r1i1p1f1-r4i1p1f1; r6i1p1f1; r7i1p1f1; r9i1p1f1; r10i1p1f1-r19i1p1f1; r21i1p1f1-r25i1p1f1 |
| 5 | EC-Earth3-CC | 10 | r1i1p1f1; r4i1p1f1; r6i1p1f1-r13i1p1f1 |
| 6 | MRI-ESM2-0 | 12 | r1i1p1f1-r10i1p1f1; r1i2p1f1; r1i1000p1f1 |


**Table S3 Trends and their 95% confidence ranges in various data sources global SSR change (units:**
**W/m² per decade). * Indicate trends that are significant at the 5% level.**

| Type | 1955-1991 | 1991-2018 | 1955-2018 |
|---|---|---|---|
| $SSRI_{grid}$ | -1.995 ± 0.251* | 0.999 ± 0.504* | -0.494 ± 0.228* |
| $SSRIH_{grid}$ | -1.776 ± 0.230* | 0.851 ± 0.410* | -0.554 ± 0.197* |
| $SSRIH_{20CR}$ | -1.276 ± 0.205* | 0.697 ± 0.359* | -0.434 ± 0.148* |
| ERA5 | -1.162 ± 0.319* | 0.653 ± 0.350* | -0.180 ± 0.176* |


**Table S4 Trends and their 95% confidence ranges in continental and hemispheric SSRIH$_{20CR}$**
**change (Units: W/m$^2$ per decade). $*$ Indicate trends that are significant at the 5% level.**

| Continental | Time period /Trend | Time period /Trend |
|---|---|---|
| North America | 1955-1973 | 1973-2018 |
| | -3.588 ± 1.290* | 1.074 ± 0.278* |
| South America | 1955-1990 | 1990-2018 |
| | -0.408 ± 0.619 | 0.049 ± 0.768 |
| Europe | 1963-1978 | 1978-2018 |
| | -2.180 ± 1.866* | 1.081 ± 0.312* |
| Africa | 1955-1991 | 1991-2018 |
| | -1.506 ± 0.496* | 0.340 ± 0.998 |
| Asia | 1955-1990 | 1990-2018 |
| | -1.633 ± 0.473* | 0.435 ± 0.505 |
| North Hemisphere | 1955-1991 | 1991-2018 |
| | -1.457 ± 0.246* | 0.887 ± 0.415* |
| South Hemisphere | 1955-1991 | 1991-2018 |
| | -0.708 ± 0.330* | -0.076 ± 0.656* |


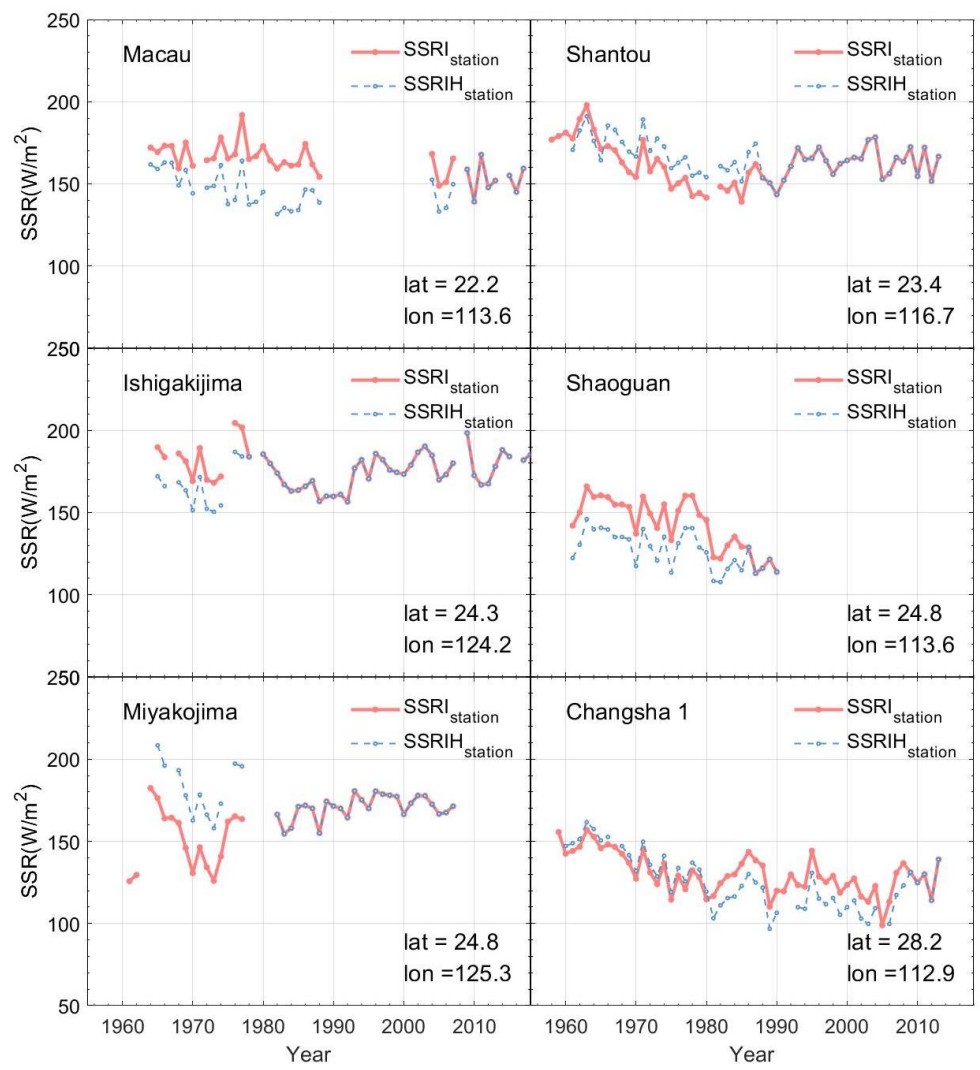

**Figure S1-1 Annual variation of SSR calculated from the original station SSR series (SSRI$_{station}$, blue line),**
**the station SSR series after homogenization (SSRIH$_{station}$, red line).**

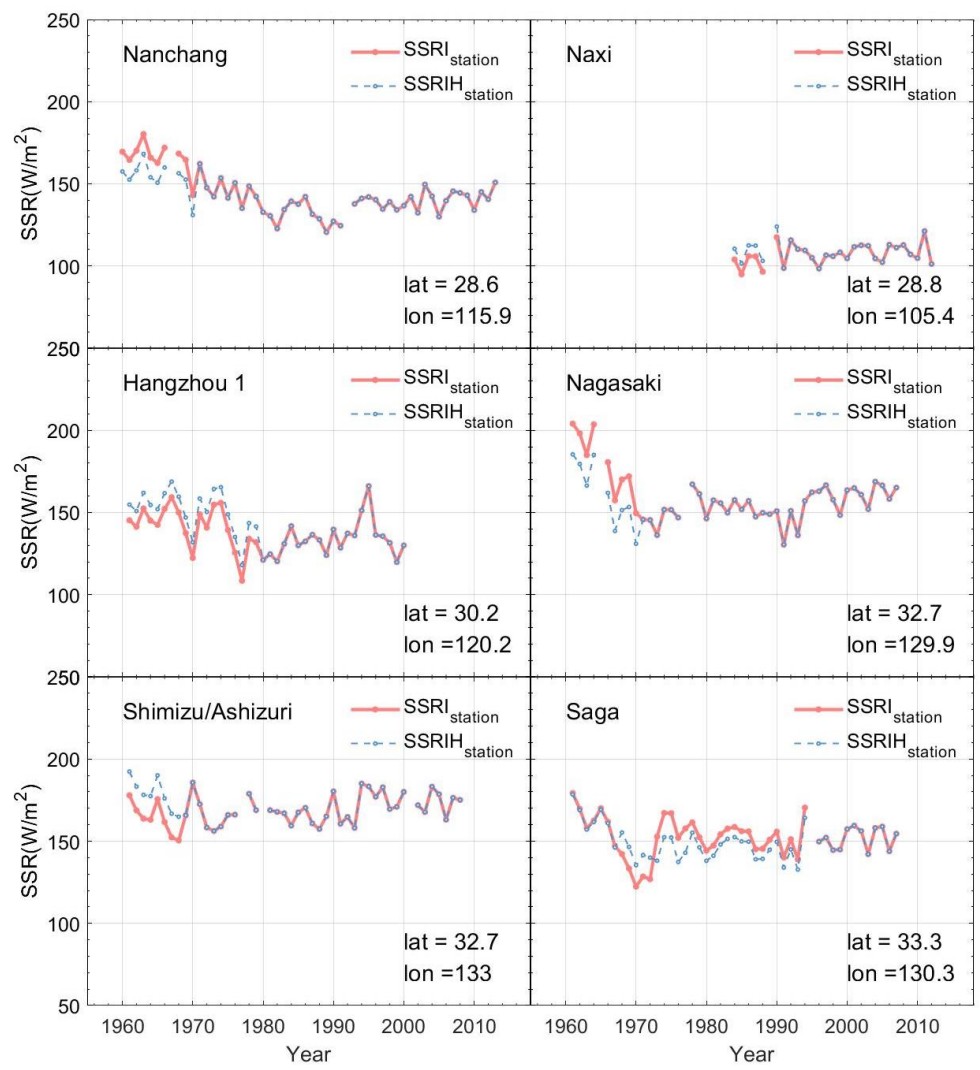

Figure S1-2 Annual variation of SSR calculated from the original station SSR series (SSRI_station, blue line),

the station SSR series after homogenization (SSRIH_station, red line).

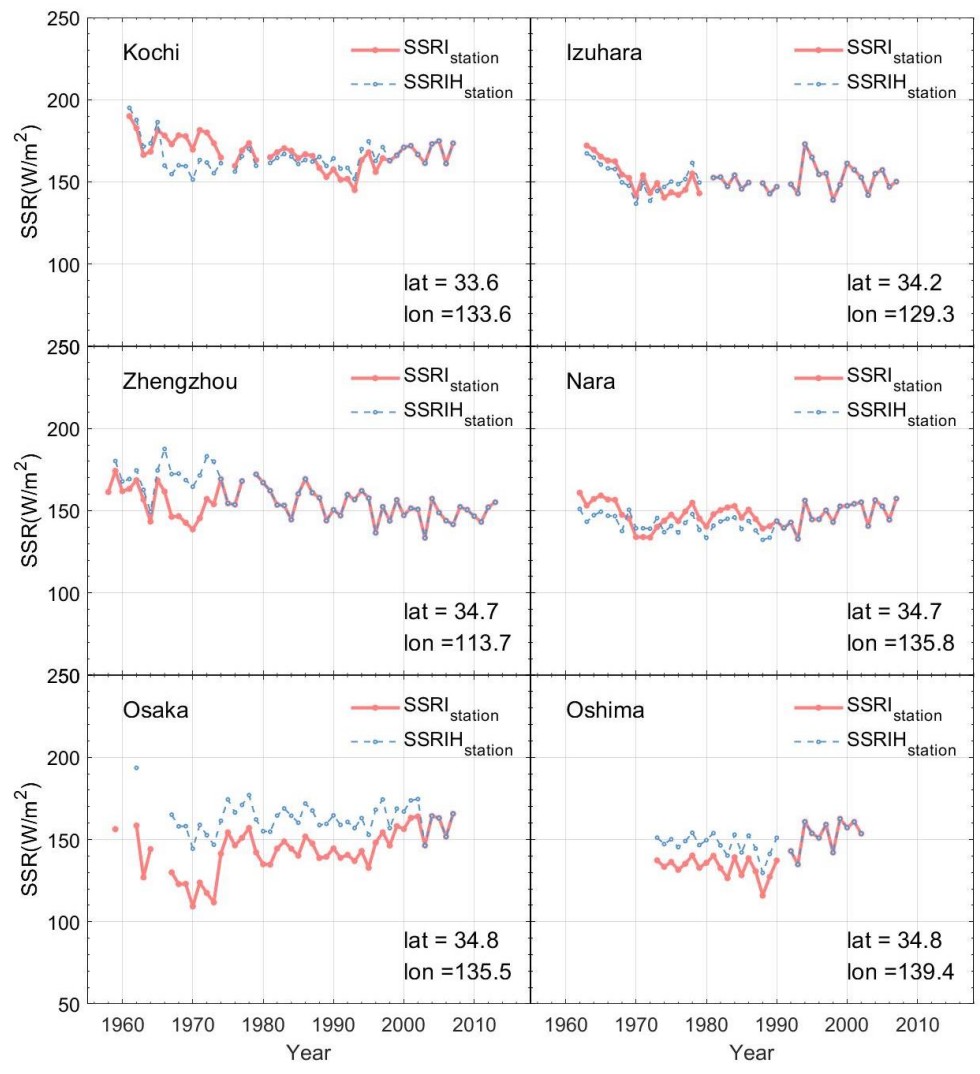

**Figure S1-3 Annual variation of SSR calculated from the original station SSR series (SSRI$_{station}$, blue line),**
**the station SSR series after homogenization (SSRIH$_{station}$, red line).**

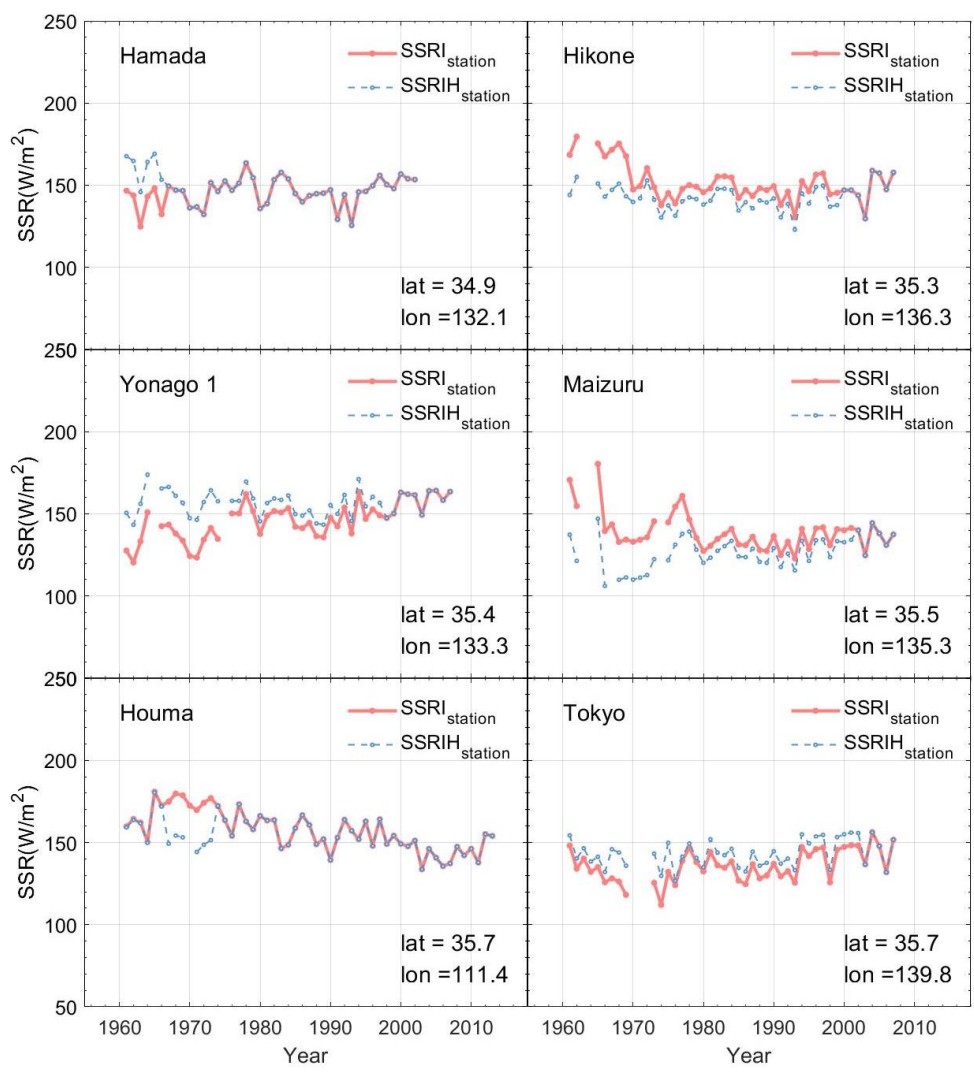

**Figure S1-4 Annual variation of SSR calculated from the original station SSR series (SSRI_station, blue line),**
**the station SSR series after homogenization (SSRIH_station, red line).**

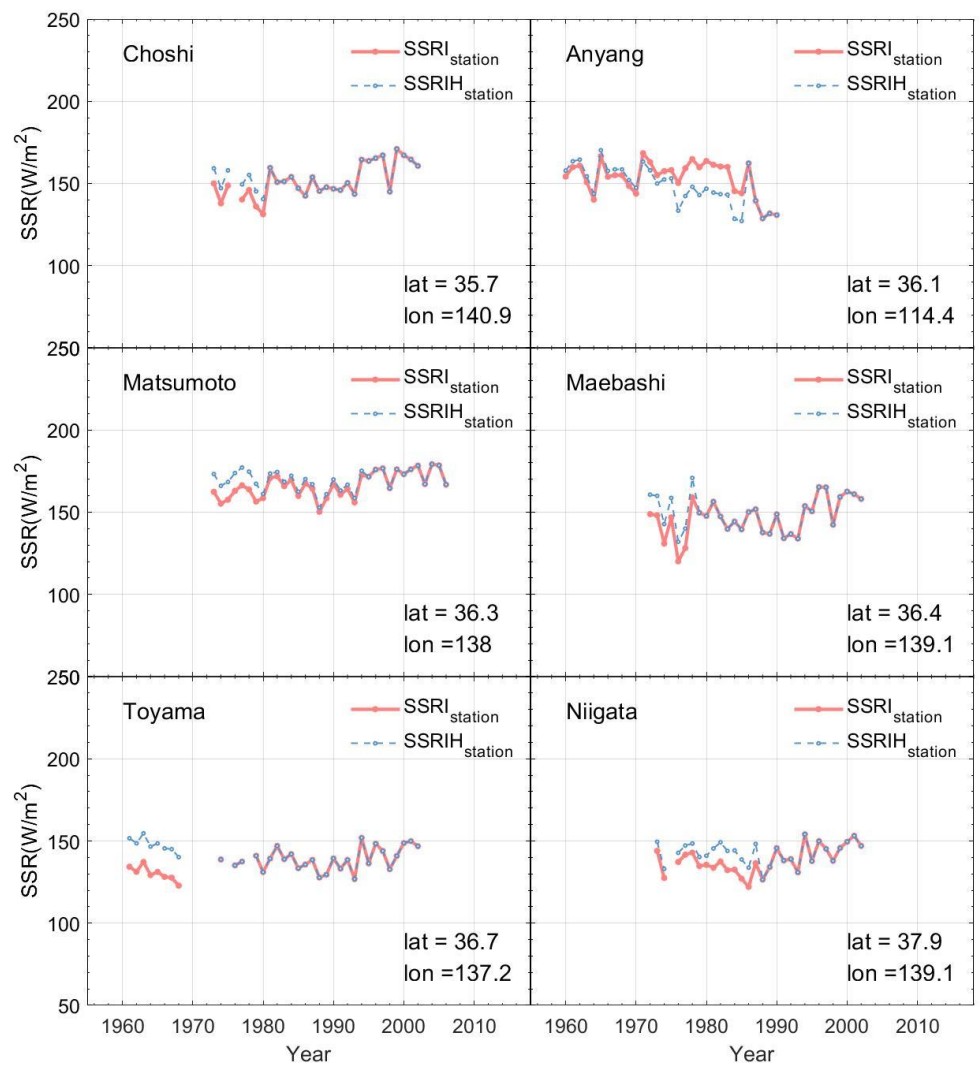

**Figure S1-5 Annual variation of SSR calculated from the original station SSR series (SSRI$_{station}$, blue line),**
**the station SSR series after homogenization (SSRIH$_{station}$, red line).**

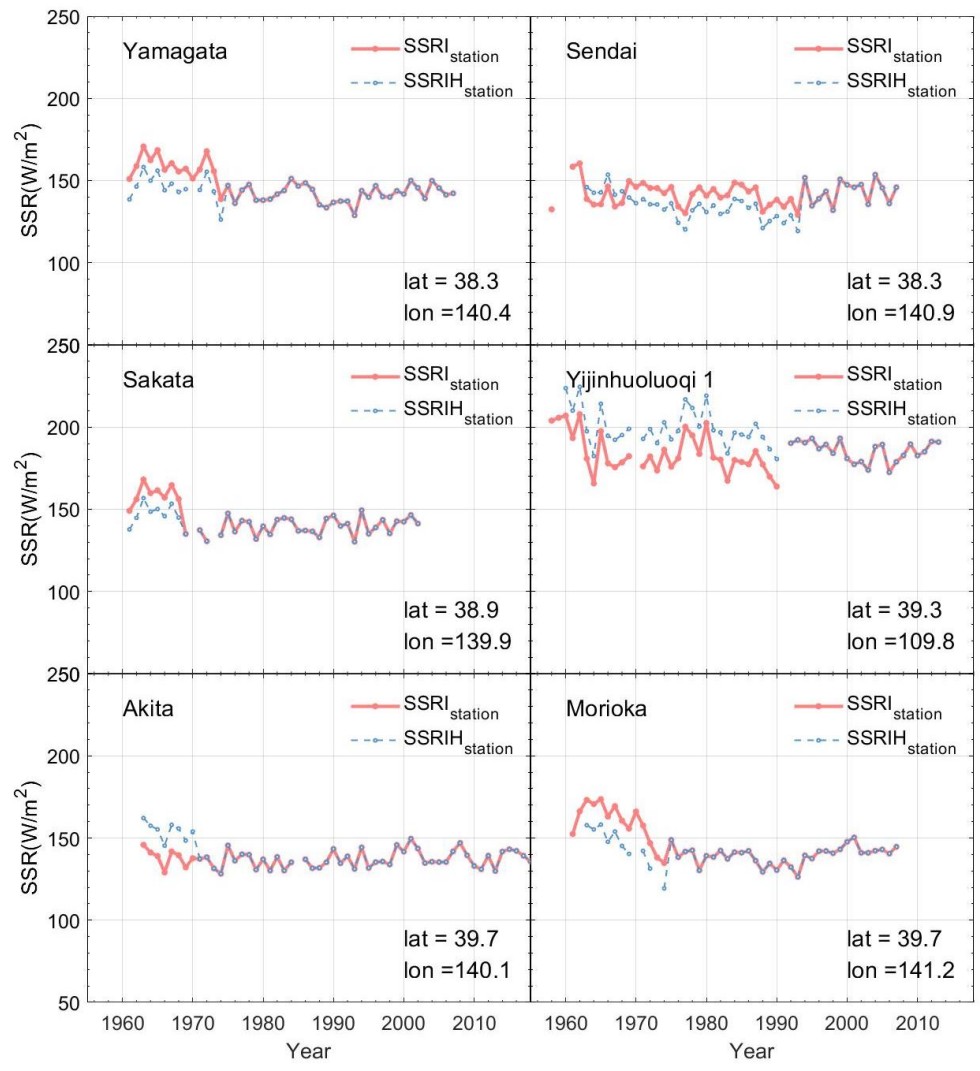

**Figure S1-6 Annual variation of SSR calculated from the original station SSR series (SSRI$_{station}$, blue line),**
**the station SSR series after homogenization (SSRIH$_{station}$, red line).**

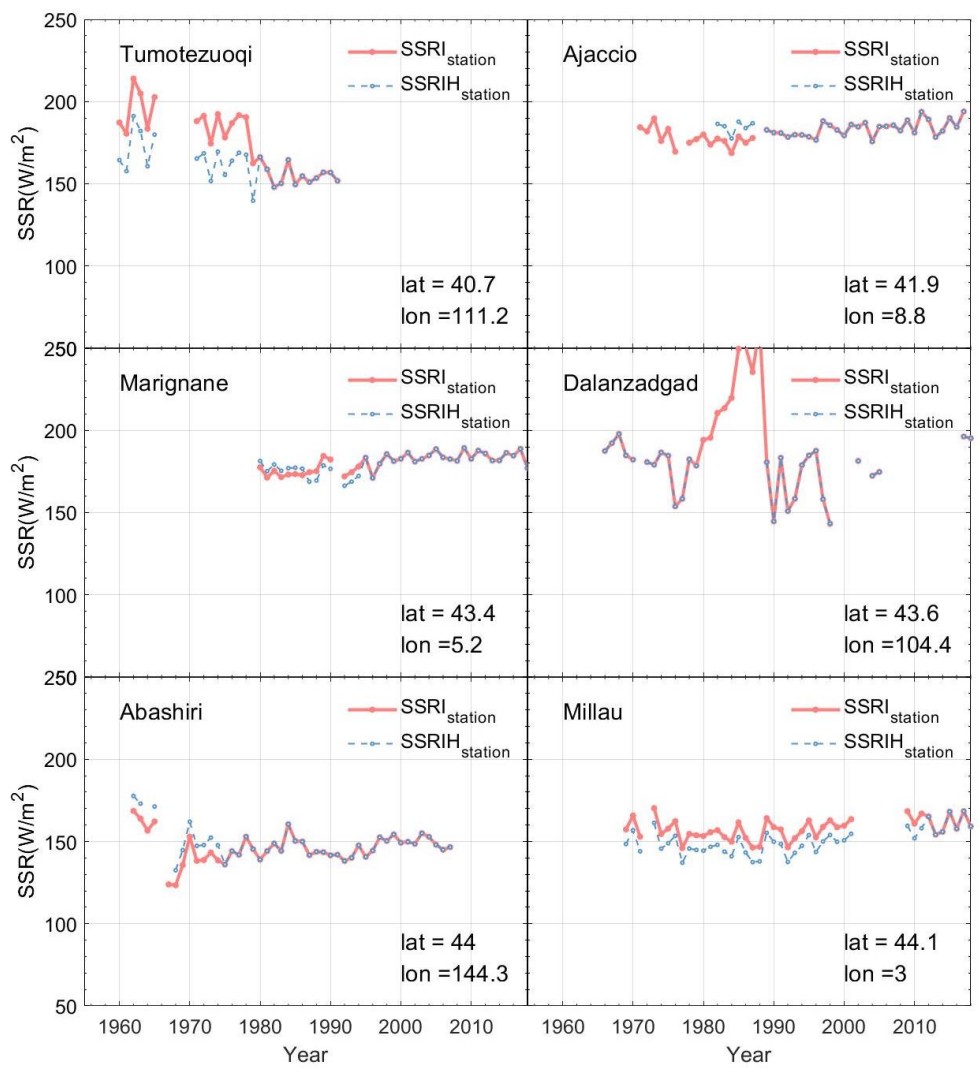

**Figure S1-7 Annual variation of SSR calculated from the original station SSR series (SSRI_station, blue line),**
**the station SSR series after homogenization (SSRIH_station, red line).**

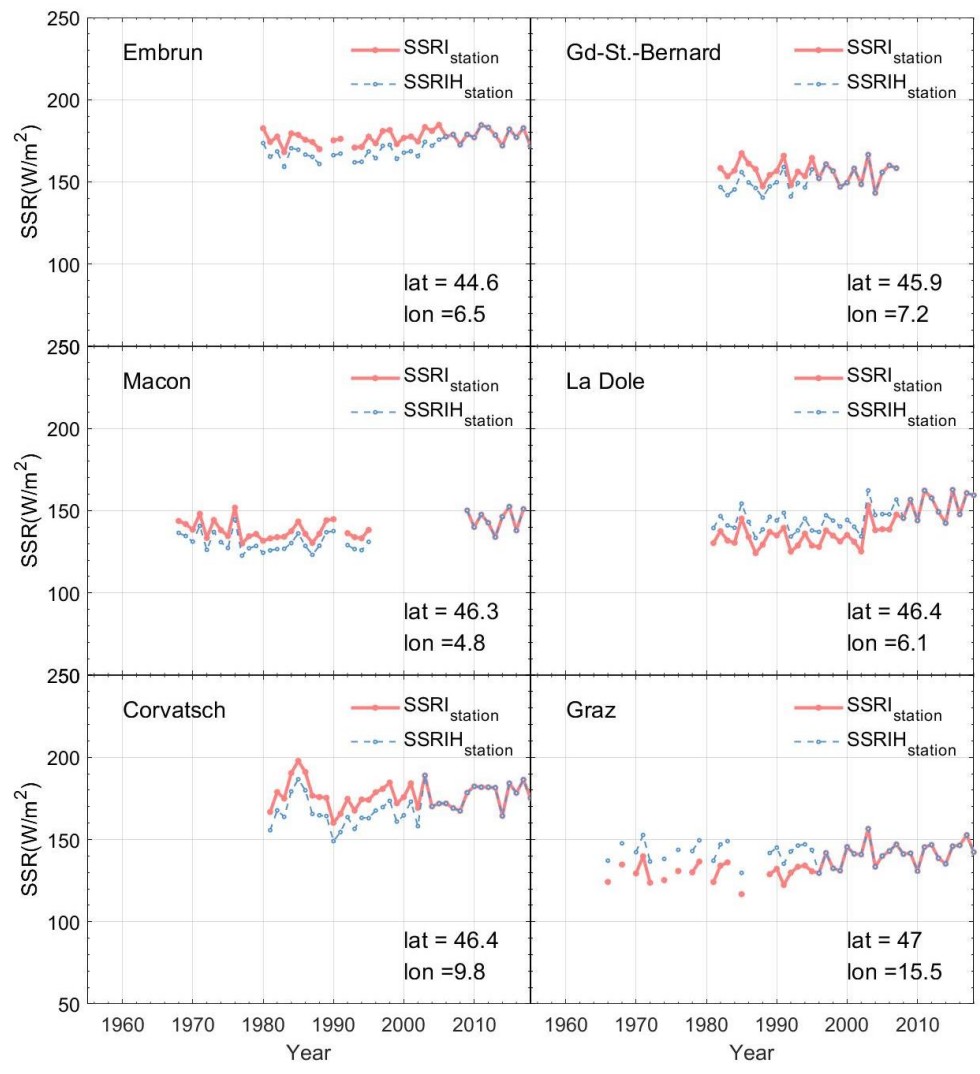

**Figure S1-8 Annual variation of SSR calculated from the original station SSR series (SSRI_station, blue line),**
**the station SSR series after homogenization (SSRIH_station, red line).**

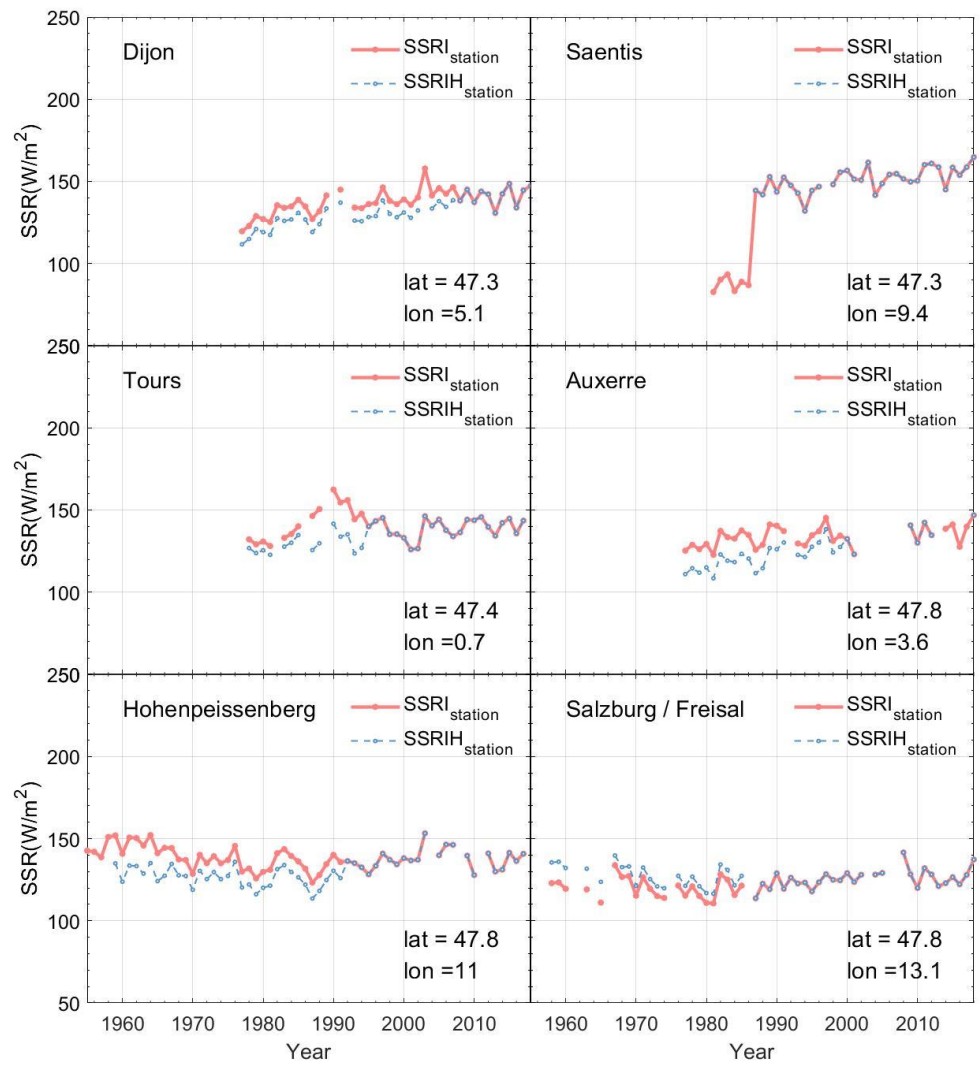


**Figure S1-9 Annual variation of SSR calculated from the original station SSR series (SSRI$_{station}$, blue line),**
**the station SSR series after homogenization (SSRIH$_{station}$, red line).**

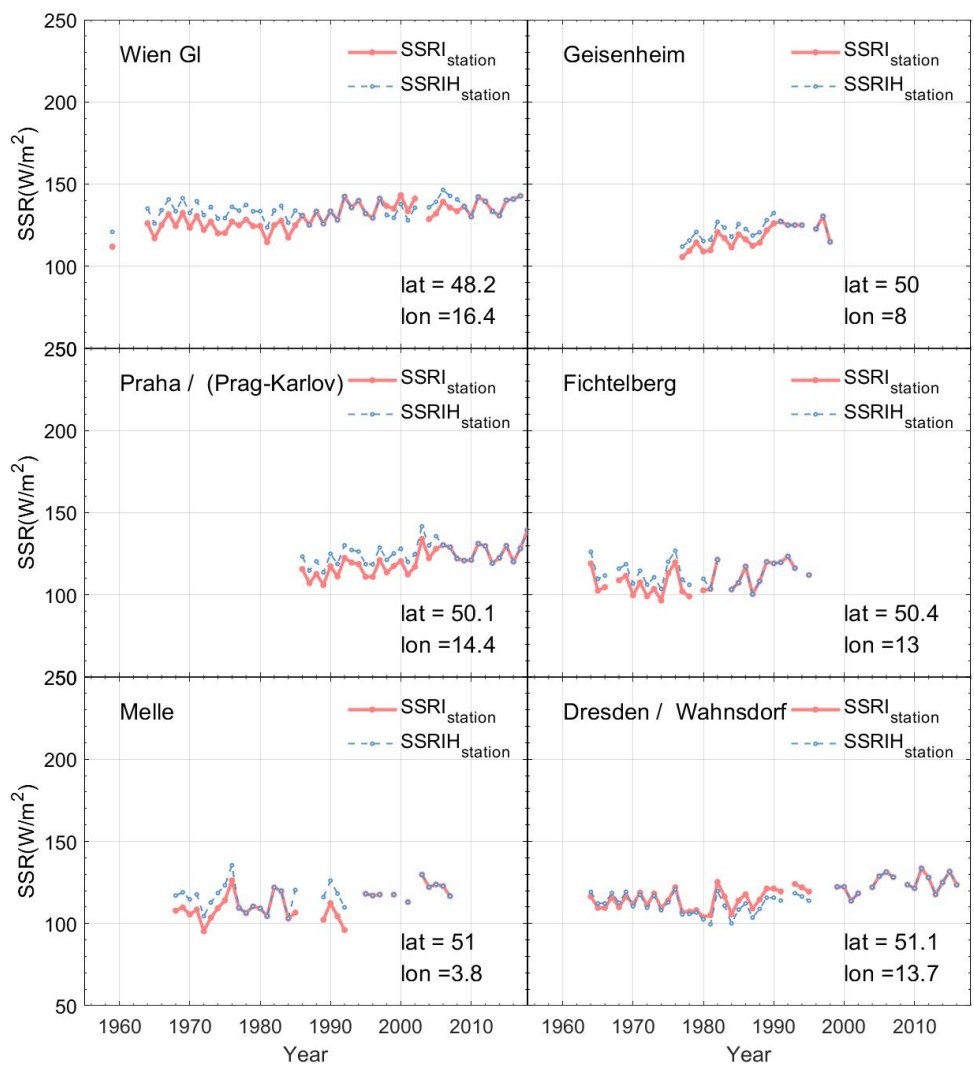

**Figure S1-10 Annual variation of SSR calculated from the original station SSR series (SSRI_station, blue line),**
**the station SSR series after homogenization (SSRIH_station, red line).**

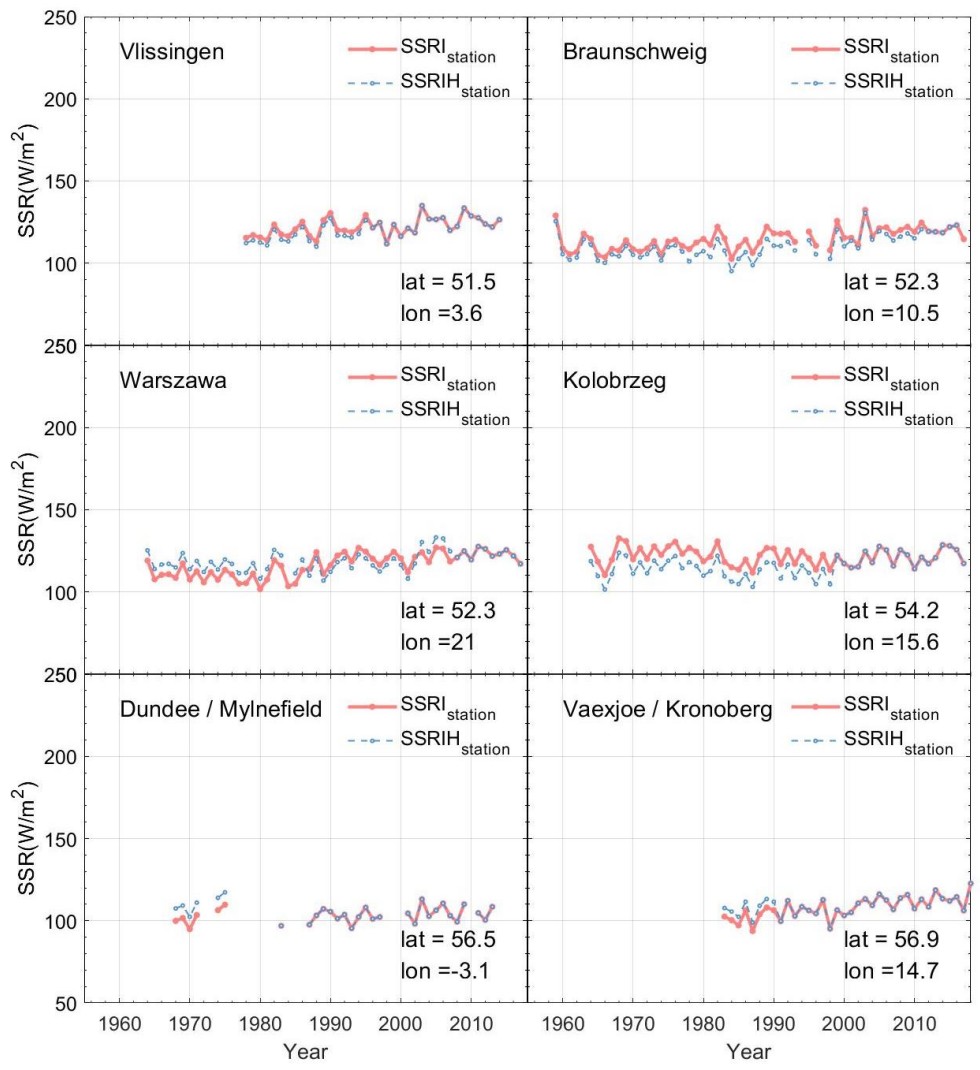

**Figure S1-11 Annual variation of SSR calculated from the original station SSR series (SSRI$_{station}$, blue line),**
**the station SSR series after homogenization (SSRIH$_{station}$, red line).**

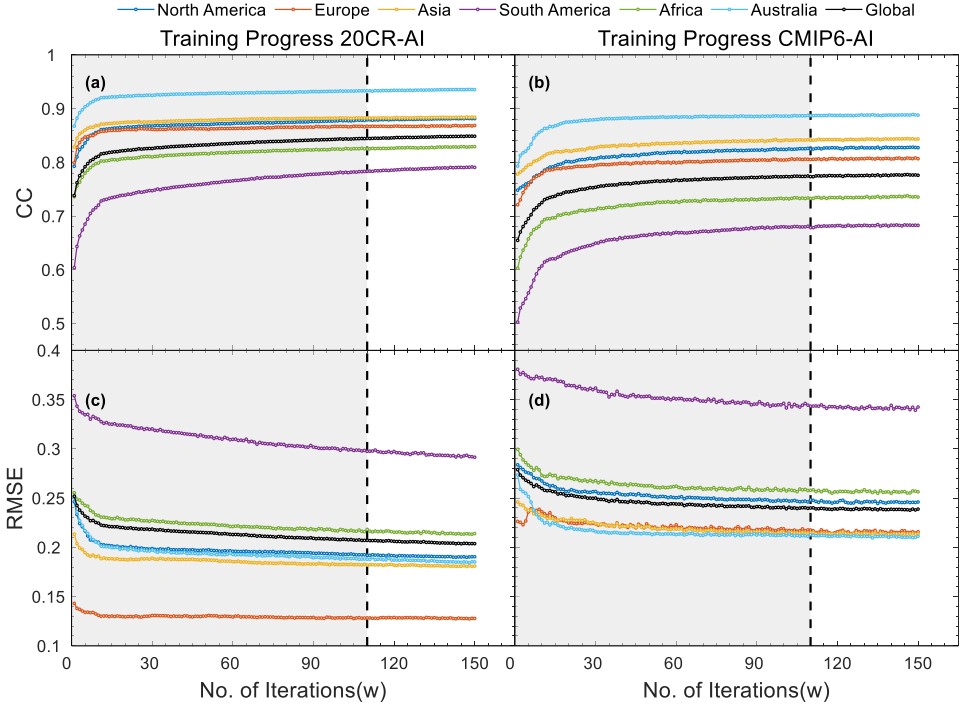

**Figure S2: 20CR-AI (CMIP6-AI) reconstruction model evaluation. Figure S3 (a/b) and (c/d) show the correlation coefficient (CC) and root mean squared error (RMSE) of the 20crAI/CMIP6AI model reconstruction results with the validation set for the different number of iterations.**

**20CR-AI**

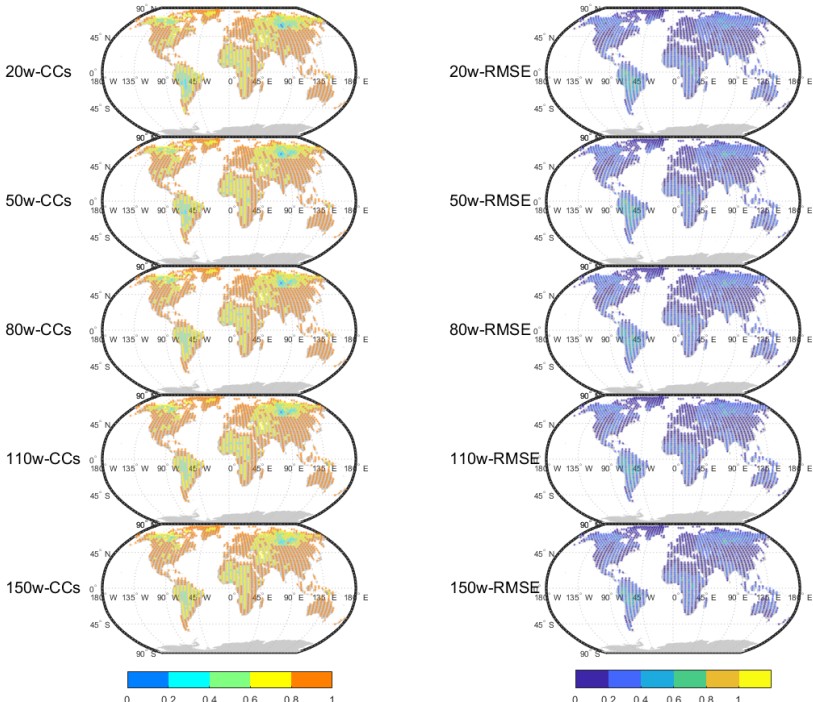

**Figure S3: 20CR-AI reconstruction model evaluation. The left and right panels show the spatial distribution**
**of the CC and the RMSE of the 20CR-AI model reconstruction results with the 20CR validation set for the**
**different number of iterations, respectively.**

**CMIP6-AI**

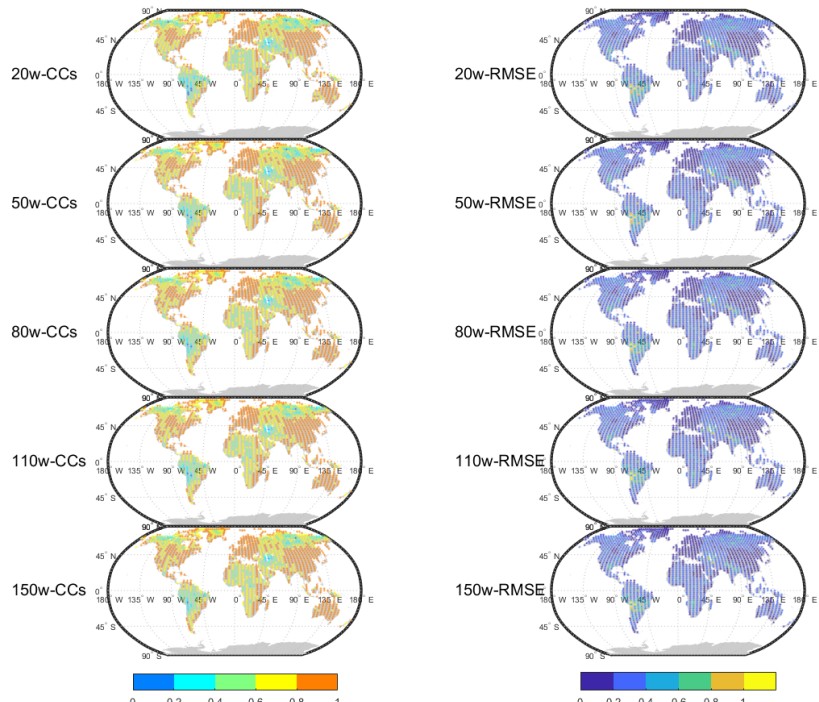

**Figure S4: same as Figure S3, but for CMIP6-AI.**

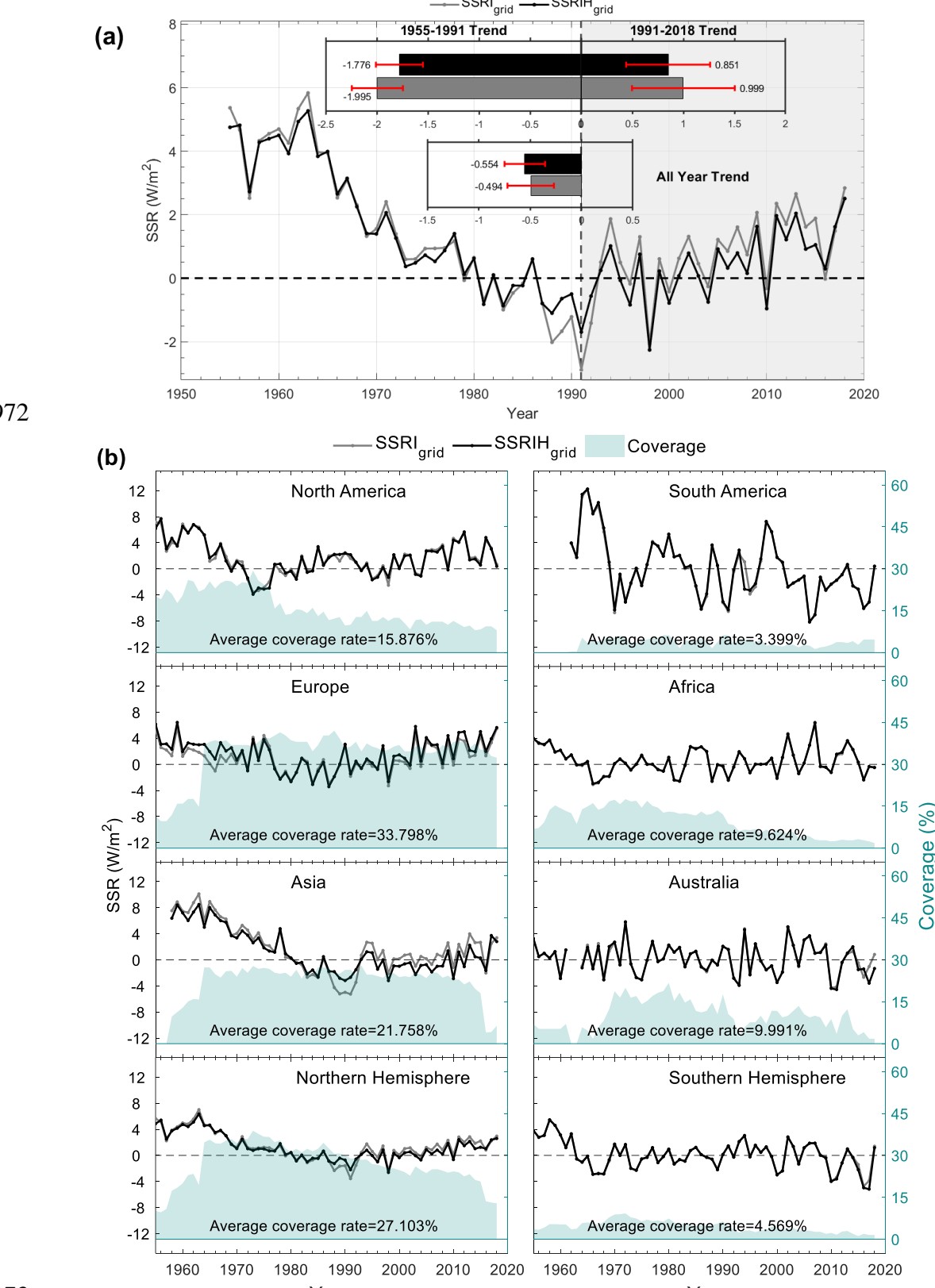



**Figure S5: Time series of the annual global (a) /regional (b) SSR anomaly variations (relative to 1971-2000)**

**before /after homogenization. The Grey /black solid line represents SSR before homogenization**
**(SSRI_grid)/SSRIH_grid annual anomalies. The histograms represent the decadal trends of the SSRI_grid /SSRIH_grid**
**(unit: W/m² per decade) and their 95% uncertainty range during three periods 1955-1988, 1988-2018 and**
**1955-2018.**

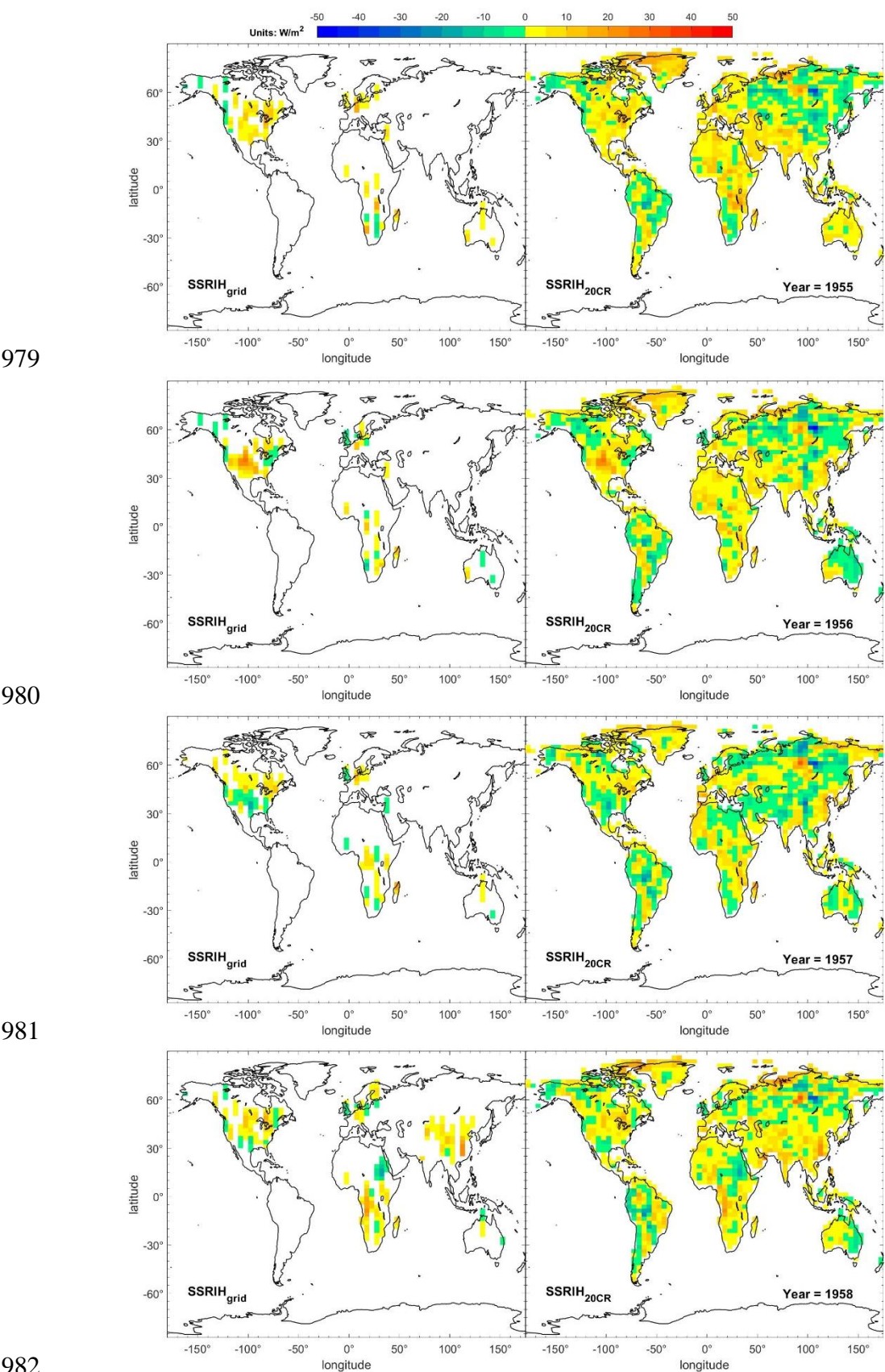





**Figure S6-1: Spatial distribution of SSRIH$_{grid}$ (column 1) and the SSR of reconstruction based on the 20CR-AI model (SSRIH$_{20CR}$ (column 2)) in typical years (1955-1958).**


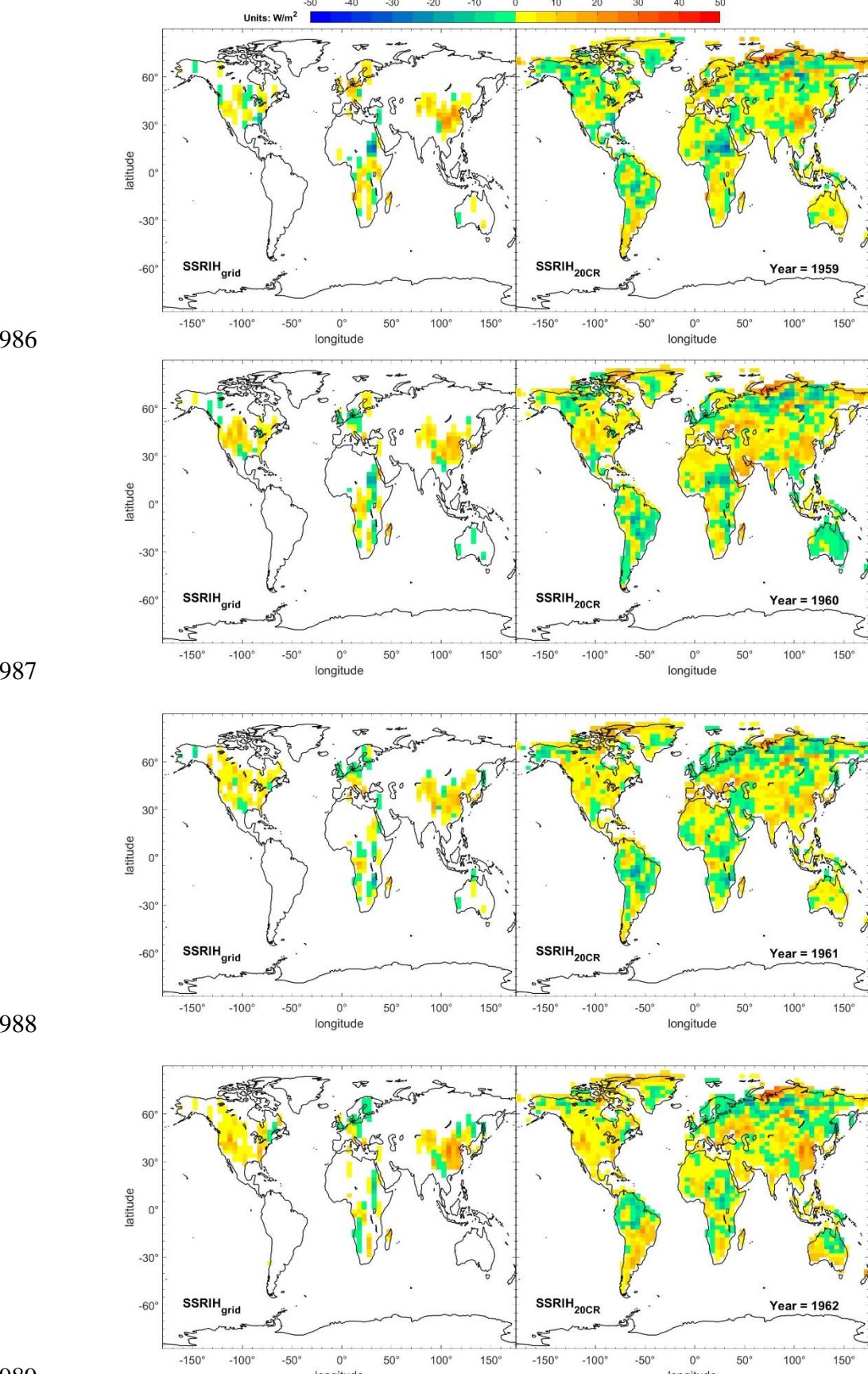





**Figure S6-2: Spatial distribution of SSRIH$_{grid}$ (column 1) and SSRIH$_{20CR}$ (column 2) in typical years (1959-1962).**



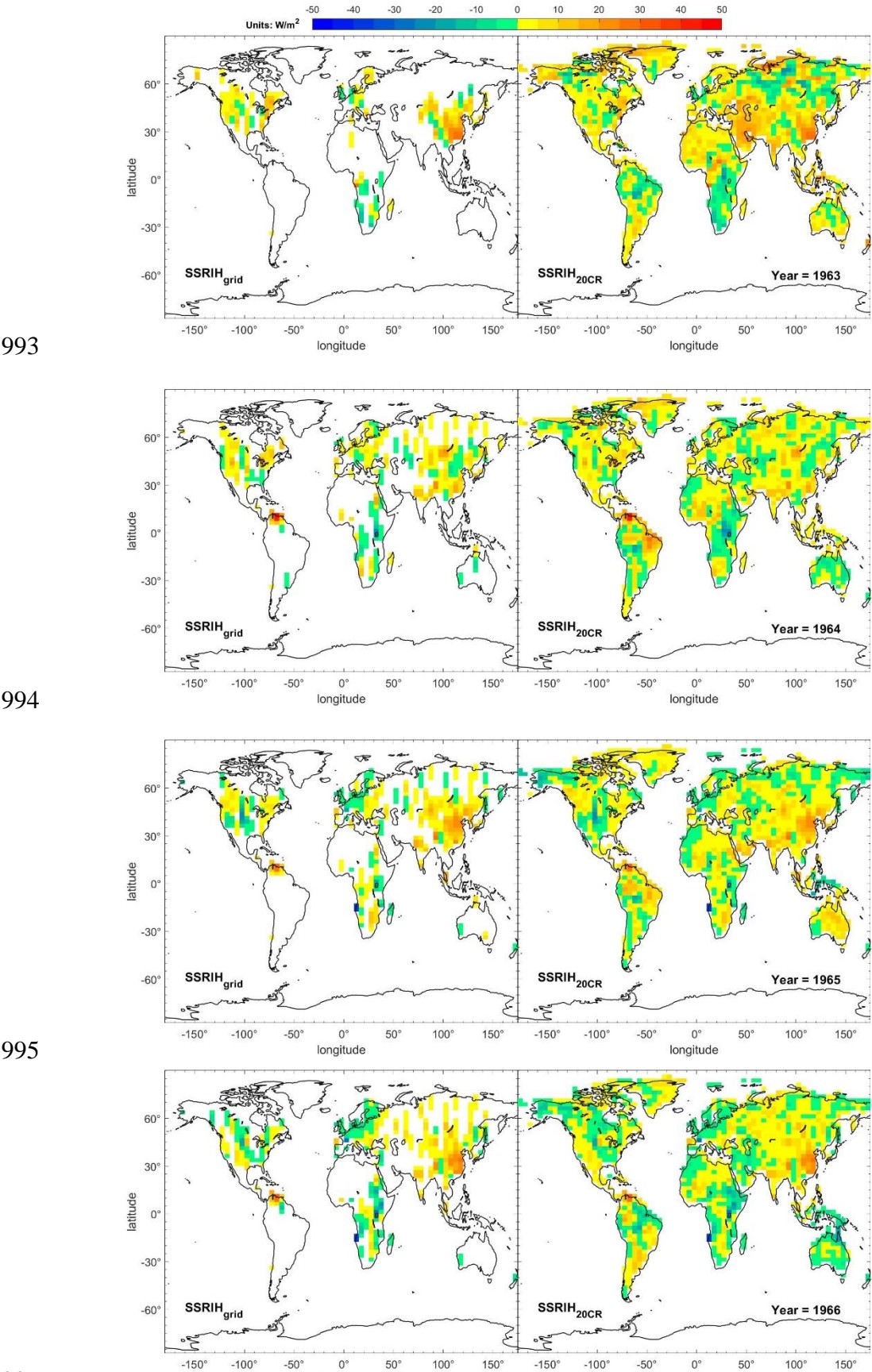




**Figure S6-3: Spatial distribution of SSRIH$_{grid}$ (column 1) and SSRIH$_{20CR}$ (column 2) in typical years (1963-**
**1966).**





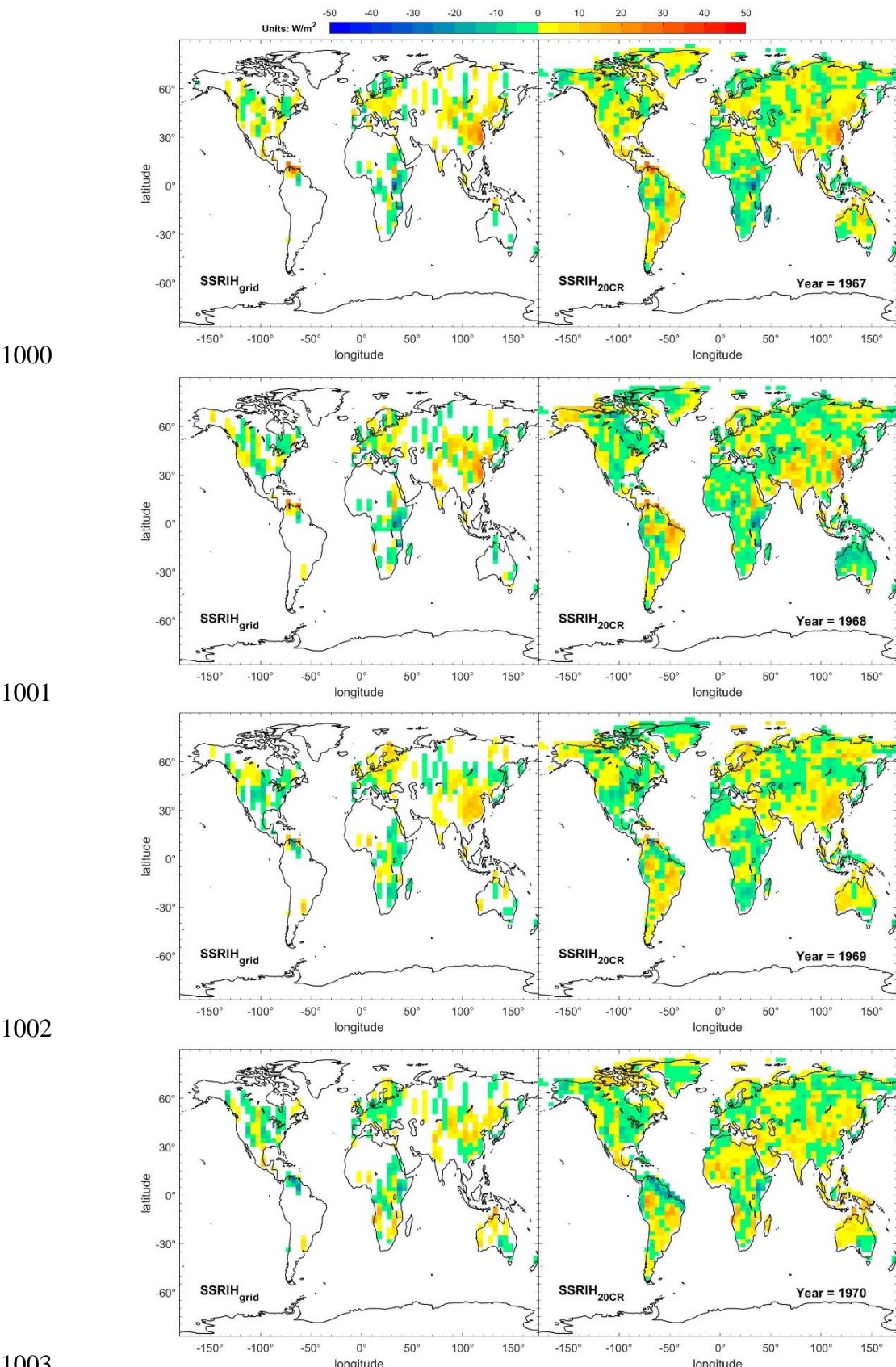

**Figure S6-4: Spatial distribution of SSRIH$_{grid}$ (column 1) and SSRIH$_{20CR}$ (column 2) in typical years (1967-1970).**





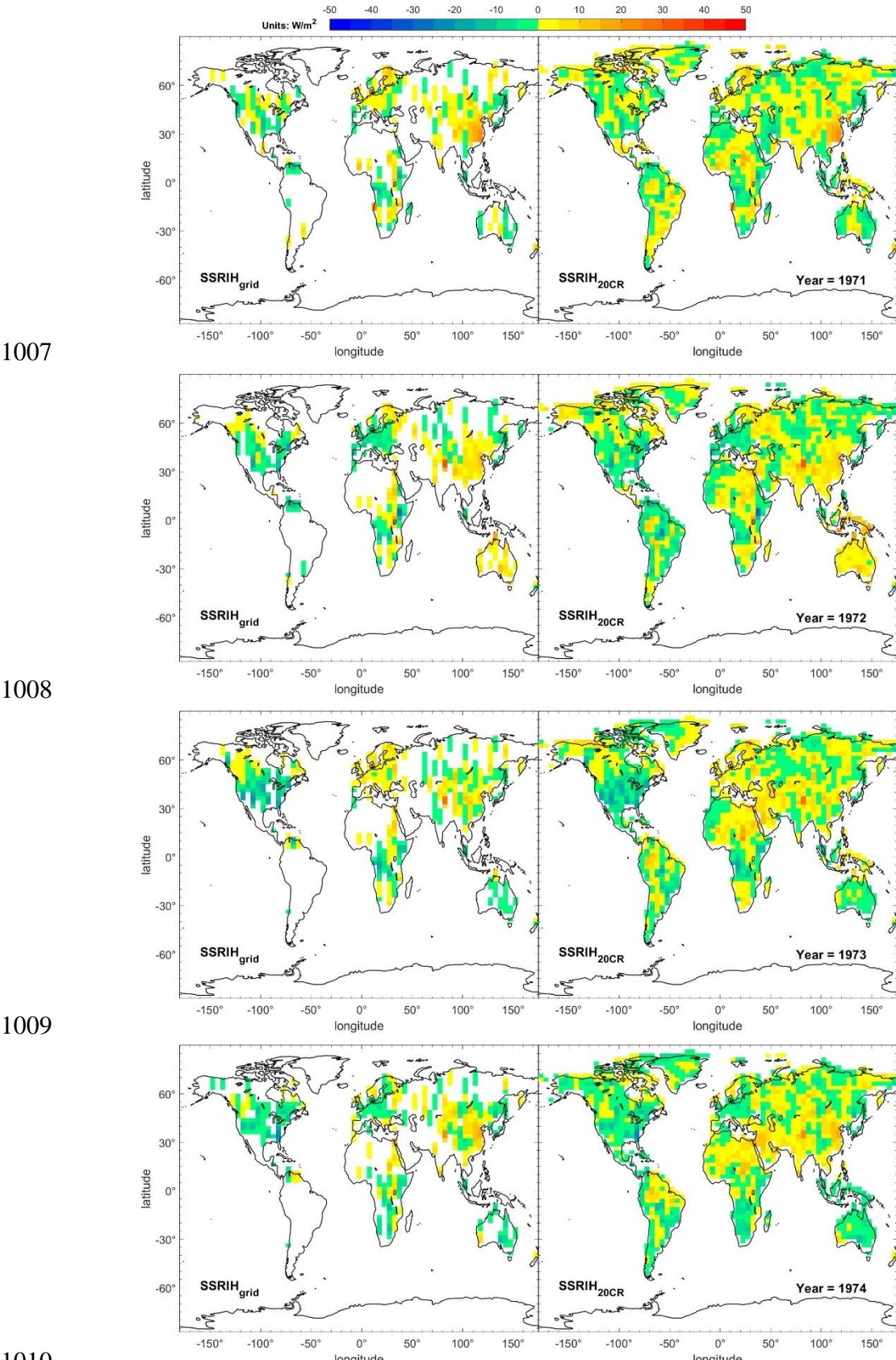




**Figure S6-5: Spatial distribution of SSRIH$_{grid}$ (column 1) and SSRIH$_{20CR}$ (column 2) in typical years (1971-**
**1974).**


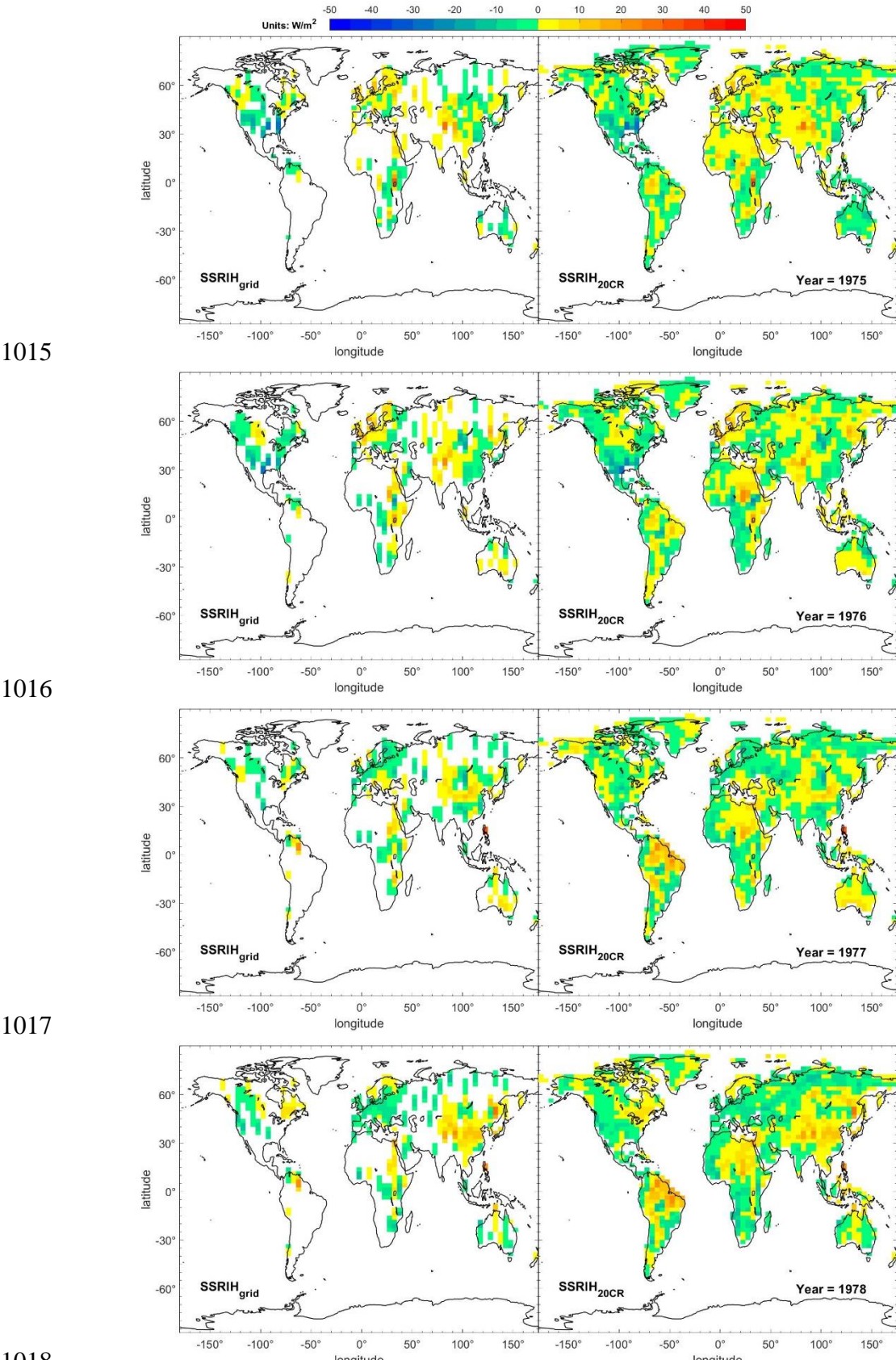



Figure S6-6: Spatial distribution of SSRIHgrid (column 1) and SSRIH20CR (column 2) in typical years (1975-
1020    1978).



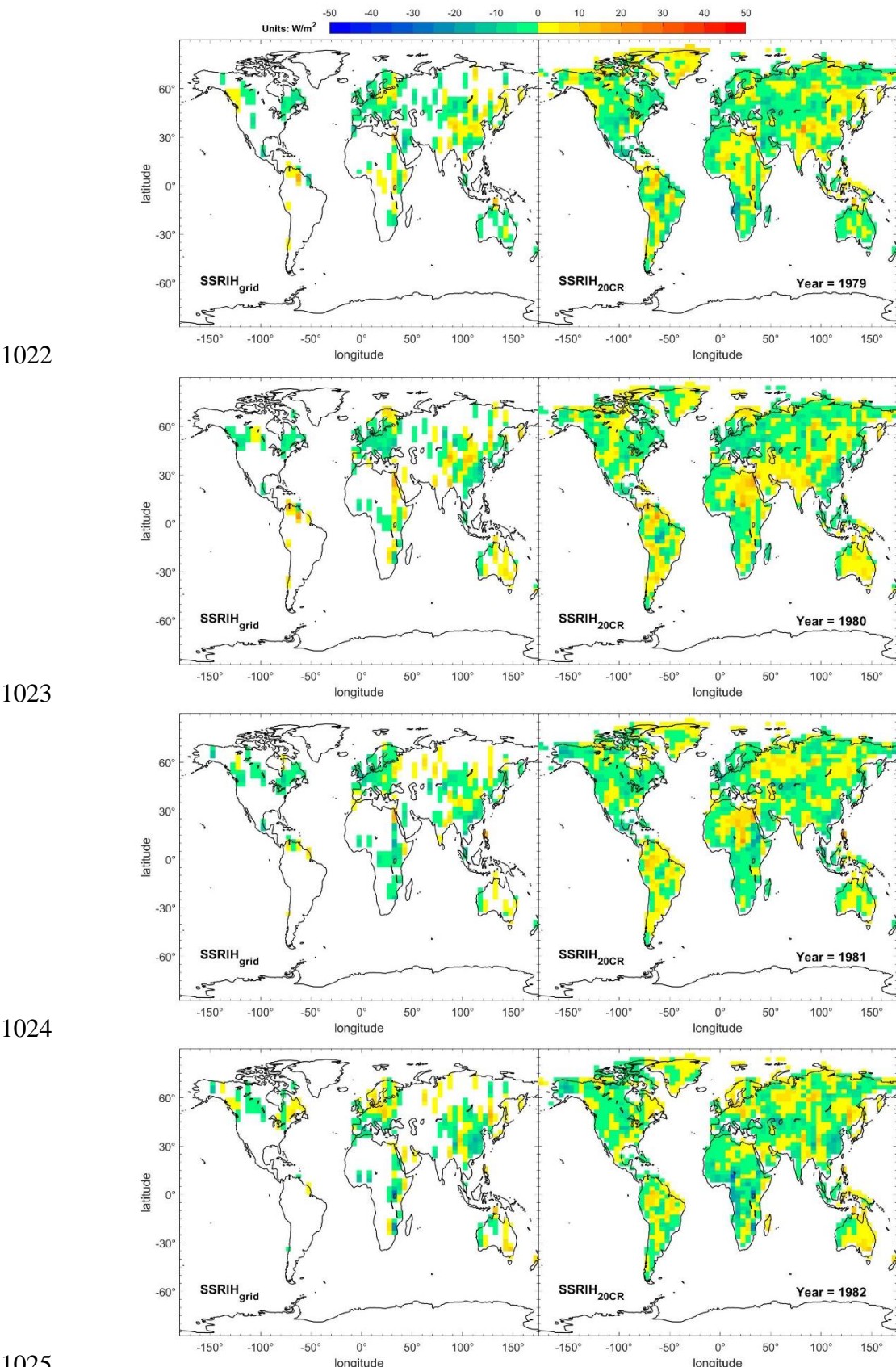



**Figure S6-7: Spatial distribution of SSRIH$_{grid}$ (column 1) and SSRIH$_{20CR}$ (column 2) in typical years (1979-**
**1982).**






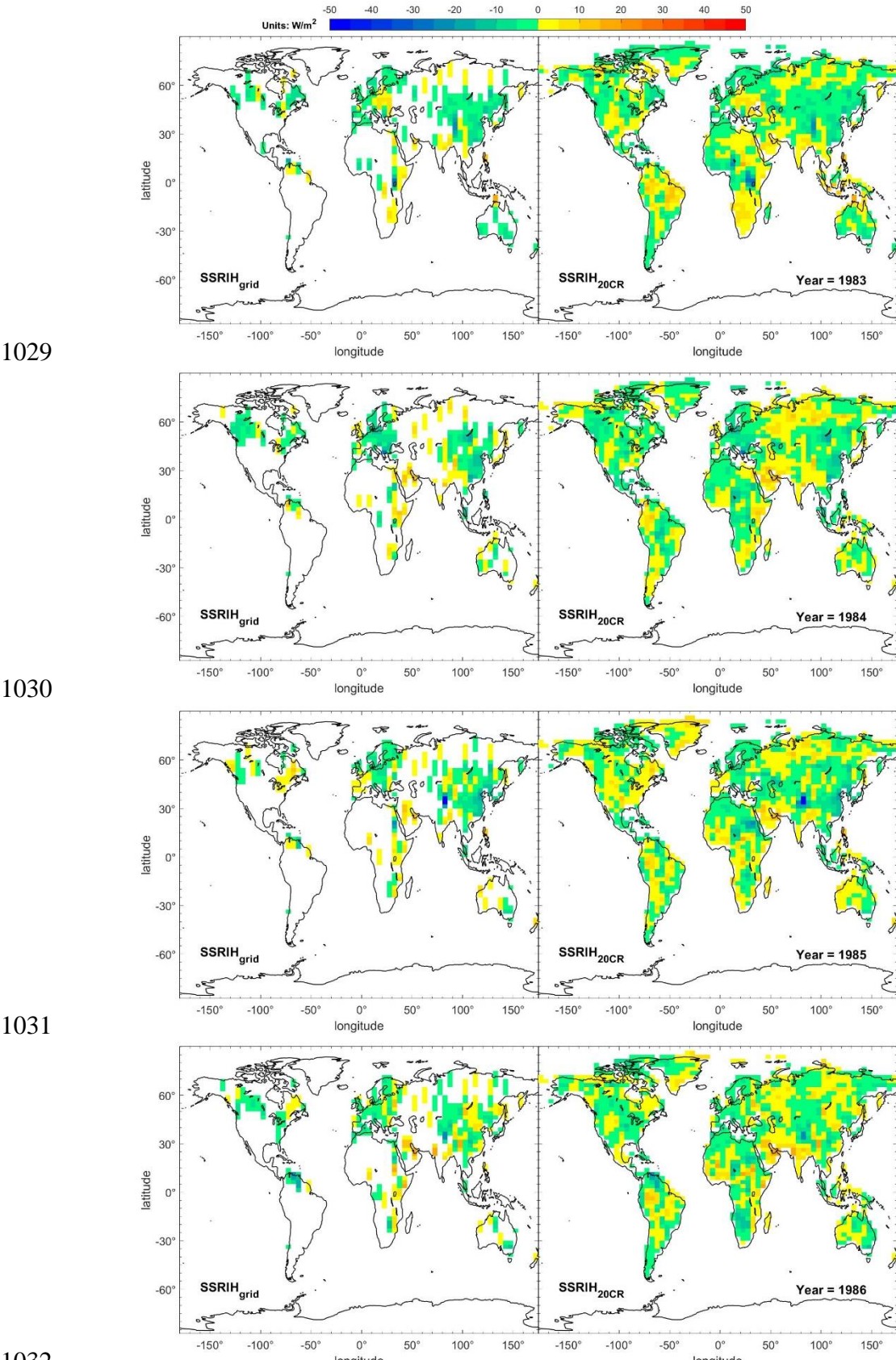

**Figure S6-8: Spatial distribution of SSRIH$_{grid}$ (column 1) and SSRIH$_{20CR}$ (column 2) in typical years (1983-1986).**



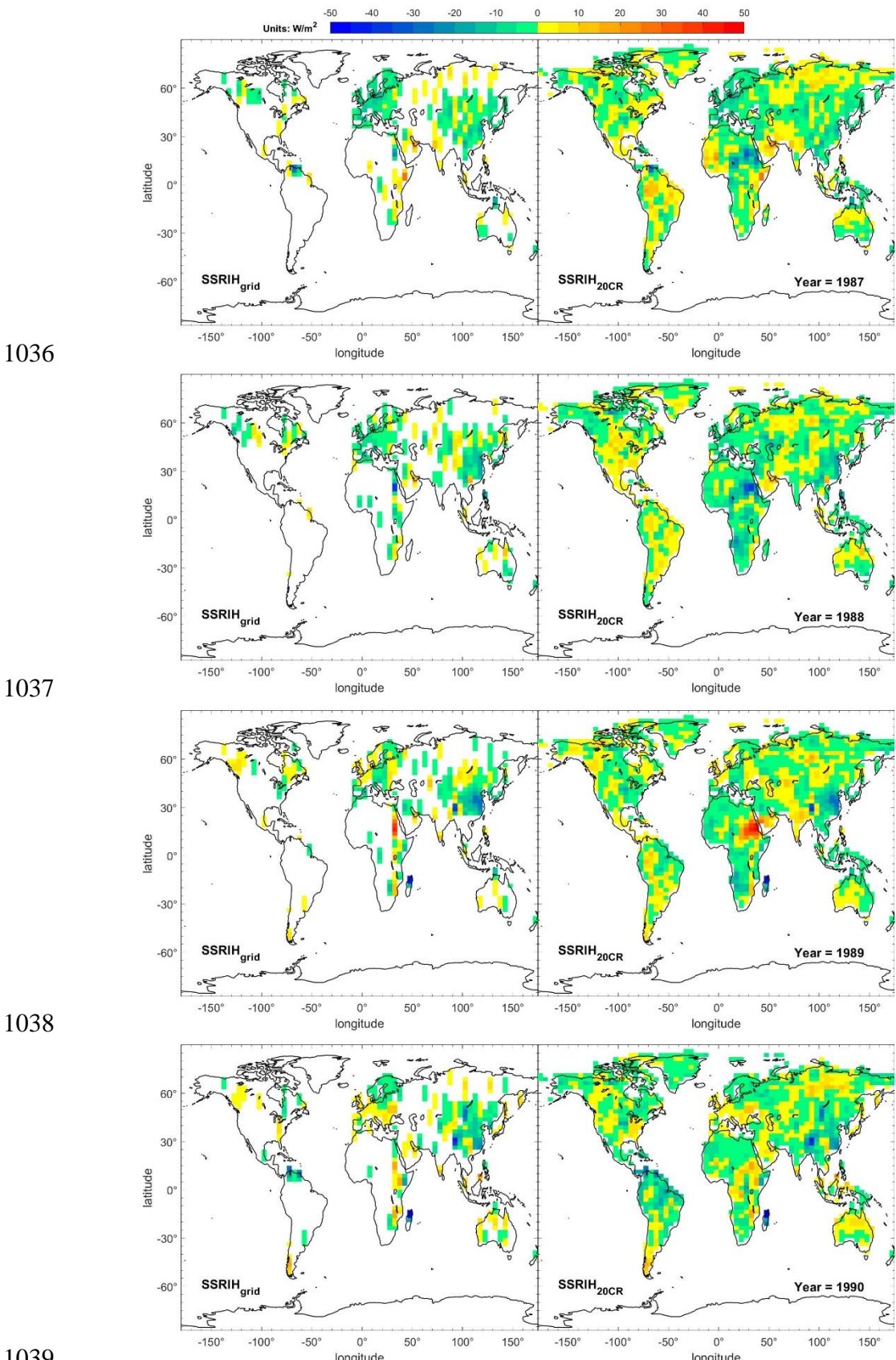




**Figure S6-9: Spatial distribution of SSRIH$_{grid}$ (column 1) and SSRIH$_{20CR}$ (column 2) in typical years (1987-**
**1990).**

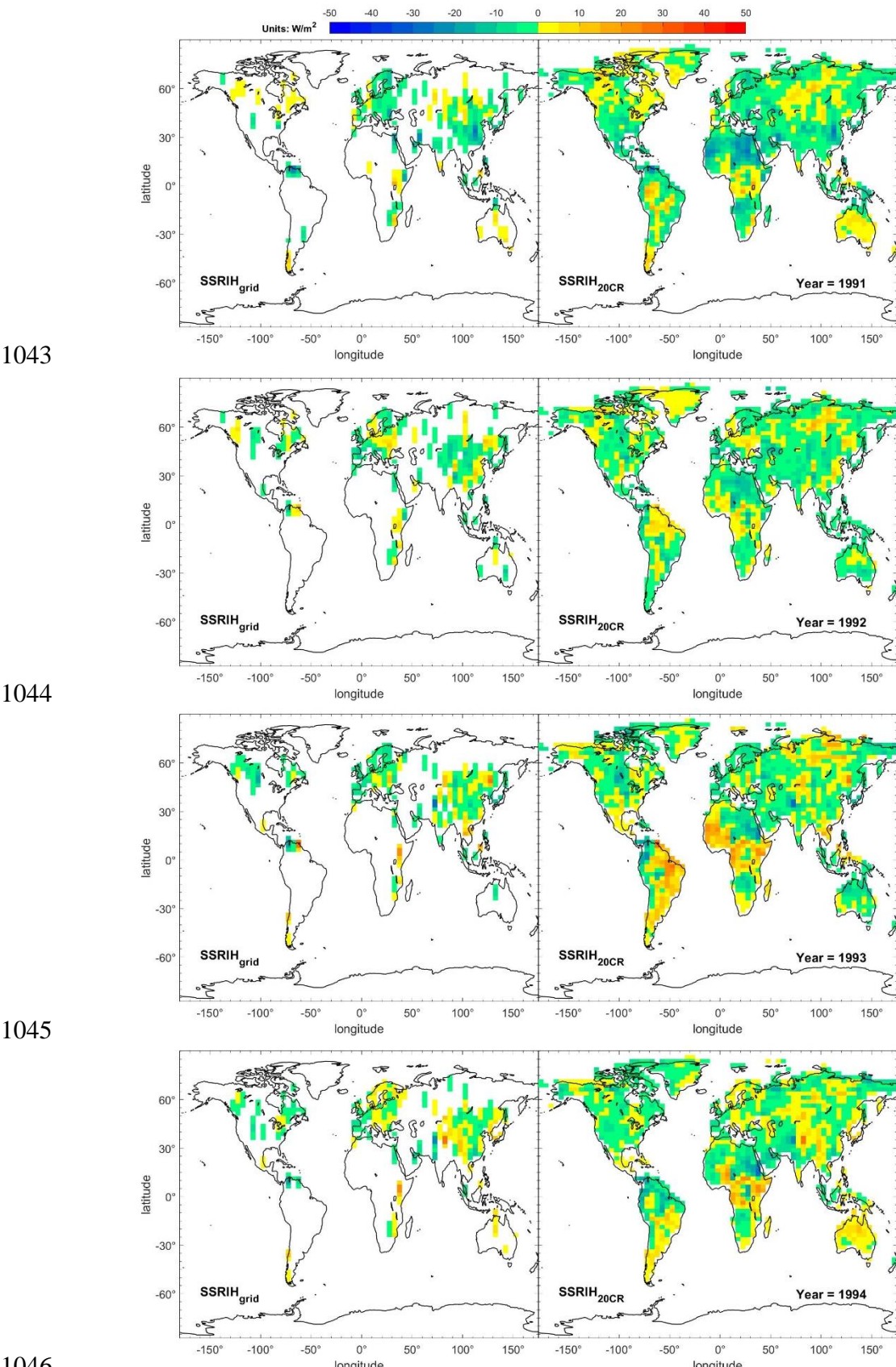





**Figure S6-10: Spatial distribution of SSRIH$_{grid}$ (column 1) and SSRIH$_{20CR}$ (column 2) in typical years (1991-**
**1994).**

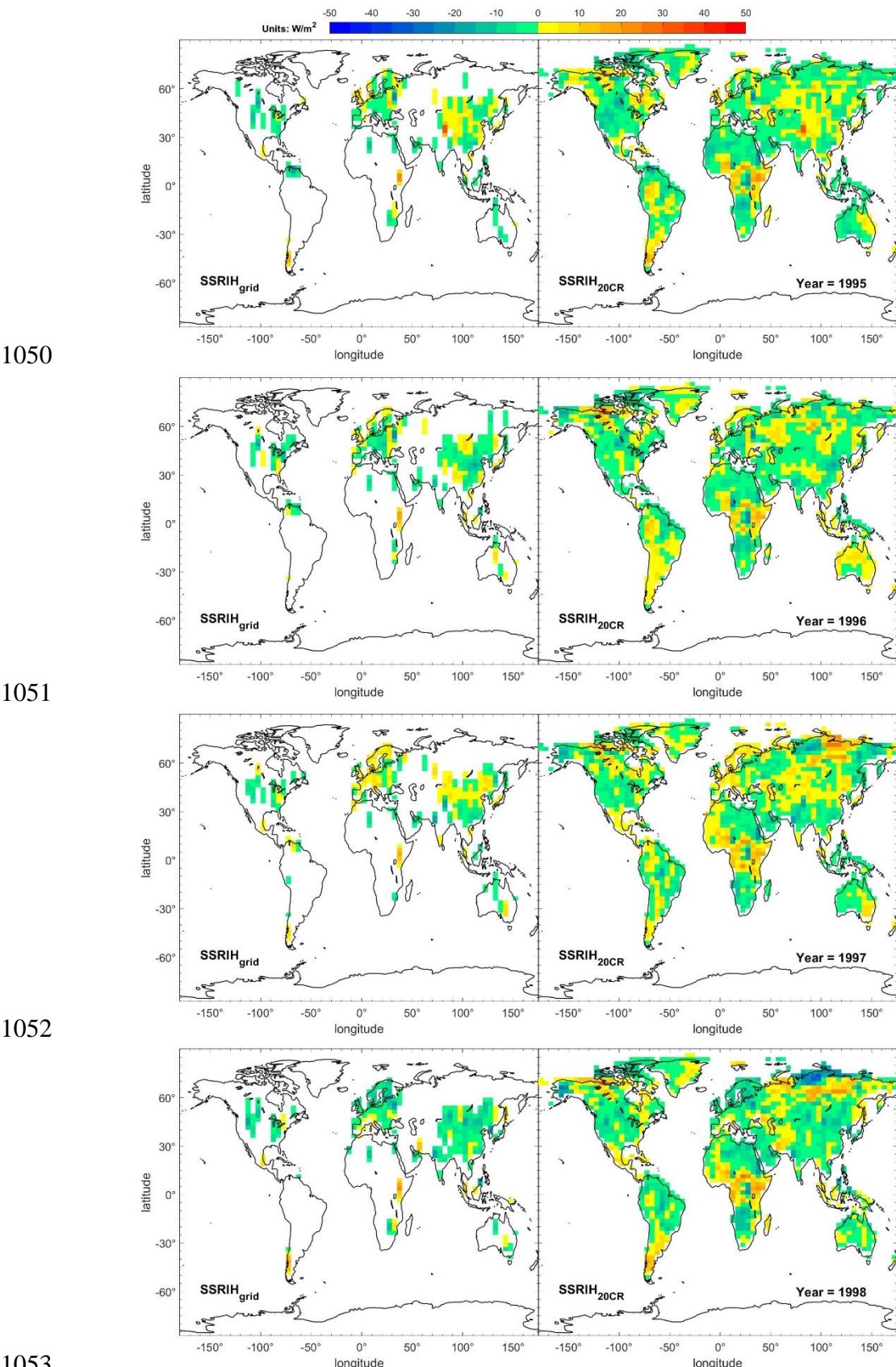





**Figure S6-11: Spatial distribution of SSRIH$_{grid}$ (column 1) and SSRIH$_{20CR}$ (column 2) in typical years (1995-1998).**



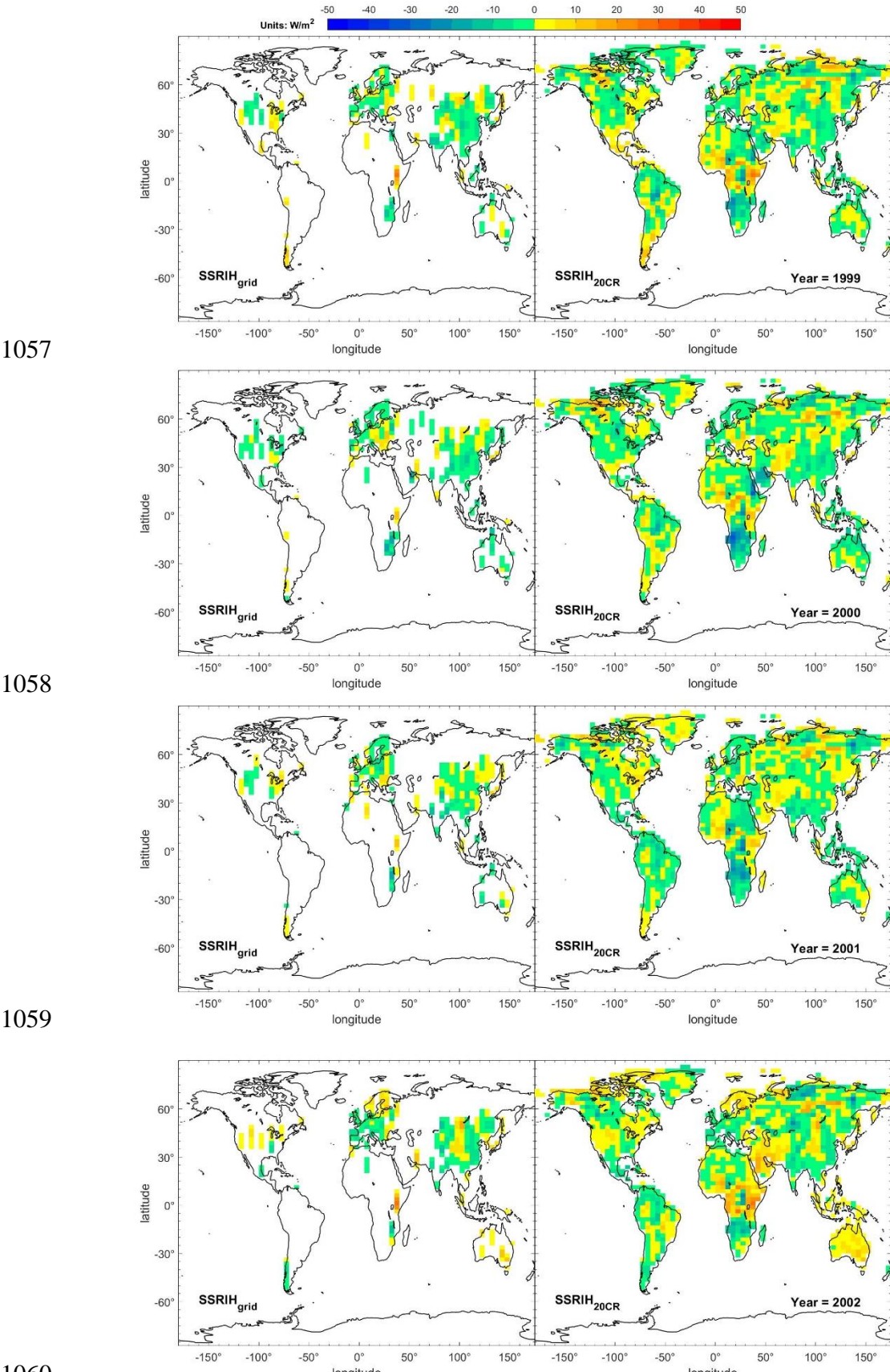




**Figure S6-12: Spatial distribution of SSRIH$_{grid}$ (column 1) and SSRIH$_{20CR}$ (column 2) in typical years (1999-**
**2002).**

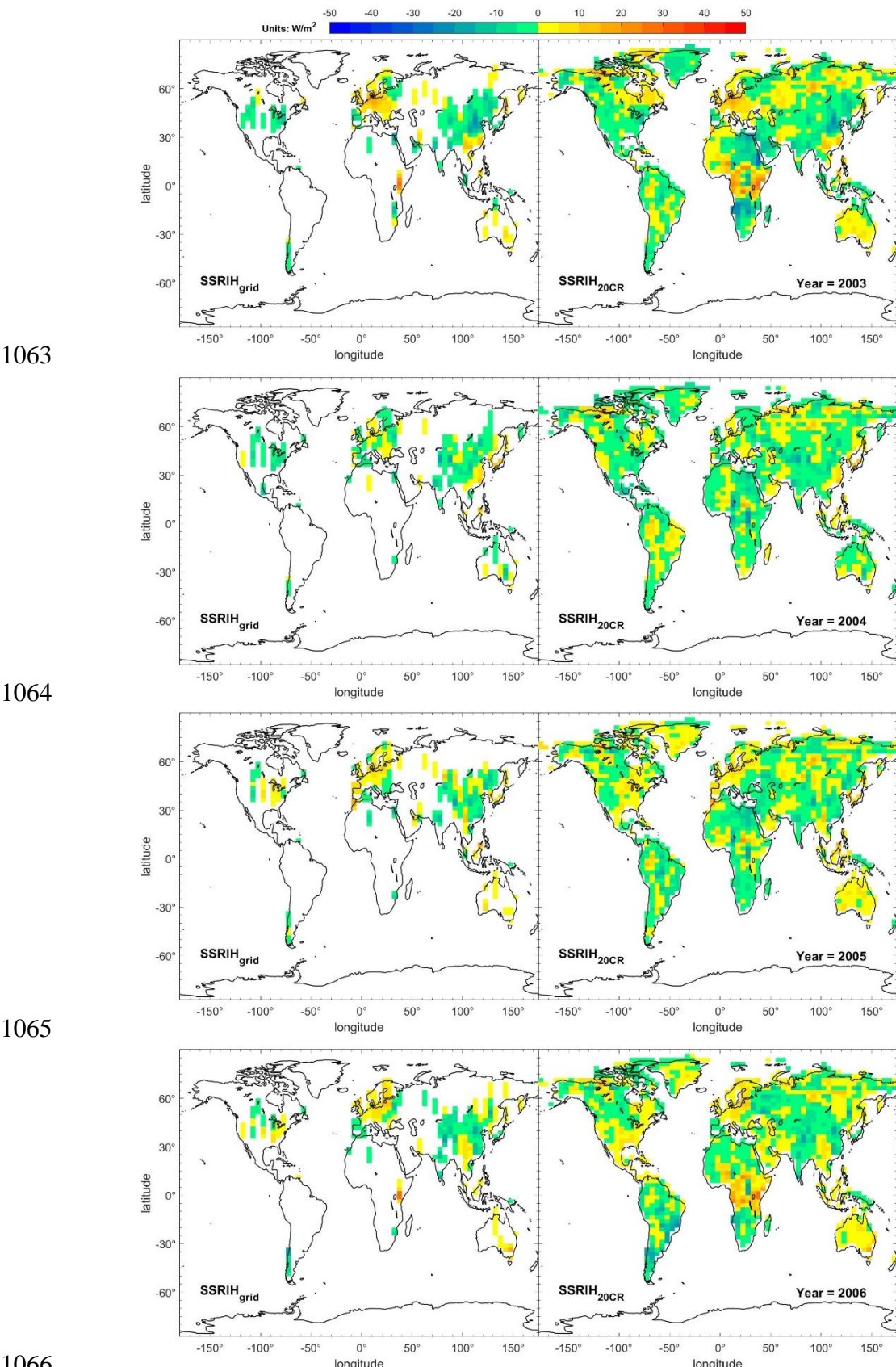





**Figure S6-13: Spatial distribution of SSRIH$_{grid}$ (column 1) and SSRIH$_{20CR}$ (column 2) in typical years (2003-2006).**



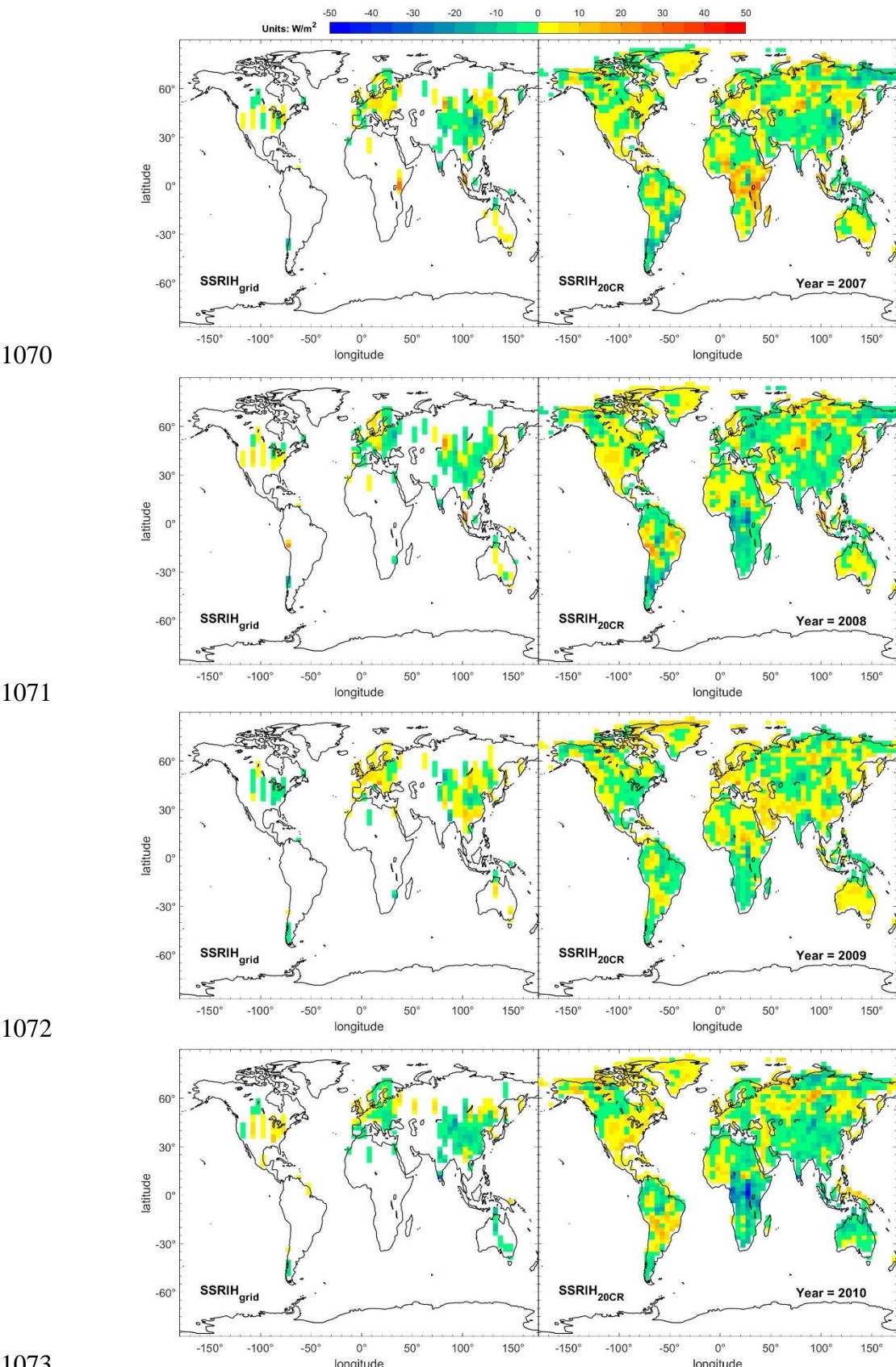




**Figure S6-14: Spatial distribution of SSRIH$_{grid}$ (column 1) and SSRIH$_{20CR}$ (column 2) in typical years (2007-2010).**



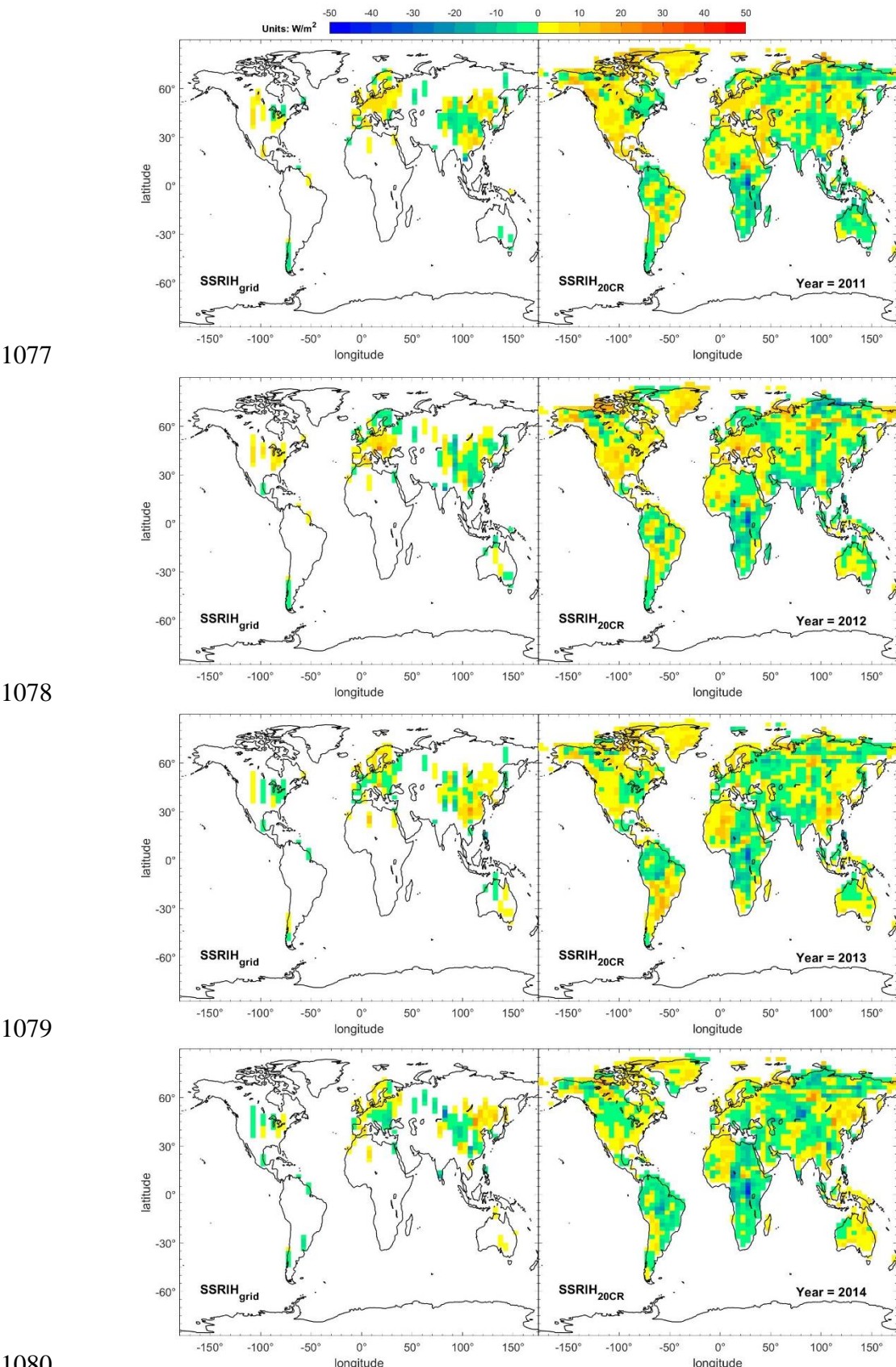




**Figure S6-15: Spatial distribution of SSRIH$_{grid}$ (column 1) and SSRIH$_{20CR}$ (column 2) in typical years (2011-2014).**







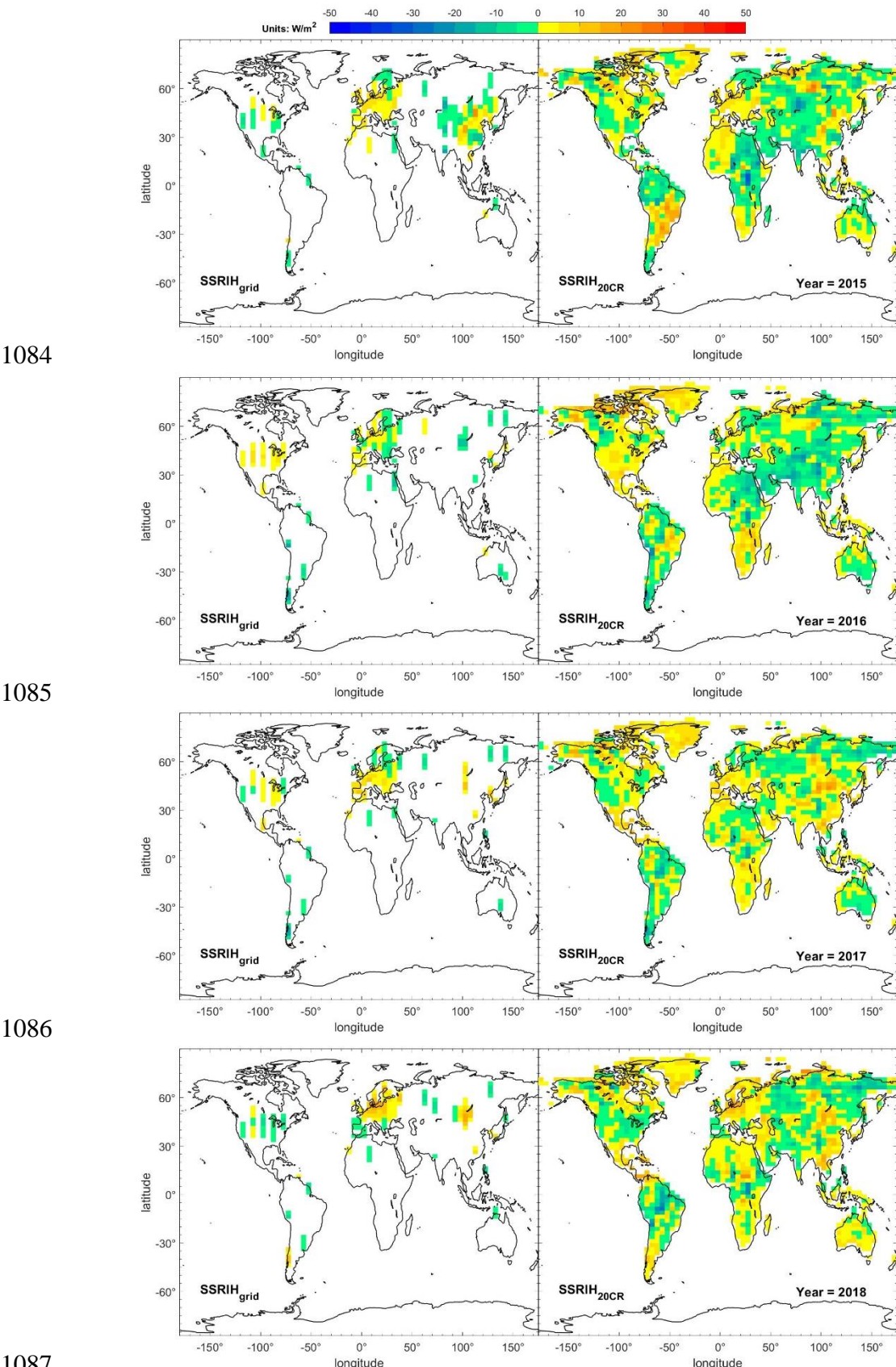

**Figure S6-16: Spatial distribution of SSRIH$_{grid}$ (column 1) and SSRIH$_{20CR}$ (column 2) in typical years (2015-2018).**



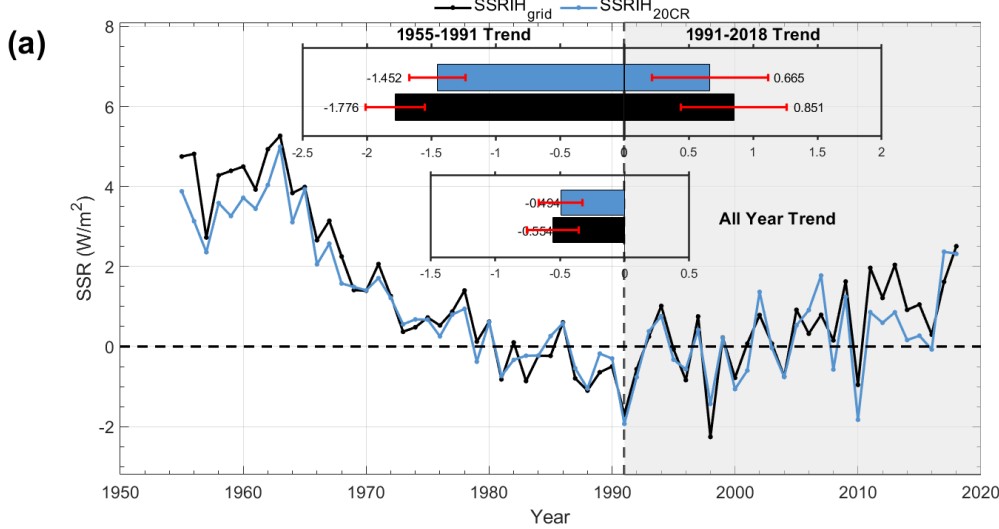


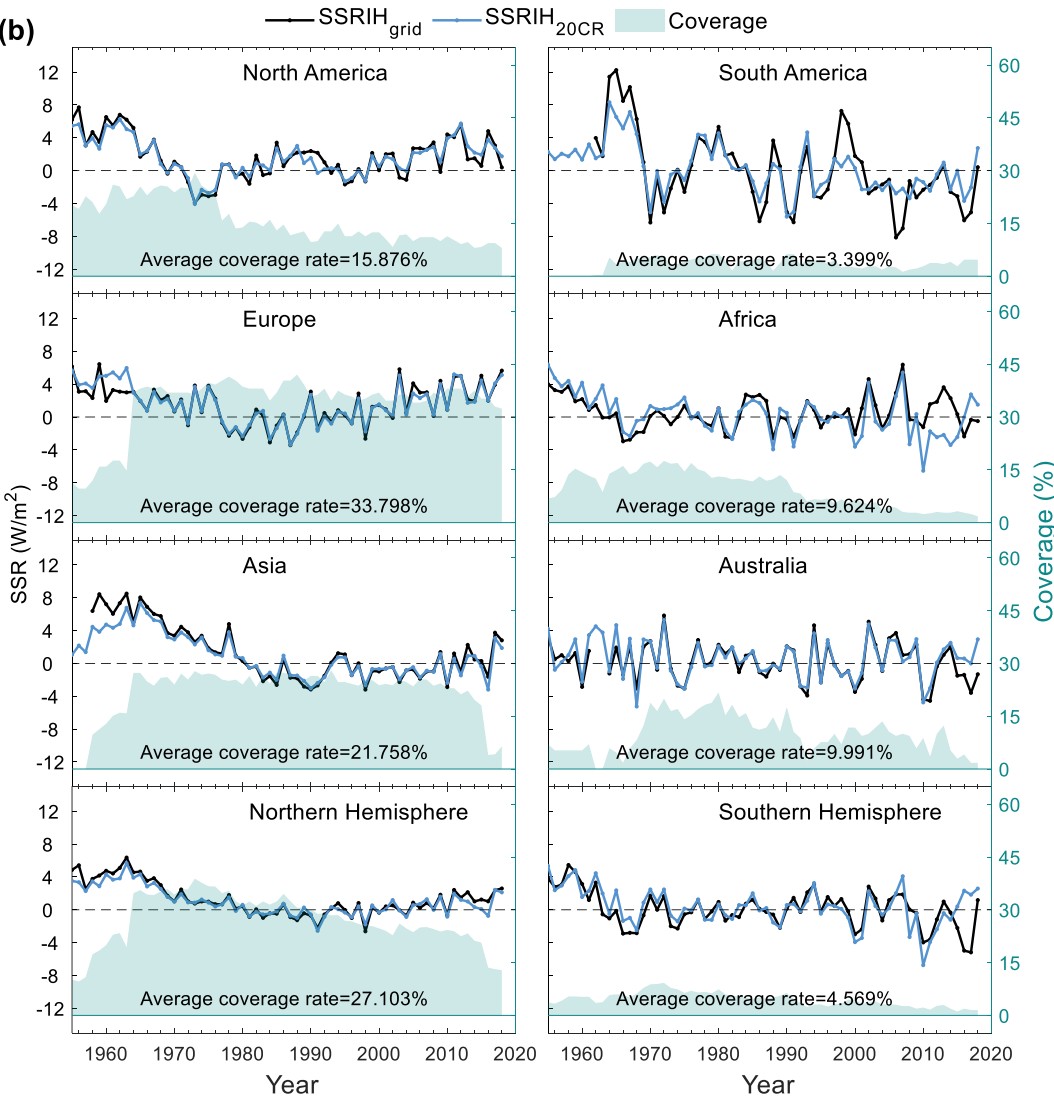


**Figure S7: Global and regional (except for Antarctica) land annual SSR anomaly variations (relative to 1971-2000) before/after reconstruction. The Black solid line represents the SSRIH_grid annual anomalies. The solid blue line represents the reduced SSRIH_20CR annual anomalies. The histograms represent the decadal**

**trends of the $SSRIH_{grid}$ /$SSRIH_{20CR}$ (unit: W/m2 per decade) and their 95% uncertainty range from 1955 to**
**1991, 1991-2018 and 1955-2018, and the $SSRIH_{20CR}$ is reduced to the grid boxes with *in situ* observations.**

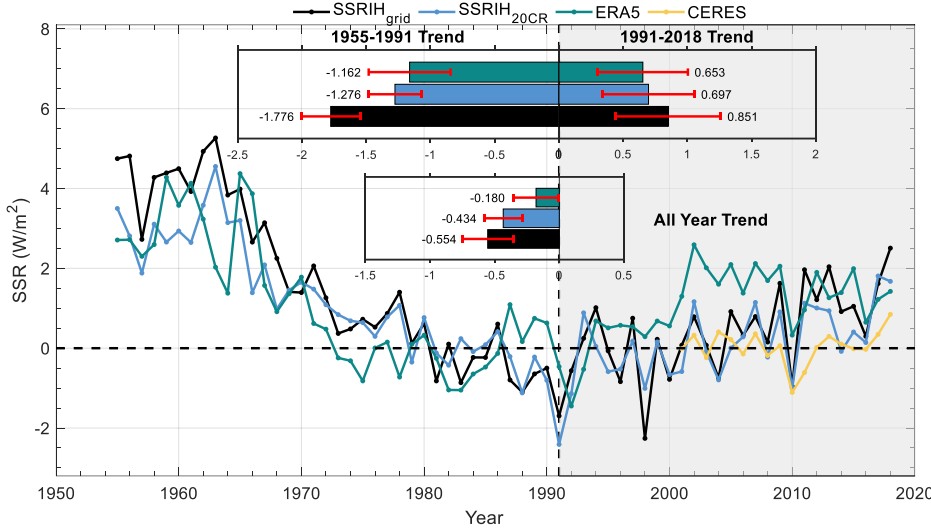


**Figure S8: Global land (except for Antarctica) annual SSR anomaly variations (relative to 1971-2000)**

**before/after reconstruction. The Black solid line represents the SSRIH$_{grid}$ annual anomalies. The solid blue**

**line represents the SSRIH$_{20CR}$ annual anomalies. The solid green line represents the ERA5 annual anomalies.**

**The solid yellow line represents the CERES annual anomalies. The histograms represent the decadal trends**

**of the SSRIH$_{grid}$ /SSRIH$_{20CR}$ / ERA5 (unit: W/m$^2$ per decade) and their 95% uncertainty range from 1955 to**

**1991, 1991-2018 and 1955-2018.**

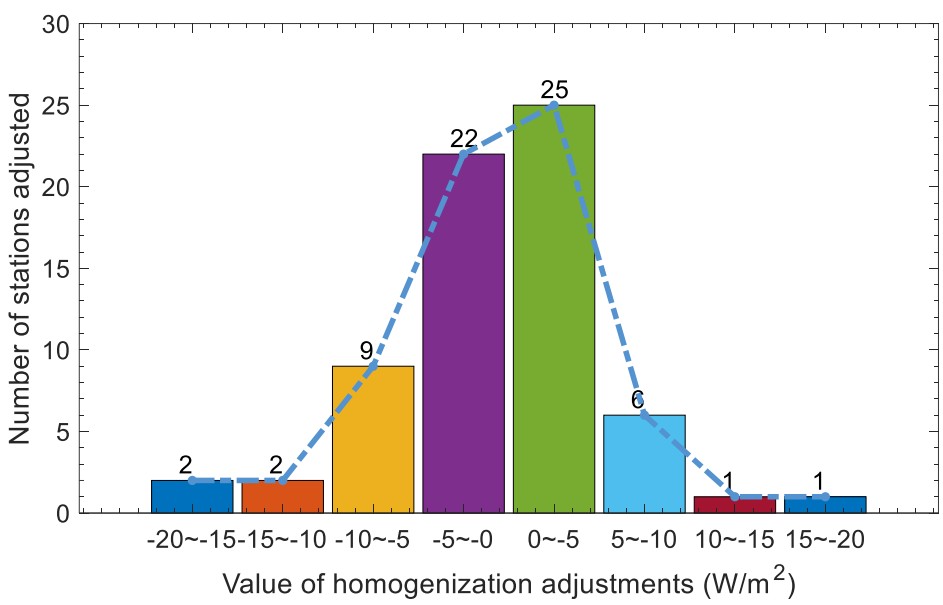


**Figure S9: Distribution of annual SSR homogenization adjustments.**
(The histogram is based on adjustments from all 66 stations adjusted in this paper)

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
