# Peer review of "An integrated and homogenized global surface solar"

_Earth System Science Data, 2023_

## Author Comment (AC1)

**Response to the Reviewer's Comments**

**Reviewer's comments:**

This manuscript reconstructs the first global, long-term (1955 - 2018), gap-free, gridded surface solar radiation (SSR) dataset by integrating nine SSR datasets using a CNN method. The main inputs include long-term ground networks, regional homogenized products, ERA reanalysis, etc. Overall, the proposed dataset is significant to the research community and will play an important role in evaluating climate modeling and analyzing global dimming and brightening. In addition, the authors did very comprehensive work in data processing. I would recommend the author include more product inter-comparison with other long-term SSR products to demonstrate the reliability and superiority of the proposed dataset. Some details of the data processing are still required.

**Response:**

Thank you for the positive comments and your suggestions concerning our manuscript (essd-2023-178).

These comments /suggestions are all valuable and very helpful for revising and improving our manuscript, as well as the important guiding significance to our research. We have studied the comments carefully and have made corrections which we hope meet with approval. As suggested by the reviewer, we have compared our dataset with the ERA5 and CERES data, and the results will be shown in the SI. The main corrections to the manuscript and responses to the reviewer's comments are as follows.

**Major:**

1. The manuscript adequately emphasizes the importance of a long-term SSR dataset for global dimming analysis but lacks a review of existing SSR datasets. Since ESSD is a data journal, I recommend including a paragraph about existing SSR datasets and their limitations in the literature review section.

**Response:** Thank you for the above suggestions.

We have systematically reviewed the limitations of existing SSR datasets in the previous paper (Jiao et al., 2022).

In the Introduction section of this manuscript, the second paragraph includes a description of some existing SSR datasets and their limitations but omits a review of the reanalysis and modal data. We have revised the Introduction section and added more references.

Specifically:

Lines 55-72 included a review of station datasets and their limitations. We present SSR datasets with global and regional coverage and point out their inhomogeneity and limited coverage.

Lines 76-80 provided a review of existing SSR satellite datasets and their limitations. We have made some additional revisions to this section. The new additions are as follows: "The spatial, temporal, and spectral coverage of a single satellite is limited, and multiple satellite data are therefore often used in tandem with each other; however, such a discontinuity in time and space can introduce inhomogeneity into a dataset (Evan et al., 2007; Feng and Wang, 2021; Shao et al., 2022)."

We have included a paragraph about existing reanalysis and model SSR datasets and their limitations in the literature review section. The new additions are as follows: "Reanalysis products are an important complement containing long-term SSR data, therefore have been widely used in climate studies (Zhou et al. 2017; Huang et al. 2018; Urraca et al. 2018a; Zhou et al. 2018; Jiao et al., 2022) due to the dynamically consistent and spatiotemporally complete atmospheric fields with high resolution and open access to data. However, existing studies have shown that reanalysis products generally overestimate multi-year mean SSR values compared to observations over land. With the continuous development of climate system simulations, model data from the Coupled Model International Program (CMIP) have become an important resource for conducting climate change research (Gates et al., 1999; Zhou et al., 2019). Previous studies have shown that the models used in CMIP6 overestimate the global mean SSR (Wild, 2020; Jiao et al., 2022; He, et al., 2023;)."

Lines 81-88 presented a brief review of SSR reconstruction using a machine learning approach.

Reference

Evan, A.T., Heidinger, A.K., Vimont, D.J., 2007. Arguments against a physical long-term trend in global ISCCP cloud amounts. Geophys. Res. Lett. 34 (4), L04701.

Feng, F., Wang, K., 2021. Merging high-resolution satellite surface radiation data with meteorological sunshine duration observations over China from 1983 to 2017. Remote Sens. 13 (4), 602.

Shao, C., Yang, K., Tang, W., He, Y., Jiang, Y., Lu, H., Fu, H., and Zheng, J.: Convolutional neural network-based homogenization for constructing a long-term global surface solar radiation dataset, Renewable and Sustainable Energy Reviews, 169, 10.1016/j.rser.2022.112952, 2022.

Gates, W. L., Boyle, J. S., Covey, C., Dease, C. G., Doutriaux, C. M., Drach, R. S., Fiorino, M., Gleckler, P. J., Hnilo, J. J., Marlais, S. M., Phillips, T. J., Potter, G. L., Santer, B. D., Sperber, K. R., Taylor, K. E., and Williams, D. N.: An Overview of the Results of the Atmospheric Model Intercomparison Project (AMIP I), Bulletin of the American Meteorological Society, 80, 29-55, 10.1175/15200477(1999)080<0029:Aootro>2.0.Co;2, 1999.

He, J., Hong, L., Shao, C., and Tang, W.: Global evaluation of simulated surface shortwave radiation in CMIP6 models, Atmospheric Research, 292,10.1016/j.atmosres.2023.106896, 2023.

Huang, J., L. J. Rikus, Y. Qin, and J. Katzfey, 2018: Assessing model performance of daily solar irradiance forecasts over Australia. Sol. Energy, 176, 615–626, https://doi.org/10.1016/ j. solener.2018.10.080.

Jiao, B., Li, Q., Sun, W., and Martin, W.: Uncertainties in the global and continental surface solar radiation variations: inter-comparison of in-situ observations, reanalyses, and model simulations, Climate Dynamics, 1-18, doi:10.1007/s00382-022-06222-3, 2022.

Urraca, R., T. Huld, F. J. Martinez-de-Pison, and A. Sanz-Garcia, 2018a: Sources of uncertainty in annual global horizontal irradiance data. Sol. Energy, 170, 873–884, https://doi.org/ 10.1016/j.solener.2018.06.005.

Wild, M.: The global energy balance as represented in CMIP6 climate models, Clim Dyn, 55, 553-577, 10.1007/s00382-020-05282-7, 2020.

Zhou, C., and Q. Ma, 2017: Evaluation of eight current reanalyses in simulating land surface temperature from 1979 to 2003 in China. J. Climate, 30, 7379–7398, https://doi.org/10.1175/JCLID-16-0903.1.

Zhou, C., Y. He, and K. Wang, 2018: On the suitability of current atmospheric reanalyses for regional warming studies over China. Atmos. Chem. Phys., 18, 8113–8136, https://doi.org/ 10.5194/acp-18-8113-2018.

Zhou, W., Gong, L., Wu, Q., Xing, C., Wei, B., Chen, T., Zhou, Y., Yin, S., Jiang, B., Xie, H., Zhou, L., and Zheng,

S.: Correction to: PHF8 upregulation contributes to autophagic degradation of E-cadherin, epithelialmesenchymal transition and metastasis in hepatocellular carcinoma, J Exp Clin Cancer Res, 38, 445,

10.1186/s13046-019-1452-0, 2019.

2. The evaluation methods utilized in this work primarily compared ground-measured series and CMIP6 simulations, which may have ignored uncertainty due to the latter's limitations. To strengthen the paper, I suggest providing additional comparisons with independent global datasets, such as short-term remote sensing data or long-term reanalysis datasets, to demonstrate the proposed data's temporal stability, long period, and high accuracy.

**Response:**

In fact, we did not compare ground-measured series and CMIP6 simulations, but only used the CMIP6 SSR data as a training set to develop our CNN model used in this manuscript.

As suggested by the reviewer, we have compared our dataset with ERA5 and CERES data, and the results are shown below. We will include these results in the SI.

[Figure]

**Figure S8: Global land (except for Antarctica) annual SSR anomaly variations (relative to 1971-2000) before/after reconstruction. The Black solid line represents the SSRIH$_{grid}$ annual anomalies. The solid blue line represents the SSRIH$_{20CR}$ annual anomalies. The solid green line represents the ERA5 annual anomalies.**

**The solid yellow line represents the CERES annual anomalies. The histograms represent the decadal trends of the SSRIH$_{grid}$ /SSRIH$_{20CR}$ / ERA5 (unit: W/m$^2$ per decade) and their 95% uncertainty range from 1955 to 1991, 1991-2018 and 1955-2018.**

3. The description of the CNN modelling in section 3.2 needs clarity. Please provide details about the sampling of input data and the measures taken to prevent overfitting.

**Response:** Thank you for the comments.

A description of the input data (no sampling) is given in Section 5.1, lines 380-385, and a more detailed description of the CNN is given in the SI.

In this manuscript (SI), we will add descriptions of the measures taken to prevent overfitting of the CNN modelling. "We set the batch size to 16 in the first 500000 iterations and fine-tuned it to 18 in the last 10000000 iterations, for a total of 1500000 iterations, to suppress the overfitting phenomenon generated during the training process, and validate the model every 10000 times and early stopping if the validation shows a decreasing trend, the final number of training times used is 1100000. Second, L2 regularization is also added to regulate the loss function.

$$\mathcal{L}_{total} = \mathcal{L}_{valid} + 6\mathcal{L}_{hole} + 0.05\mathcal{L}_{perceptual} + 120\left(\mathcal{L}_{style_{out}} + \mathcal{L}_{style_{comp}}\right) + 0.1\mathcal{L}_{tv} + \alpha\|\omega\|_2^2,$$

4. Additionally, Figure 4 clarifies how one CMIP6/20CR model was selected from their ensembles. Randomly selecting one model as input may lead to errors due to biases among different CMIP6 models.

**Response:** Thank you for your comments.

Rather than randomly selecting one model as input, we selected all 80 members of the 20CR as input (1 for evaluation and to test reconstruction, the other 79 for training the CNN model). Similarly, we selected 125 members out of a total of 507 members from several CMIP6 large ensemble models (with more than 10 realizations/runs) with high correlation coefficients with observations as input to train and validate the CNN model (1 for evaluation and to test reconstruction, the other 124 for training the CNN

model).

We have slightly revised Figure 4 to avoid ambiguity.

[Figure]

**Figure 4: Flowchart of AI reconstruction.**

5. In Figure S1, there appears to be a bias between the homogenized series and the original observation series for several years, followed by a good match. While I understand that the homogenization algorithm revised the series, I am concerned that this processing may introduce discontinuities in the time dimension. Please address this issue, as it is observed in nearly all sites.

**Response:** We totally understand your concern.

Figure S1 shows a comparison of the interannual variability of the station series before and after homogenization. To succeed in future observations, it is generally assumed that the most recent series are correct, while only the previous series are adjusted. This situation is therefore exactly the phenomenon caused by homogenization adjustments.

6. Considering that the proposed dataset covers 1955-2018 after reconstruction, it would be valuable to discuss the benefits of the product compared to long-term reanalysis data. It is important to acknowledge that reanalysis data also assimilate actual observations globally.

**Response:** Thank you for this comment.

This manuscript discusses the adjustment and reconstruction of *in situ* observational data, which serves as a benchmark for other comprehensive datasets, such as satellites, reanalyzes and model simulations.

As the reviewer points out, the reanalysis data assimilates some observations, but it is based on a state-of-the-art model and assimilation system. It does not contain a time function and is therefore affected by the data numbers, types, or quality of the assimilated observations.

7. 'data quality' or 'quality check' information should be given in the data file, which is required by the ESSD.

**Response:** Thank you for your advice.

The quality control procedure for the observations used in this manuscript includes extreme value checking, internal and spatial consistencies, etc. However, as our data sources (including the GEBA dataset, WRDC, CMA, etc.) have been systematically quality controlled by the data providers, the quality control of the raw data sources is not the main focus of this manuscript.

**Other:**

1. Title: Change "an artificial intelligence method" to "convolutional neural network" for more precise terminology. Additionally, please note that "AI" is often associated with models that can perform or think like human beings, which differs from "machine learning" or even "CNN."

**Response:** Thank you for your rigorous consideration. We changed the title to "An integrated and homogenized global surface solar radiation dataset and its reconstruction based on a convolutional neural network approach"

2. Line 105: Specify that there are 125 total inputs.

**Response:** Thank you very much for your comments. We selected 125 members out of a total of 507 members from several CMIP6 large ensemble models (with more than 10 realizations /runs) with high correlation coefficients with observations as input to train and validate the CNN model (1 for evaluation and to test reconstruction, the other 124 for training the CNN model).

3. Line 183: Revise the phrases "much better" and "excellent resource" or provide evidence to support these claims. Carefully review the entire context and be mindful of similar words.

**Response:** Thank you for pointing out this error.

Changed "Compared to previous model comparison projects, the CMIP6 project has a much better experimental design and more model development centres involved, as well as providing a much more significant amount of data." to "Specifically, CMIP6 is considered as the current state of the art way of producing future climate simulations, including predicting future SSR based on different climate scenarios (Zhou et al, 2019)."

Changed "excellent resource" to "important resource"

Reference

Zhou, W., Gong, L., Wu, Q., Xing, C., Wei, B., Chen, T., Zhou, Y., Yin, S., Jiang, B., Xie, H., Zhou, L., and Zheng, S.: Correction to: PHF8 upregulation contributes to autophagic degradation of E-cadherin, epithelialmesenchymal transition and metastasis in hepatocellular carcinoma, J Exp Clin Cancer Res, 38, 445, 10.1186/s13046-019-1452-0, 2019.

4. Line 206: Explain the rationale behind "five times." Consider removing years associated with major global volcanic eruptions (e.g., 1992) if they might impact the analysis.

**Response:**

We are very sorry for our negligence of the clerical error. It should be three times the standard deviation. The $3\sigma$ criterion is also called PauTa criterion, which assumes that a group of data obeys or approximately obeys the normal distribution and only contains random errors. The standard deviation of this set of data is calculated and an interval is determined according to a certain probability. It is considered that the error outside this interval is a gross error rather than a random error, which should be eliminated (Olanow et al ,1998). Based on this criterion, 247 records were deleted. This represents approximately 0.4% of all station records.

In this manuscript, since we reconstruct the monthly SSR data through a CNN approach (image inpainting without time function as mentioned above), the extreme values associated with global volcanic eruptions (which may be spatially responded to) do not have a significant effect on the reconstruction.

Reference

Olanow C W, Koller W C. 1998 An algorithm (decision tree) for the management of Parkinson's disease: Treatment guidelines vol 50 no 3 (Neurology).

5. Line 223: Clarify the meaning of "potential reference pool" in this context.

**Response:** Thanks for your question. The potential reference pool contains all stations that can be used as reference series (Xu et al, 2013).

Reference

Xu, W., Li, Q., Wang, X. L., Yang, S., Cao, L., and Feng, Y.: Homogenization of Chinese daily surface air temperatures and analysis of trends in the extreme temperature indices, Journal of Geophysical Research: Atmospheres, 118, 9708-9720, doi:10.1002/jgrd.50791, 2013.

6. Line 274: Change "non-hole" to "gap-free."

**Response:** Changed. Thanks.

7. Line 294: Address the concern that taking a simple average between two sites with different time spans (e.g., 1950-1970 and 1960-1980) may result in discontinuity.

**Response:**

Thank you so much for your careful check.

In this manuscript, we followed the climate anomaly method (CAM) to calculate the global, regional and grid box average SSR change (Jones et al, 2001; Sun et al, 2021; Li et al, 2021). In a single 5*5 grid box, we also calculate the average climate anomalies among all stations, which avoids the problems you mention by calculating the simple average of the absolute values (Li et al., 2009).

Reference

Jones, P., Osborn, T., Briffa, K., Folland, C., Horton, E., Alexander, L., Parker, D., and Rayner, N.: Adjusting for sampling density in grid box land and ocean surface temperature time series, Journal of Geophysical Research: Atmospheres, 106, 3371-3380, doi:10.1029/2000JD900564, 2001.

Sun, W., Li, Q., Huang, B., Cheng, J., Song, Z., Li, H., Dong, W., Zhai, P., and Jones, P.: The Assessment of Global Surface Temperature Change from 1850s: The C-LSAT2.0 Ensemble and the CMST-Interim Datasets, Advances in Atmospheric Sciences, 38, 875-888, 10.1007/s00376-021-1012-3, 2021.

Li Q, Sun W, Yun X, Huang B, Dong W, Wang X, Zhai P and Phil Jones: An updated evaluation of the global mean Land Surface Air Temperature and Surface Temperature trends based on CLSAT and CMST, Climate Dynamics, 56:635-650, DOI: 10.1007/s00382-020-05502-0, 2021.

Li W, Li Q, Jiang Z: Discussion on Feasibility of Gridding the Historic Temperature Data in China with Kriging Method, Journal of Nanjing Institute of Meteorology, 30(2): 246-252, 2009.

8. Line 216: Specify the extrapolation methods used.

**Response:** No extrapolation is used in this manuscript.

9. All trend statistics should include a significance test.

**Response**: Thanks for the reminder. A table of trends (including a significance test) and their uncertainties for each region is presented below and attached to the SI.

Table S3 Trends evaluation in Continental and hemispheric $SSRIH_{20CR}$ change from different scales (Units: $W/m^2$ per decade).

| Continental | Time period /Trend | Time period /Trend |
|---|---|---|
| North America | 1955-1973 | 1973-2018 |
| | -3.588±1.290 | 1.074±0.278 |
| South America | 1955-1990 | 1990-2018 |
| | -0.408±0.619 | 0.049±0.768 |
| Europe | 1963-1978 | 1978-2018 |
| | -2.180±1.866 | 1.081±0.312 |
| Africa | 1955-1991 | 1991-2018 |
| | -1.506±0.496 | 0.340±0.998 |
| Asia | 1955-1990 | 1990-2018 |
| | -1.633±0.473 | 0.435±0.505 |
| North Hemisphere | 1955-1991 | 1991-2018 |
| | -1.457±0.246 | 0.887±0.415 |
| South Hemisphere | 1955-1991 | 1991-2018 |
| | -0.708±0.330 | -0.076±0.656 |

Table S4 Trend assessment in various data sources Global SSR change from different scales (units: W/m2 per decade).

| Type | 1955-1991 | 1991-2018 | 1955-2018 |
|---|---|---|---|
| $SSRI_{grid}$ | -1.995±0.251 | 0.999±0.504 | -0.494±0.228 |
| $SSRIH_{grid}$ | -1.776±0.230 | 0.851±0.410 | -0.554±0.197 |
| $SSRIH_{20CR}$ | -1.276±0.205 | 0.697±0.359 | -0.434±0.148 |
| ERA5 | -1.162±0.319 | 0.653±0.350 | -0.180±0.176 |

---

## Author Comment (AC2)

**Reviewer #2:**

This manuscript is interesting and convincing. The Manuscript develops the first, long-term (1955-2018), homogenized, gap-free global land SSR anomalies dataset by training improved partial convolutional neural network deep learning methods. Authors analyzed the global land (except for Antarctica) /regional scale SSR trends and spatio-temporal variations. Comparative validations /evaluations show that the $SSRIH_{20CR}$ provides a reliable benchmark for global SSR variations. Therefore, this manuscript may be considered for formal publication with minor modifications after addressing the following issues:

**Response:**

Thank you for the positive comments and your suggestions concerning our manuscript (essd-2023-178). These comments /suggestions are all valuable and very helpful for revising and improving our manuscript, as well as the important guiding significance to our research. We have studied the comments carefully and made corrections (please refer to the detailed revision after each comment) which we hope to get approval.

1. The resolution of the SSR data in this paper is only 5°×5°. Why not develop a product with higher resolution? What's the difficulty? Is it necessary?

**Response:** Thanks for your question.

Firstly, the reason why we did not develop a higher-resolution SSR dataset is the scarcity of the *in situ* observations. There are only about 1,000 compliant SSR sites worldwide. Even if we obtain higher resolution SSR datasets through interpolation techniques, they are only based on objective analysis in mathematical methods and do not add more SSR observation information, nor can they represent local scale SSR changes. Secondly, the SSR anomaly dataset in this manuscript is a benchmark dataset, which is designed to reflect large-scale SSR climate change. The global SSR data in 5°× 5° resolution already represents the long-term changes in SSR, as SSR and

temperature are similar in that they are both highly spatially representative.

2. Remote sensing inversion based on satellite measurements or some fusion products can provide space-time continuous SSR data, whether global or regional. I suggest that the authors clarify the reason why is the long-term trends of SSR data in this paper quite different from the current high-resolution satellite fusion data?

**Response:** Thanks for your question.

Firstly, the high-resolution satellite fusion datasets cover a too short period to investigate their decadal and multi-decadal variations.

Secondly, note that satellite fusion SSR datasets is also largely a modelled product, since satellites can only accurately measure the TOA fluxes, but not at the surface, since the atmosphere perturbs the surface signal received at the satellite sensor. Therefore, although it is a good estimation, it can still deviate from the real world (Zhang et al., 2016).

Thirdly, the spatial, temporal, and spectral coverage of a single satellite is limited, and multiple satellite data are therefore often used in tandem with each other; however, such a discontinuity in time and space can introduce inhomogeneity into a dataset (Tang et al., 2019; Feng and Wang, 2021; Shao et al., 2022).

Finally, the purpose of the application differs between the two datasets. The spatial resolution of the satellite fusion datasets are higher than those of the products of our dataset and will contribute to high-resolution photovoltaic applications. The SSR dataset in this manuscript is a benchmark dataset, which is designed to reflect large-scale SSR climate change.

Therefore, many aspects (including long-term trends) of SSR data in this manuscript are quite different from the current high-resolution satellite fusion data.

Reference:

Feng, F., Wang, K., 2021. Merging high-resolution satellite surface radiation data with meteorological sunshine duration observations over China from 1983 to 2017. Remote Sens. 13 (4), 602.

Shao, C., Yang, K., Tang, W., He, Y., Jiang, Y., Lu, H., Fu, H., and Zheng, J.: Convolutional neural network-based homogenization for constructing a long-term global surface solar radiation dataset, Renewable and Sustainable Energy Reviews, 169, 10.1016/j.rser.2022.112952, 2022.

Tang W., Yang, K., Qin, J, Li, X., Niu, X.: A 16-year dataset (2000–2015) of high-resolution (3 h, 10 km) global surface solar radiation. Earth System Science Data, 11, 1905-1915, 2019.

Zhang X, Liang S, Wang G, Yao Y, Jiang B, Cheng J. Evaluation of the reanalysis surface incident shortwave radiation products from NCEP, ECMWF, GSFC, and JMA using satellite and surface observations. Rem Sens 2016;8(3):225.

3. The introduction provides a detailed overview of existing SSR datasets. However, the limitations of existing datasets are described rather briefly.

**Response:** Thank you for the above suggestions. It is really true as the first Reviewer suggested that we need to include a paragraph about existing SSR datasets and their limitations in the literature review section.

In the Introduction section of this manuscript, the second paragraph includes a description of some existing SSR datasets and their limitations but the limitations of existing datasets are described rather briefly. We have revised the Introduction section and added more references.

We provided a review of existing SSR satellite datasets and their limitations (**Page 4, Lines 78-83**). We have made some additional revisions to this section. The new additions are as follows: "The spatial, temporal, and spectral coverage of a single satellite is limited, and multiple satellite data are therefore often used in tandem with each other; however, such a discontinuity in time and space can introduce inhomogeneity into a dataset (Evan et al., 2007; Feng and Wang, 2021; Shao et al., 2022)." (**Page 4, Lines 83-86**)

We have included a paragraph about existing reanalysis and model SSR datasets and their limitations in the literature review section. The new additions are as follows: "Reanalysis products are an important complement containing long-term SSR data, therefore have been widely used in climate studies (Zhou et al. 2017; Huang et al. 2018; Urraca et al. 2018a; Zhou et al. 2018; Jiao et al., 2022) due to the dynamically consistent and spatiotemporally complete atmospheric fields with high resolution and open access

to data. However, existing studies have shown that reanalysis products generally overestimate multi-year mean SSR values compared to observations over land (He, et al., 2022). With the continuous development of climate system simulations, model data from the Coupled Model International Program (CMIP) have become an important resource for conducting climate change research (Gates et al., 1999; Zhou et al., 2019). Previous studies have shown that the models used in CMIP6 overestimate the global mean SSR (Wild, 2020; Jiao et al., 2022; He, et al., 2023)." (**Page 4, Lines 86-95**)

Reference

Evan, A.T., Heidinger, A.K., Vimont, D.J., 2007. Arguments against a physical long-term trend in global ISCCP cloud amounts. Geophys. Res. Lett. 34 (4), L04701.

Feng, F., Wang, K., 2021. Merging high-resolution satellite surface radiation data with meteorological sunshine duration observations over China from 1983 to 2017. Remote Sens. 13 (4), 602.

Shao, C., Yang, K., Tang, W., He, Y., Jiang, Y., Lu, H., Fu, H., and Zheng, J.: Convolutional neural network-based homogenization for constructing a long-term global surface solar radiation dataset, Renewable and Sustainable Energy Reviews, 169, 10.1016/j.rser.2022.112952, 2022.

Gates, W. L., Boyle, J. S., Covey, C., Dease, C. G., Doutriaux, C. M., Drach, R. S., Fiorino, M., Gleckler, P. J., Hnilo, J. J., Marlais, S. M., Phillips, T. J., Potter, G. L., Santer, B. D., Sperber, K. R., Taylor, K. E., and Williams, D. N.: An Overview of the Results of the Atmospheric Model Intercomparison Project (AMIP I), Bulletin of the American Meteorological Society, 80, 29-55, 10.1175/15200477(1999)080<0029:Aootro>2.0.Co;2, 1999.

He, J., Hong, L., Shao, C., and Tang, W.: Global evaluation of simulated surface shortwave radiation in CMIP6 models, Atmospheric Research, 292,10.1016/j.atmosres.2023.106896, 2023.

He, Y., Wang, K., and Feng, F.: Improvement of ERA5 over ERA-Interim in simulating surface incident solar radiation throughout China, Journal of Climate, 34, 3853-3867, 2021.

Huang, J., L. J. Rikus, Y. Qin, and J. Katzfey, 2018: Assessing model performance of daily solar irradiance forecasts over Australia. Sol. Energy, 176, 615–626, https://doi.org/10.1016/ j. solener.2018.10.080.

Jiao, B., Li, Q., Sun, W., and Martin, W.: Uncertainties in the global and continental surface solar radiation variations: inter-comparison of in-situ observations, reanalyses, and model simulations, Climate Dynamics, 1-18, doi:10.1007/s00382-022-06222-3, 2022.

Urraca, R., T. Huld, F. J. Martinez-de-Pison, and A. Sanz-Garcia, 2018a: Sources of uncertainty in annual global horizontal irradiance data. Sol. Energy, 170, 873–884, https://doi.org/ 10.1016/j.solener.2018.06.005.

Wild, M.: The global energy balance as represented in CMIP6 climate models, ClimDyn, 55, 553-577, 10.1007/s00382-020-05282-7, 2020.

4. It is proposed to provide a more detailed description of the CNN method. Better to provide details about the measures taken to prevent overfitting.

**Response:** Thank you for your comments.

In this manuscript (SM), we will add descriptions of the measures taken to prevent overfitting of the CNN modelling. "We set the batch size to 16 in the first 500000 iterations and fine-tuned it to 18 in the last 10000000 iterations, for a total of 1500000 iterations, to suppress the overfitting phenomenon generated during the training process, and validate the model every 10000 times and early stopping if the validation shows a decreasing trend, the final number of training times used is 1100000. Second, L2 regularization is also added to regulate the loss function (**Page 14, Lines 362-367**).

$$\mathcal{L}_{total} = \mathcal{L}_{valid} + 6\mathcal{L}_{hole} + 0.05\mathcal{L}_{perceptual} + 120\left(\mathcal{L}_{style_{out}} + \mathcal{L}_{style_{comp}}\right) +$$

$0.1\mathcal{L}_{tv} + \alpha\|\omega\|_2^2$" (**Page 3, Lines 54-62 in the SM**).

5. Trends for the regional scales also need to be tested for significance.

**Response:** Thanks for your reminder. A table of trends (including a significance test) and their uncertainties for each region is presented below and attached to the SM (**Pages 5-6, Lines 65-70 in the SM**).

Table S3 Trend assessment in various data sources Global SSR change from different scales (units: W/m$^2$ per decade).

| Type | 1955-1991 | 1991-2018 | 1955-2018 |
|---|---|---|---|
| SSRI$_{grid}$ | -1.995±0.251 | 0.999±0.504 | -0.494±0.228 |
| SSRIH$_{grid}$ | -1.776±0.230 | 0.851±0.410 | -0.554±0.197 |
| SSRIH$_{20CR}$ | -1.276±0.205 | 0.697±0.359 | -0.434±0.148 |
| ERA5 | -1.162±0.319 | 0.653±0.350 | -0.180±0.176 |

Table S4 Trends evaluation in Continental and hemispheric $SSRIH_{20CR}$ change from different scales (Units: $W/m^2$ per decade).

| Continental | Time period /Trend | Time period /Trend |
|---|---|---|
| North America | 1955-1973 | 1973-2018 |
| | -3.588±1.290 | 1.074±0.278 |
| South America | 1955-1990 | 1990-2018 |
| | -0.408±0.619 | 0.049±0.768 |
| Europe | 1963-1978 | 1978-2018 |
| | -2.180±1.866 | 1.081±0.312 |
| Africa | 1955-1991 | 1991-2018 |
| | -1.506±0.496 | 0.340±0.998 |
| Asia | 1955-1990 | 1990-2018 |
| | -1.633±0.473 | 0.435±0.505 |
| North Hemisphere | 1955-1991 | 1991-2018 |
| | -1.457±0.246 | 0.887±0.415 |
| South Hemisphere | 1955-1991 | 1991-2018 |
| | -0.708±0.330 | -0.076±0.656 |

6. Figure 1 &4: The font size should be bigger.

**Response:** Thank you for pointing out this problem in the manuscript. We have redrawn Figures 1 (**Page 32, Line 813**) &4 (**Page 36, Line 822**) and enlarged the font size.

[Figure]

**Figure 1: Flowchart of quality control (QC) (first step), homogenization (second step) and integration (third step).**

[Figure]

**Figure 4: Flowchart of AI reconstruction.**

7. The number of decimals should be consistent throughout. for example: Figure 9 and Line 441.

**Response:** We gratefully appreciate for your valuable suggestion. Considering the Reviewer's suggestion, we have redrawn Figure 6 (**Page 39, Lines 831-832**), Figure 9 (**Page 43, Line 852**), Figure S5b (**Page 22, Lines 119-124 in the SM**) and S7b (**Page 41, Lines 239-244 in the SM**).

We have revised the number of decimals (three valid decimals) throughout the manuscript (**Throughout**).

[Figure]

**Figure 6: Reconstruction capabilities of the AI model. (a) Global land (except for Antarctica) means time-series analysis and AI model reconstruction evaluation. The red line is the SSR of the reconstruction based on the 20CR-AI /CMIP6-AI model (SSR$_{20CR}$ /SSR$_{CMIP6}$); The grey line is the masked datasets with missing values of the SSRIH$_{grid}$. The solid black line is the 20CR and CMIP6 validation set (the SSR from the 1th member of 20CRv3 /CMIP6). (b) Comparisons of the SSR$_{20CR}$ (columns 1, 3) /SSR$_{CMIP6}$ (columns 2, 4) with the SSR from the 20CR and CMIP6 validation set. Colour bars represent counts with the same values for both. Figures also show the SSR$_{20CR}$ (SSR$_{CMIP6}$) correlation coefficient (CC), root mean squared error (RMSE) and fitting equation compared to the original dataset in different regions.**

[Figure]

**Figure 9: Same as Figure 8, but for regional annual anomaly variations. The green colour filling diagram represents the variation in grid box coverage (before reconstruction).**

[Figure]

**Figure S5: Time series of the annual global (a) /regional (b) SSR anomaly variations (relative to 1971-2000) before /after homogenization.**

[Figure]

**Figure S7: Same as Figure 8 and Figure 9, but the SSRIH$_{20CR}$ is reduced to the grid boxes with in situ observations.**

8. Some sentences need to be polished and/or improved.

For example:

Lines 50-54: They allowed for the first time the detection of decadal changes in SSR known as "dimming and brightening" (Wild et al., 2005), especially considering that they cover a longer period concerning another type of data like for example satellite data (Pfeifroth et al., 2018) even if observational data often have uneven distribution and missing data with respect to the satellite data, especially in areas with complex orography (Manara et al., 2020).

**Response:** We agree with the comments. We have split this long sentence into two simple sentences.

They allowed for the first time the detection of decadal changes in SSR known as "dimming and brightening" (Wild et al., 2005), especially considering that they cover a longer period concerning another type of data like for example satellite data (Pfeifroth et al., 2018). Even observational data often have uneven distribution and missing data with respect to satellite data, especially in areas with complex orography (Manara et al., 2020) (**Page 3, Line 52**).

Lines 353-355: At the regional scale, the $SSRIH_{grid}$ has a generally similar variation to the $SSRI_{grid}$, and the $SSRIH_{grid}$ is usually more representative of climate change than $SSRI_{grid}$ at individual 355 stations. Remove "is"

**Response:** Remove "is" (**Page 15, Line 380**).

---

## Author Response (AR2)

**Response to the reviewers and the editor**

essd-2023-178

Title: An integrated and homogenized global surface solar radiation dataset and its reconstruction based on a convolutional neural network approach

Journal: Earth System Science Data

Dear authors,

Thanks for your efforts to improve the manuscript. However, several remaining concerns and minor suggestions need to be addressed first. Kindly make the necessary revisions accordingly, and we look forward to reviewing your revised paper.

Sincerely,

Jing Wei, Editor

Response to the Editor:

Dear Dr Wei,

Thank you for your letter and for the reviewer's comments concerning our manuscript entitled "An Integrated and Homogenized Global Surface Solar Radiation Dataset and its Reconstruction Based on a Convolutional Neural Network Approach" (essd-2023-178).

We have revised the manuscript and responded to the second round of comments carefully which we hope will meet with approval. The "error estimation and source of error" would be applicable to the first half of this manuscript, but not to the reconstructed dataset. We cannot estimate the reconstruction uncertainty range because we did not adopt an ensemble reconstruction under different parameters (not applicable to the AI or similar approaches). Also, a little strange to see that there was much different judgment in the "originality" and "uniqueness" of the manuscript in reviewer #1's two rounds of comments.

Moreover, as suggested by Polina Shvedko from the editorial office, we have combined the manuscript and the supplement into one file.

The main corrections in the paper and the responses to the reviewer's comments are as follows.

Qingxiang Li

2023-08-30

**Response to the Reviewer's Comments**

**Reviewer's comments:**

**Reviewer #1:**

I appreciate the authors' responses to my comments. However, I still have several queries regarding the manuscript and its corresponding responses:

1. In alignment with the review criteria outlined in this guide (https://www.earth-system-science-data.net/peer_review/review_criteria.html), it is imperative to furnish "error estimates and sources of error." in the data files (Item 2)

**Response:** Thank you for your comments.

The "error estimation and source of error" is applicable to the first half of this manuscript (data homogenized and gridded dataset), but not to the reconstructed dataset. We cannot estimate its uncertainties because we did not develop an ensemble reconstruction under different parameters.

The sources of error in the observational dataset can be divided into three types: (1) station error, the uncertainties of individual station anomalies; Including measurement errors (which are not the focus of the considerations in this manuscript) and errors due to homogenization. The errors due to homogenization adjustment are always approximately normally distributed (Jones et al., 2008, see their Figure 5; also see Figure S9 in the SM and below) and therefore have limited impacts on the global average SSR change (Figure S5 a, b). (2) sampling error, the uncertainties in a grid box mean caused by estimating the mean from a small number of point values (Jones et al., 1997); and (3) bias error. It generally refers to systematic errors such as urbanization together, which has not been discussed here. However, even the sum of the above errors is much smaller than the errors due to limited data coverage (Li, et al., 2010, see their Figure 5). So, the focus of this study is to eliminate this kind of error through the CNN reconstruction.

We have added the above description to the manuscript (**Pages 18-19, Lines 484-494**).

[Figure]

**Figure S9 Distribution of annual SSR homogenization adjustments.**

(The histogram is based on adjustments from all 66 stations adjusted in this paper)

**Page 82, Lines 1111-1113**

Reference:

Jones, P. D., Lister, D. H., and Li, Q.: Urbanization effects in large-scale temperature records, with an emphasis on China, *Journal of Geophysical Research*, 113, 10.1029/2008jd009916, 2008.

Jones, P. D., Osborn, T. J., and Briffa, K. R.: Estimating Sampling Errors in Large-Scale Temperature Averages, *Journal of Climate*, 10, 2548-2568, 1997.

Li, Q., Dong, W., Li, W., Gao, X., Jones, P., Kennedy, J., and Parker, D.: Assessment of the uncertainties in temperature change in China during the last century, *Chinese Science Bulletin*, 55, 1974-1982, 10.1007/s11434-010-3209-1, 2010.

2. A comparative analysis with CERES in the response letter reveals a substantially amplified annual variability in the proposed dataset post-2000. Could you offer plausible explanations for this disparity?

In addition, the overall trend of ERA is negative after 2004, which is completely different from the positive trend of SSRIHGrid, while CERES is stable

correspondingly. Any explanations about this point? CERES has been widely consdiered as the benchmark for SSR reanalysis and GCM assessment, while here the proposed dataset shows clear difference in variablity and trend.

**Response:** Thanks for your question.

It is generally believed that the most reliable benchmark data is still *in situ* observation data for local SSR change in a certain station. CERES has been considered as the benchmark for SSR reanalysis and GCM assessment because of its comprehensive coverage (Wild, M., 2009). However, note that CERES SSRs are also largely a modelled product since satellites can only accurately measure the TOA fluxes, but not at the surface, since the atmosphere perturbs the surface signal received at the satellite sensor (Wild, M., 2016; 2020). And the difference in the SSR trends since 2000 from *in situ* observations and satellites (including CERES) has been extensively discussed in the previous studies (Wild, M., 2012).

We have also got the same comparison result between the *in situ* and ERA5 SSR in the previous paper (Jiao et al., 2022). In this manuscript, we may not give the exact reasons why these are different, but the *in situ* observed SSRs before and after reconstruction show highly consistent long-term /short-term trends, which suggests that our reconstruction does not bring much inhomogeneity into the global mean SSR series.

Reference:

Jiao, B., Li, Q., Sun, W., and Wild, M.: Uncertainties in the global and continental surface solar radiation variations: inter-comparison of in-situ observations, reanalyses, and model simulations, *Climate Dynamics,* 1-18, doi:10.1007/s00382-022-06222-3, 2022.

Wild, M.: Global dimming and brightening: A review, *Journal of Geophysical Research*, 114, 10.1029/2008jd011470, 2009.

Wild, M.: Enlightening global dimming and brightening, *Bulletin of the American Meteorological Society*, 93, 27-37, doi:10.1175/BAMS-D-11-00074.1, 2012.

Wild, M.: Decadal changes in radiative fluxes at land and ocean surfaces and their relevance for global warming, *WIREs Climate Change*, 7, 91-107, https://doi.org/10.1002/wcc.372, 2016.

Wild, M.: The global energy balance as represented in CMIP6 climate models, *Climate Dynamics*, 55, 553-577, 10.1007/s00382-020-05282-7, 2020.

3. Could you elucidate how the uncertainties pertaining to the trends are quantified, both in Table S3 and within the manuscript itself? Additionally, I noted the absence of any significance test and result statements, such as p-values.

**Response:** Thanks for your question.

There are two ways to show the uncertainties pertaining to the trends: the first is by giving p values as you suggest, and the second is to give the trends and their 95% confidence ranges. In this manuscript, we use the second way.

To make it clearer, we added some annotations in the revised manuscript (**Table S3 & S4, Pages 45-46, Lines 924-931**).

Table S3 Trends and their 95% confidence ranges in various data sources global SSR change (units: W/m$^2$ per decade). * Indicate trends that are significant at the 5% level.

| Type | 1955-1991 | 1991-2018 | 1955-2018 |
|---|---|---|---|
| SSRI$_{grid}$ | -1.995 ± 0.251* | 0.999 ± 0.504* | -0.494 ± 0.228* |
| SSRIH$_{grid}$ | -1.776 ± 0.230* | 0.851 ± 0.410* | -0.554 ± 0.197* |
| SSRIH$_{20CR}$ | -1.276 ± 0.205* | 0.697 ± 0.359* | -0.434 ± 0.148* |
| ERA5 | -1.162 ± 0.319* | 0.653 ± 0.350* | -0.180 ± 0.176* |

Table S4 Trends and their 95% confidence ranges in continental and hemispheric SSRIH$_{20CR}$ change (Units: W/m$^2$ per decade). * Indicate trends that are significant at the 5% level.

| Continental | Time period /Trend | Time period /Trend |
|---|---|---|
| North America | 1955-1973 | 1973-2018 |
| | -3.588 ± 1.290* | 1.074 ± 0.278* |
| South America | 1955-1990 | 1990-2018 |
| | -0.408 ± 0.619 | 0.049 ± 0.768 |
| Europe | 1963-1978 | 1978-2018 |
| | -2.180 ± 1.866* | 1.081 ± 0.312* |
| Africa | 1955-1991 | 1991-2018 |
| | -1.506 ± 0.496* | 0.340 ± 0.998 |
| Asia | 1955-1990 | 1990-2018 |
| | -1.633 ± 0.473* | 0.435 ± 0.505 |
| North Hemisphere | 1955-1991 | 1991-2018 |
| | -1.457 ± 0.246* | 0.887 ± 0.415* |
| South Hemisphere | 1955-1991 | 1991-2018 |
| | -0.708 ± 0.330* | -0.076 ± 0.656* |

4. Referring to Line 29: the trend of SSRIH20CR is nearly 28% lower than SSRIHgrid, which is hard to be considered as 'slightly smaller'. In addition, any explanations about such trend difference? There are similar issues for the brighting period.

**Response:** Thank you for pointing out this problem in the manuscript.

We have changed 'slightly smaller' to 'smaller' (**Page 2, Line 29**).

As can be seen in Figure 8, the larger differences in $SSRIH_{20CR}$ and $SSRIH_{grid}$ variations mainly appeared during the 1950s-60s and in the 2010s, which coincided with the years with relatively few stations. This phenomenon shows the reconstruction in this manuscript reduces errors /uncertainties due to limited coverage, especially for the 1950s-60s and 2010s.

Could I deduce that the CNN enhancement has led to the mitigation of anomalies in the proposed dataset from SSRIHgrid to SSRIH20CR? If this is indeed the case, has the smoothing effect extended to realistic anomalies or extremes?

**Response:** Thanks for your question.

The CNN approach learns relatively realistic SSR trends (relationships not only in time but also in space) from a large amount of data (Teuwen, J., et al., 2020). When we use these implicit relationships (CNN models) to fill in the missing data, it attempts to restore the true SSR trends, rather than determining the SSR trends. The values already in the $SSRIH_{grid}$ remain unchanged in the $SSRIH_{20CR}$ when we use CNN to reconstruct SSR. Therefore, the CNN enhancement has not led to the mitigation of anomalies in the proposed dataset from $SSRIH_{grid}$ to $SSRIH_{20CR}$. However, when more grid SSR series are applied to the calculation of global average SSR, it may result in a visual smoothing effect.

It has been proven that the lack of a fully sampled global temperature benchmark dataset has led to a certain degree of underestimation of warming relative to the pre-industrial period (Gulev et al., 2021; Sun, et al., 2021; 2022). Similarly, in this manuscript, we consider that the errors /uncertainties due to limited data coverage in

the SSRIH$_{20CR}$ have been eliminated through reconstruction by CNN, instead of the smoothing effect extended to realistic anomalies or extremes. Therefore, a more homogeneous and comprehensive global long-term SSR climatic dataset (SSRIH$_{20CR}$) can eliminate the errors due to limited coverage and provide a better benchmark for observational constraints on the global surface energy balance /budget.

Reference:

Gulev, S. K., Thorne, P. W., J. Ahn, F. J. D., Domingues, C. M., Gerland, S., Gong, D., Kaufman, D. S., Nnamchi, H. C., Quaas, J., Rivera, J. A., Sathyendranath, S., Smith, S. L., Trewin, B., Shuckmann, K. v., and Vose, R. S.: In: Climate Change 2021: The Physical Science Basis., Climate Change 2021: The Physical Science Basis. Contribution of Working Group I to the Sixth Assessment Report of the Intergovernmental Panel on Climate Change, in, edited by: [Masson-Delmotte, V., Zhai, P., Pirani, A., Connors, S. L., Péan, C., Berger, S., Caud, N., Chen, Y., Goldfarb, L., Gomis, M. I., Huang, M., Leitzell, K., Lonnoy, E., Matthews, J. B. R., Maycock, T. K., Waterfield, T., Yelekçi, O., Yu, R., and (eds.)], B. Z., Cambridge University Press. 2021., 287–422. Cambridge University Press, 2021.

Sun, W., Li, Q., Huang, B., Cheng, J., Song, Z., Li, H., Dong, W., Zhai, P., and Jones, P.: The Assessment of Global Surface Temperature Change from 1850s: The C-LSAT2.0 Ensemble and the CMST-Interim Datasets, *Advances in Atmospheric Sciences*, 38, 875-888, 10.1007/s00376-021-1012-3, 2021.

Sun, W., Yang, Y., Chao, L., Dong, W., Huang, B., Jones, P., and Li, Q.: Description of the China global Merged Surface Temperature version 2.0, *Earth System Science Data*, 14, 1677-1693, 10.5194/essd-14-1677-2022, 2022.

Teuwen, J. and Moriakov, N.: Chapter 20 - Convolutional neural networks, in: Handbook of Medical Image Computing and Computer Assisted Intervention, edited by: Zhou, S. K., Rueckert, D., and Fichtinger, G., Academic Press, 481-501, https://doi.org/10.1016/B978-0-12-816176-0.00025-9, 2020.